# ReSi: A Comprehensive Benchmark for Representational Similarity Measures

**Max Klabunde**[1*]  **Tassilo Wald**[2,3,4*]  **Tobias Schumacher**[5,6*†]
**Klaus Maier-Hein**[2,3,4,7,8]  **Markus Strohmaier**[5,9,10]  **Florian Lemmerich**[1]

[1]University of Passau    [2]Medical Image Computing, German Cancer Research Center (DKFZ)
[3]Helmholtz Imaging, DKFZ    [4]University of Heidelberg    [5]University of Mannheim
[6]RWTH Aachen University    [7]Heidelberg University Hospital
[8]National Center for Tumor Diseases (NCT) Heidelberg
[9]GESIS - Leibniz Institute for the Social Sciences    [10]Complexity Science Hub

## Abstract

Measuring the similarity of different representations of neural architectures is a fundamental task and an open research challenge for the machine learning community. This paper presents the first comprehensive benchmark for evaluating representational similarity measures based on well-defined groundings of similarity. The representational similarity (ReSi) benchmark consists of (i) six carefully designed tests for similarity measures, (ii) 24 similarity measures, (iii) 14 neural network architectures, and (iv) seven datasets, spanning the graph, language, and vision domains. The benchmark opens up several important avenues of research on representational similarity that enable novel explorations and applications of neural architectures. We demonstrate the utility of the ReSi benchmark by conducting experiments on various neural network architectures, real-world datasets, and similarity measures. All components of the benchmark are publicly available[1] and thereby facilitate systematic reproduction and production of research results. The benchmark is extensible; future research can build on it and expand on it. We believe that the ReSi benchmark can serve as a sound platform catalyzing future research that aims to systematically evaluate existing and explore novel ways of comparing representations of neural architectures.

## 1 Introduction

Representations are fundamental concepts of deep learning that have garnered significant interest due to their ability to shed light on the opaque inner workings of neural networks. Studying and analyzing them has enabled insight into numerous problems, for example understanding learning dynamics (Morcos et al., 2018; Mehrer et al., 2018), catastrophic forgetting (Ramasesh et al., 2021), and language changes over time (Hamilton et al., 2016a). Such analyses commonly involve measuring similarity of representations, which resulted in a plethora of similarity measures proposed in the literature (Klabunde et al., 2023; Sucholutsky et al., 2023). However, these similarity measures have often been proposed in an ad hoc manner, without a comprehensive comparison to existing similarity measures. Moreover, they have often been proposed in conjunction with new quality criteria that were deemed desirable, with previously defined quality criteria being ignored. So far, only the few most popular measures have been compared (Ding et al., 2021; Hayne et al., 2024) or analyzed in more detail (Dujmović et al., 2023; Cui et al., 2022; Davari et al., 2022).

In this work, we present the first comprehensive benchmark for representational similarity measures. It comprises six tests that postulate different ground truth assumptions about the similarities between representations that measures could capture. We implemented these tests across several architectures and datasets in the graph, language, and vision domains. The ReSi benchmark enables tests for 24 similarity measures that have been proposed in the literature, and we illustrate how the results can

---

*Equal contribution. Author order among the co-first authors may be adjusted for individual use.
†Corresponding author. tobias.schumacher@uni-mannheim.de
[1]https://github.com/mklabunde/resi

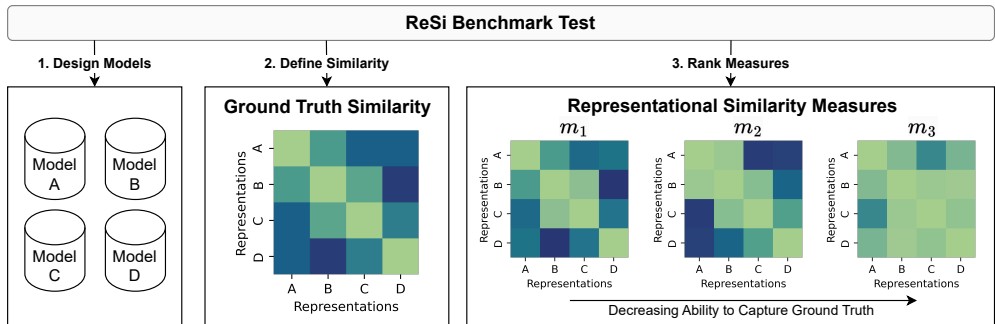

Figure 1: *Grounding similarity.* In all tests within the `ReSi` benchmark, we design a set of models for which we can establish a ground-truth about the similarity of their representations. The left heatmap illustrates the true similarity between a set of models, the other heatmaps the similarity values that different similarity measures assign to each model pair via their representations. We rank similarity measures by their ability to capture the ground truth. In practice, a ground-truth similarity between models is usually hard to attain. For the `ReSi` benchmark, we design tests where similarity is practically grounded.

provide insights into the properties and strengths and weaknesses of these measures. That way, `ReSi` can be useful as a test environment for new measures, a reference that guides the choice of measures in an application at hand, and a tool to obtain a deeper understanding of the differences between representations that are relevant to neural network behavior. All benchmark code and the corresponding models are openly accessible online.

## 2 GROUNDING REPRESENTATIONAL SIMILARITY

Before presenting the `ReSi` benchmark, we briefly introduce key terms and notations for representational similarity, and discuss how ground-truths for representational similarity can be established.

### 2.1 REPRESENTATIONAL SIMILARITY

The `ReSi` benchmark is designed to evaluate the quality of measures that aim to quantify similarity of neural representations. Such representations can be derived by applying a neural network model

$$f = f^{(L)} \circ f^{(L-1)} \circ \cdots \circ f^{(1)}, \tag{1}$$

where each function $f^{(l)} : \mathbb{R}^{D'} \longrightarrow \mathbb{R}^{D}$ denotes a single layer, on a set of $N$ inputs $\{\boldsymbol{X}_i\}_{i=1}^{N}$—for simplicity, we assume these inputs to be vectors in $\mathbb{R}^p$ even though, as in the vision domain, these can also be multidimensional matrices. By stacking these inputs to an input matrix $\boldsymbol{X} \in \mathbb{R}^{N \times p}$, one can then slightly abuse notation and, at any layer $l$, extract the model's *representation*

$$\boldsymbol{R} := \boldsymbol{R}^{(l)} = \left(f^{(l)} \circ f^{(l-1)} \circ \cdots \circ f^{(1)}\right)(\boldsymbol{X}) \in \mathbb{R}^{N \times D}, \tag{2}$$

where the rows $\boldsymbol{R}_i = f(\boldsymbol{X_i}) \in \boldsymbol{R}^D$ denote *instance representations*. *Representational similarity measures* compare full representation matrices $\boldsymbol{R}, \boldsymbol{R}'$, and can thus be defined as mappings

$$m : \mathbb{R}^{N \times D} \times \mathbb{R}^{N \times D'} \longrightarrow \mathbb{R} \tag{3}$$

that assign a scalar similarity score $m(\boldsymbol{R}, \boldsymbol{R}')$ to a pair of representations $\boldsymbol{R}, \boldsymbol{R}'$. For brevity, throughout this work we will often denote these measures as *similarity measures*. Unless noted otherwise, we always consider representations from the final hidden layer of a neural network.

### 2.2 GROUNDING SIMILARITY

Approaches to measuring similarity of representations vary widely. For example, similarity can be related to comparing pairwise distances in a representation (Kornblith et al., 2019), the ability

**Grounding by Prediction**  **Grounding by Design**

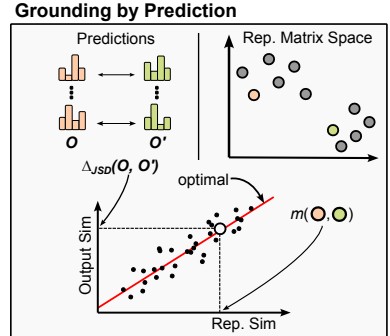 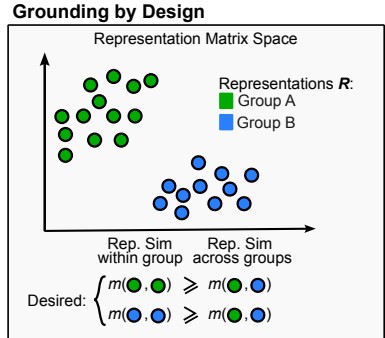

Figure 2: *Illustration of grounding approaches.* We consider two approaches to establish ground-truths for representational similarity. When *grounding by prediction*, we evaluate whether differences in representation matrices correspond to differences in predictions of models, as, for instance, measured by Jensen-Shannon divergence (JSD). Ideally, a representational similarity measure $m$ perfectly correlates with output similarity. When *grounding by design*, we design groups of models that are similar within and dissimilar across groups. A representational similarity measure $m$ should distinguish these groups accordingly.

to align two representations (Williams et al., 2021; Li et al., 2015), or their topology (Barannikov et al., 2022)—for a broader overview of approaches, see also the survey by Klabunde et al. (2023). While these approaches are usually justified by theoretical or practical desiderata, specific practical differences in models are not necessarily captured by these approaches—model difference is multifaceted. One can distinguish models by their behavior like accuracy, robustness to augmentations or domain shifts, by their preference regarding texture or shapes, or by aspects like differences in training data or their human-likeness, to name a few examples. This multidimensionality leads to a crucial problem in measuring representational similarity, namely the lack of a general ground truth that measures should reflect.

Due to this plurality in model behavior, a number of possible targets to ground representational similarity have been proposed in the literature. For our purposes, we focus on the following two broad approaches to establish a ground truth for similarity, which we also illustrate in Figure 2.
**1. Grounding by Prediction.** A straightforward way to obtain a ground truth for model similarity is to consider the differences in the predictive behaviors of a pair of models: when two models yield different predictions, they should also differ in their representations. This approach allows one to ignore where the source of difference between the models originates, but simultaneously implies that one cannot be sure whether ground-truth similarities stem from differences in the classifier or the representations. Previous efforts that have used this way to ground similarity include the study by Ding et al. (2021), who correlated representational similarity with accuracy difference, or the work by Barannikov et al. (2022), who grounded similarity to differences in individual predictions.
**2. Grounding by Design.** Through careful design, one can construct groups of representations for which one can impose a ground truth about similarity by relation. For instance, one can demand that representations from the same group be more similar to each other than representations from different groups. An example of such a design is shown by Kornblith et al. (2019), who trained multiple models of the same architecture and formed groups of representations based on the depth of their producing layer. Then, they demanded that representations of the same layer should be more similar to each other than to representations from other layers. These relative comparisons have the advantage that no deeper insights about the often opaque and nonlinear similarity measures are needed. However, the validity of the evaluation hinges on the validity of the assumption that the representations actually follow the expected groups, which requires clear justification.

# 3 ReSi: A Benchmark for Representational Similarity Measures

We present the main components of the `ReSi` benchmark. It consists of (i) six carefully designed tests for similarity measures, (ii) 24 similarity measures, (iii) 14 neural network architectures, and

Table 1: *Overview of the tests included in the* `ReSi` *benchmark.* We design six tests that define different ground truths for representational similarity.

| Test | Intuition | Grounding Type |
|---|---|---|
| Correlation to Accuracy Difference | Can variation in performance of classifiers trained on representations be captured? | Prediction |
| Correlation to Output Difference | Can variation in individual predictions of classifiers trained on representations be captured? | Prediction |
| Label Randomization | Can models that were trained on different labels be distinguished? | Design |
| Shortcut Affinity | Can models that rely on different features be distinguished? | Design |
| Augmentation | Can models with different levels of training augmentation be distinguished? | Design |
| Layer Monotonicity | Does layer similarity decrease with increased layer distance within a model? | Design |

(iv) seven datasets, spanning the graph, language, and vision domains. In addition, we discuss how the results for our benchmark are evaluated.

## 3.1 TESTS

We provide descriptions of the six tests implemented within the `ReSi` benchmark. We begin with discussing two tests that are grounded by predictions, and then describe four tests that are grounded by design. An overview of these tests is given in Table 1. In all tests, models were trained for classification tasks. Additional details of the test configurations can be found in Appendix B.

**Test 1: Correlation to Accuracy Difference.** When two models are trained under similar conditions but differ in their performance, this can be seen as a signal that the underlying representations are different. Following this intuition, we correlate the representational similarity of a pair of models with the absolute difference in their accuracies, thus similarity is *grounded by predictions*.
**Training Protocol.** We roughly follow the protocol established by Ding et al. (2021), for which ten models are trained on each dataset, varying only by the training seeds. Afterward, we compute the accuracy of the models on the test set.

**Test 2: Correlation to Output Difference.** This test follows the same intuition as Test 1, with the difference that it focuses on differences in individual outputs of each model rather than their aggregated accuracy scores. Given that models with similar accuracy may still yield different predictions on individual instances, this provides a more fine-grained signal which can serve as a more robust grounding for representational similarity. Thus, similarity is again *grounded in predictions.*
**Training Protocol.** Due to the similarities to Test 1, we use the same models for this experiment.

**Test 3: Label Randomization.** In this experiment, we train models on the same input data, but with labels randomized to different degrees. The expectation is that models that learn to predict the true labels learn different representations compared to the models that learn to memorize random labels. These differently trained models are grouped by their degree of label randomization during training. We evaluate whether similarity within a group is greater than between groups. Therefore, representational similarity is *grounded by design* of these groups.
**Training Protocol.** For each domain, we create at least two groups of models. We always train one group on fully correct labels and one group on fully random labels. Additional groups have partially randomized labels (25%, 50%, 75%). Across all domains, we always trained five models per group.

**Test 4: Shortcut Affinity.** The ability to identify whether two models use similar or different features can be a desirable property for a similarity measure. Hence, we create a scenario, in which we control feature usage by introducing artificial shortcut features to the training data. Specific features correspond to each label, thus leaking them, but not necessarily perfectly—we can add incorrect shortcut features for some instances. Groups of models are then formed based on the degree to which the shortcut features in the training data match the true labels. Similarly to Test 3, we assess whether similarities within the same groups are greater than between groups. That way, representational similarity is *grounded by design* of these groups.
**Training Protocol.** For each domain, we construct one group of models trained on data with shortcuts that always leak the correct label, one group trained on data with completely random shortcut features, and additional groups in which the labels were leaked on a fixed percentage of training samples. Representations are then extracted from a fixed test set, on which random shortcuts have been introduced. Each group consists of five models, in which varying training seeds affected model training and shortcut features.

Table 2: *Similarity measures included in the* ReSi *benchmark.* For each measure, we present their type of measure as specified in the survey by Klabunde et al. (2023), their abbreviations as used in our plots and tables, a reference, and the preprocessing they require. RSM abbreviates *Representational Similarity Matrix*, i.e., a matrix of pairwise similarities between the instances. CCA denotes *Canonical Correlation Analysis.*

| Measure Type | Abbreviation | Measure | Reference | Preprocessing |
|---|---|---|---|---|
| CCA | PWCCA | Projection-Weighted CCA | Morcos et al. (2018) | |
| | SVCCA | Singular Value CCA | Raghu et al. (2017) | |
| Alignment | AlignCos | Aligned Cosine Similarity | Hamilton et al. (2016b) | |
| | AngShape | Orthogonal Angular Shape Metric | Williams et al. (2021) | Centered columns and unit matrix norm |
| | HardCorr | Hard Correlation Match | Li et al. (2015) | |
| | LinReg | Linear Regression | Kornblith et al. (2019) | Centered columns |
| | OrthProc | Orthogonal Procrustes | Ding et al. (2021) | Centered columns and unit matrix norm |
| | PermProc | Permutation Procrustes | Williams et al. (2021) | |
| | ProcDist | Procrustes Size-and-Shape-Distance | Williams et al. (2021) | Centered columns |
| | SoftCorr | Soft Correlation Match | Li et al. (2015) | |
| RSM | CKA | Centered Kernel Alignment | Kornblith et al. (2019) | Centered columns |
| | DistCorr | Distance Correlation | Székely et al. (2007) | |
| | EOS | Eigenspace Overlap Score | May et al. (2019) | |
| | GULP | GULP | Boix-Adserà et al. (2022) | Centered rows and row norm to $\sqrt{N}$ |
| | RSA | Representational Similarity Analysis | Kriegeskorte et al. (2008) | |
| | RSMDiff | RSM Norm Difference | Yin & Shen (2018) | |
| Neighbors | 2nd-Cos | Second Order Cosine Similarity | Hamilton et al. (2016a) | |
| | Jaccard | Jaccard Similarity | Wang et al. (2022) | |
| | RankSim | Rank Similarity | Wang et al. (2022) | |
| Topology | IMD | IMD Score | Tsitsulin et al. (2020) | |
| | RTD | Representation Topology Divergence | Barannikov et al. (2022) | |
| Statistic | ConcDiff | Concentricity Difference | Wang et al. (2022) | |
| | MagDiff | Magnitude Difference | Wang et al. (2022) | |
| | UnifDiff | Uniformity Difference | Wang & Isola (2020) | |

**Test 5: Augmentation.** Augmentation is commonly used to "teach" models to become invariant to changes in the input domain which do not affect the labels. In this test, we evaluate whether the similarity measures are able to capture robustness to such changes, by developing models with varying amounts of augmentation in their training data. Similar to before, we train groups of models with different degrees of augmentation. Models of the same group should yield more similar representations than models from other groups trained on differently augmented data—again, similarity is *grounded by design* of the models.

**Training Protocol.** In all domains, we trained one group of reference models on standard data, and additional groups of models with varying degree of augmentation on the training data. Each group consists of five models, which only vary in their training seed. Representations are computed on the standard, non-augmented test data.

**Test 6: Layer Monotonicity.** As noted in Section 2.1, the individual layers $f^{(l)}$ of a neural network $f$ all yield representations $\boldsymbol{R}^{(l)}$ that can be compared. Given that these layers also represent a sequence of transformations of the input data $\boldsymbol{X}$, i.e., $\boldsymbol{R}^{(l)} = f^{(l)}(\boldsymbol{R}^{(l-1)})$, it seems intuitive that representations of neighboring layers should be more similar than representations of layers that are further away, given the greater number of transformations between the further-apart layers[2]. Thus, in this experiment, we extract layer-wise representations of a neural network and test whether a representational similarity measure can distinguish pairs of layers based on their distances from each other. Similarity of representations is, therefore, *grounded by design*.

**Training Protocol.** We reuse the trained models of Test 1 and Test 2, as no changes were made in the training scenario – only for graph neural networks, we increased the number of layers to five inner layers to enable a sufficient number of comparisons.

## 3.2 Representational Similarity Measures

ReSi evaluates 24 different similarity measures, for which we provide a brief overview in Table 2. We used the reference implementations of the measures where possible. Otherwise, we closely followed the given definitions and recommendations, also in the preprocessing of representations. Explicit definitions of all measures and details on hyperparameter choices can be found in Appendix A.

---

[2]This could be violated if there were skip connections, but we have controlled for this (see Appendix B.1)

## 3.3 MODELS

Overall, the `ReSi` benchmark utilizes a range of various graph, language, and vision models. Details on parameter choices can be found in Appendix B. All trained models are publicly available.

**Graphs.** As graph neural network architectures, we chose the classic *GCN* (Kipf & Welling, 2017), *GraphSAGE* (Hamilton et al., 2017), and *GAT* (Veličković et al., 2018) models, using their respective implementation from `PyTorch Geometric` (Fey & Lenssen, 2019). In addition, we considered the position-sensitive *P-GNN* model (You et al., 2019), using the implementation of the authors. For each experiment and dataset, we trained these models from scratch.

**Language.** We chose the popular *BERT* architecture (Devlin et al., 2019) and its *ALBERT* variant (Lan et al., 2020) as well as *SmolLM2-1.7B* (Allal et al., 2025) as an LLM. For BERT, we use the 25 models pre-trained with different seeds from Sellam et al. (2022). For ALBERT and SmolLM2, we use a single pretrained model (see Appendix B.2 for details). In our experiments, we fine-tuned these models on the given datasets to avoid computationally expensive pre-training. We use the representation of the token that is passed into the final classifier, i.e., CLS for BERT and ALBERT, final token for SmolLM2, as well as mean-pooled representations over all tokens (also see Appendix B.1).

**Vision.** We focus on prominent classification architecture families, namely *ResNets* (He et al., 2016), *ViTs* (Dosovitskiy et al., 2021) and older *VGG's* (Simonyan & Zisserman, 2015). To capture the effect of scaling architecture sizes, we included ResNet18, ResNet34, ResNet101, VGG11, VGG19, ViT-B/32 and the ViT-L/32 architectures. All models were trained from scratch, apart from the ViTs which were initialized with pre-trained weights from ImageNet21k (Deng et al., 2009).

## 3.4 DATASETS

We provide a brief overview of the datasets used within the `ReSi` benchmark. More detailed descriptions of the datasets can be found in Appendix B.2.1.

**Graphs.** We focus on graph datasets that provide multiclass labels for node classification, and for which dataset splits into training, validation and test sets are already available. Specifically, we select *Cora* (Yang et al., 2016), *Flickr* (Zeng et al., 2020), and *OGBN-Arxiv* (Hu et al., 2020). For the Cora graph, we extract representations from the complete test set of 1,000 instances, whereas for Flickr and OGBN-Arxiv, we subsampled the test set to 10,000 instances for computational reasons.

**Language.** We use two classification datasets: SST2 (Socher et al., 2013) is a collection of sentences extracted from movie reviews, labeled with their sentiment. MNLI (Williams et al., 2018) is a dataset of premise-hypothesis pairs labeled with the true relation of these pairs. We used the validation and validation-matched subsets to extract representations for SST2 and MNLI, respectively.

**Vision.** We use ImageNet100 (IN100), a random subsample of 100 classes of ImageNet1k (Russakovsky et al., 2015) and CIFAR-100 (Krizhevsky & Hinton, 2009). IN100 reduces training time while keeping image resolution and content similar to ImageNet1k. The image resolution was fixed to 224x224 on IN100 and 32x32 on CIFAR-100 except for ViT models, which used 224x224.

## 3.5 EVALUATION

Lastly, we describe how we evaluate and quantify the performance of representational similarity measures within the `ReSi` benchmark. Due to the different nature of the two approaches we use to ground representational similarity, we present the corresponding evaluation approaches separately.

**Grounding by Prediction.** When grounding similarity of representations with predictions of their corresponding classification models $f$, we specifically consider the outputs $\boldsymbol{O} := f(\boldsymbol{X}) \in \mathbb{R}^{N \times C}$, where we assume that each row $\boldsymbol{O}_i = f(\boldsymbol{X}_i) \in \mathbb{R}^C$ consists of the instance-wise class probability scores for $C$ given classes.

In Test 1, we leverage these outputs to correlate representational similarity with absolute differences in *accuracy*. Thus, letting $q_{\text{acc}}(\boldsymbol{O}) := q_{\text{acc}}(\boldsymbol{O}, \boldsymbol{y})$ denote the accuracy of an output $\boldsymbol{O}$ with respect to ground-truth labels $\boldsymbol{y} \in \mathbb{R}^N$, we compute the absolute difference in accuracies

$$\Delta_{\text{acc}}(\boldsymbol{O}, \boldsymbol{O'}) = |q_{\text{acc}}(\boldsymbol{O}) - q_{\text{acc}}(\boldsymbol{O'})|. \tag{4}$$

Then, given a similarity measure $m$, and letting $\mathcal{F}$ denote the set of models trained for this test, for all model pairs $f, f' \in \mathcal{F}$ we collect the representational similarity scores $m(\boldsymbol{R}, \boldsymbol{R'})$ as well as the accuracy differences $\Delta_{\text{acc}}(\boldsymbol{O}, \boldsymbol{O'})$, and report the *Spearman correlation* between these sets of values, together with the levels of statistical significance.

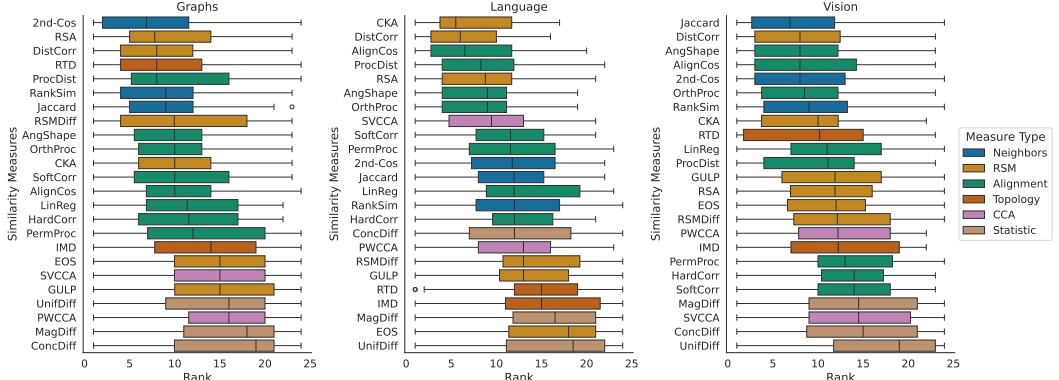

Figure 3: *Aggregated ranks of measures across all models and tests, separated by domain.* Lower is better. Measures are ordered by their median rank, and categorized according to the taxonomy by Klabunde et al. (2023). Tied measures all receive the best rank of their group. Boxplots indicate quartiles of rank distributions, the whiskers extend up to 1.5 times the inter-quartile range. No single measure or category stands out across all domains.

In Test 2, we consider differences in instance-wise predictions $\boldsymbol{O}_i, \boldsymbol{O}_i'$ rather than differences in aggregate performance scores. Thus, for all pairs of models $f, f' \in \mathcal{F}$, we compute the disagreement

$$\Delta_{\text{Dis}}(\boldsymbol{O}, \boldsymbol{O}') = \frac{1}{N} \sum_{i=1}^{N} \mathbb{1}\{\arg\max_j \boldsymbol{O}_{i,j} \neq \arg\max_j \boldsymbol{O}_{i,j}'\}, \tag{5}$$

between their hard predictions, and the average *Jensen-Shannon divergence (JSD)*

$$\Delta_{\text{JSD}}(\boldsymbol{O}, \boldsymbol{O}') = \frac{1}{2N} \sum_{i=1}^{N} \text{JSD}(\boldsymbol{O}_i \| \boldsymbol{O}_i') \tag{6}$$

of the class-wise probability scores, and report the *Spearman correlation* of both measures with the corresponding set of representational similarities $m(\boldsymbol{R}, \boldsymbol{R}')$.

**Grounding by Design.** For all tests of this category, we do not consider functional outputs of models as a ground truth for representational similarity anymore. Instead, we have created multiple groups of representations $\mathcal{G}$, typically separated by differences in model training, through which we impose a ground truth about similarity by relation. In Test 3, 4 and 5, we postulate that for any similarity measure $m$ it should hold that representations $\boldsymbol{R}, \boldsymbol{R}' \in \mathcal{G}$ from the same group should be more similar to each other than representations $\boldsymbol{R} \in \mathcal{G}, \boldsymbol{R}^* \in \mathcal{G}^*$ from different groups, that is,

$$m(\boldsymbol{R}, \boldsymbol{R}^*) \leq m(\boldsymbol{R}, \boldsymbol{R}'), \tag{7}$$

where we assume that for $m$, higher values indicate more similarity. Then, a trivial performance measure is given by the *conformity rate*, i.e., the relative amount of times a similarity measure $m$ satisfies (7) across all combinations of groups and representations. Given that this measure can, however, be biased by the ratio of intergroup vs. intragroup pairs, we additionally adapt the *Area under the Precision-Recall curve (AUPRC)* measure to our context. This is done by assigning a label $y(\boldsymbol{R}, \boldsymbol{R}') := \mathbb{1}\{\mathcal{G} = \mathcal{G}'\}$ to each pair of representations $\boldsymbol{R} \in \mathcal{G}, \boldsymbol{R}' \in \mathcal{G}'$ that can be compared within the given set of groups, and then interpreting the representational similarities $m(\boldsymbol{R}, \boldsymbol{R}')$ as decision scores based on which one should be able to "predict" the label $y(\boldsymbol{R}, \boldsymbol{R}')$.

Finally, in Test 6, we group representations $\boldsymbol{R}^{(l)}$ by the layers $l$ they were extracted from. However, in contrast to the previous tests, we postulate that an ordinal relationship between the layers has to hold. Specifically, given a model with $L$ layers, for all tuples $1 \leq i \leq j < k \leq l \leq L$ we require a measure $m$ to satisfy

$$m(\boldsymbol{R}^{(i)}, \boldsymbol{R}^{(l)}) \leq m(\boldsymbol{R}^{(j)}, \boldsymbol{R}^{(k)}), \tag{8}$$

assuming that for $m$, higher values indicate more similarity. As in the previous tests, we report the corresponding *conformity rate*. Further, we report the *Spearman correlation* between the similarities $m(\boldsymbol{R}^{(i)}, \boldsymbol{R}^{(j)})$ and the distance $j - i$ of the corresponding layers over all tuples $1 \leq i < j \leq L$.

Table 3: *Exemplary results for selected datasets and models.* We show results of GraphSAGE on Flickr for graphs, BERT on SST2 (CLS token) for language, and ResNet18 on ImageNet100 for vision. For the first two tests, we report the Spearman correlation between representational similarity and the difference in accuracy and JSD, respectively. For Tests 3-5, we report the area under the precision-recall curve, quantifying if the corresponding groups of models can be separated by the similarity measures. For Layer Monotonicity (Test 6), we report the average Spearman correlation between representational similarity and layer distance. Higher values indicate that a similarity measure better reflects the ground-truths from our tests. Our benchmark highlights that similarity measures have different strengths and weaknesses, and that there exists no measure that performs well across all tests and all domains. For example, PWCCA (first row) separates layers of GraphSAGE and BERT well, but not the layers of ResNet18.

| Type | Test
Modality | Grounding by Prediction | | | | | | Grounding by Design | | | | | | | | | Layer Monotonicity | | |
| | | Corr. to Accuracy Difference | | | Corr. to JSD Difference | | | Label Randomization | | | Shortcut Affinity | | | Augmentation | | | | | |
| | | Graph | Lang. | Vision | Graph | Lang. | Vision | Graph | Lang. | Vision | Graph | Lang. | Vision | Graph | Lang. | Vision | Graph | Lang. | Vision |
|---|---|---|---|---|---|---|---|---|---|---|---|---|---|---|---|---|---|---|---|
| CCA | PWCCA | -0.05 | -0.33* | -0.02 | 0.38** | -0.32** | 0.13 | 0.44 | 0.27 | 0.81 | 0.43 | 0.32 | 0.99 | 0.57 | 0.35 | 0.90 | **1.00** | **1.00** | 0.11 |
| | SVCCA | 0.01 | -0.08 | 0.29* | 0.23 | 0.47** | 0.21 | 0.80 | 0.64 | **1.00** | 0.93 | 0.36 | 0.55 | 0.67 | 0.61 | 0.40 | 0.44 | 0.64 | 0.20 |
| Alignment | AlignCos | 0.24 | 0.02 | -0.08 | 0.44** | 0.49** | 0.08 | 0.42 | **0.99** | 0.45 | **1.00** | 0.54 | **1.00** | 0.70 | 0.45 | 0.71 | 0.68 | 0.99 | 0.52 |
| | AngShape | 0.28 | -0.14 | 0.21 | **0.63**** | 0.40** | 0.24 | 0.43 | 0.49 | 0.72 | **1.00** | 0.43 | **1.00** | 0.76 | 0.52 | 0.71 | 0.98 | 0.99 | 0.55 |
| | HardCorr | 0.35* | -0.33* | 0.21 | 0.50** | -0.01 | 0.28 | 0.46 | 0.44 | 0.72 | **1.00** | 0.36 | 0.97 | 0.72 | 0.34 | 0.46 | 0.80 | 0.99 | 0.01 |
| | LinReg | 0.17 | -0.38** | 0.19 | 0.48** | 0.02 | 0.21* | 0.45 | 0.40 | 0.91 | 0.61 | 0.46 | 0.99 | 0.81 | 0.40 | 0.94 | **1.00** | 0.90 | 0.55 |
| | OrthProc | 0.28 | -0.14 | 0.21 | **0.63**** | 0.40** | 0.24 | 0.43 | 0.49 | 0.72 | **1.00** | 0.43 | **1.00** | 0.76 | 0.52 | 0.71 | 0.98 | 0.99 | 0.55 |
| | PermProc | -0.19 | -0.09 | 0.07 | -0.10 | 0.04 | 0.18 | 0.90 | 0.45 | 0.70 | **1.00** | 0.55 | 0.72 | 0.69 | 0.31 | 0.41 | 0.68 | 0.75 | 0.20 |
| | ProcDist | -0.06 | 0.10 | 0.08 | -0.18 | 0.49** | 0.08 | 0.62 | 0.86 | 0.72 | **1.00** | 0.52 | **1.00** | 0.81 | 0.43 | 0.58 | **1.00** | 0.89 | 0.55 |
| | SoftCorr | 0.33* | -0.33* | 0.27 | 0.53** | -0.01 | **0.45**** | 0.45 | 0.48 | 0.72 | **1.00** | 0.34 | 0.97 | 0.58 | 0.41 | 0.45 | 0.89 | 0.95 | 0.11 |
| RSM | CKA | 0.27 | -0.06 | **0.36*** | 0.58** | 0.48** | 0.30* | 0.66 | 0.59 | **1.00** | **1.00** | 0.38 | **1.00** | 0.75 | 0.61 | 0.90 | 0.89 | 0.96 | 0.87 |
| | DistCorr | **0.42**** | -0.10 | 0.31* | 0.43** | 0.51** | 0.26 | 0.43 | 0.59 | **1.00** | **1.00** | 0.39 | **1.00** | 0.79 | **0.62** | 0.83 | 0.99 | 0.98 | **0.97** |
| | EOS | -0.27 | -0.38** | 0.05 | 0.38** | -0.21** | 0.09 | 0.42 | 0.36 | 0.84 | 0.43 | 0.33 | **1.00** | 0.53 | 0.30 | 0.93 | **1.00** | 0.92 | 0.88 |
| | GULP | -0.27 | -0.36* | 0.02 | 0.38** | -0.30** | 0.07 | 0.42 | 0.28 | 0.89 | 0.43 | 0.30 | **1.00** | 0.54 | 0.33 | 0.92 | **1.00** | 0.45 | 0.53 |
| | RSA | 0.32* | -0.23 | 0.06 | **0.63**** | 0.44** | 0.12 | 0.42 | 0.48 | 0.75 | **1.00** | 0.47 | **1.00** | 0.72 | 0.61 | **0.98** | 0.99 | 0.96 | **0.97** |
| | RSMDiff | -0.16 | 0.20 | 0.09 | -0.04 | 0.24** | -0.41** | **0.92** | 0.91 | **1.00** | 0.92 | 0.37 | 0.57 | 0.93 | 0.34 | 0.45 | 0.65 | 0.84 | -0.33 |
| Neighbors | 2nd-Cos | -0.19 | 0.30* | -0.08 | 0.15 | 0.49** | -0.13 | 0.42 | 0.37 | **1.00** | **1.00** | **0.64** | **1.00** | 0.92 | 0.40 | 0.78 | 0.96 | 0.97 | 0.55 |
| | Jaccard | 0.28 | 0.17 | -0.11 | 0.42** | 0.54** | 0.36* | 0.43 | 0.35 | **1.00** | 0.83 | **0.64** | **1.00** | 0.88 | 0.39 | 0.97 | 0.97 | 0.96 | 0.55 |
| | RankSim | 0.31* | 0.14 | 0.07 | 0.30* | **0.56**** | -0.15 | 0.43 | 0.34 | **1.00** | 0.77 | **0.64** | 0.99 | 0.88 | 0.36 | 0.71 | 0.97 | 0.96 | 0.55 |
| Topology | IMD | 0.37* | -0.06 | 0.17 | 0.29 | 0.02 | -0.11 | 0.37 | 0.47 | **1.00** | 0.97 | 0.34 | 0.67 | 0.57 | 0.30 | 0.56 | 0.82 | 0.54 | -0.00 |
| | RTD | 0.13 | 0.05 | 0.09 | -0.14 | 0.33** | -0.18 | 0.59 | 0.27 | **1.00** | **1.00** | 0.39 | **1.00** | **1.00** | 0.38 | 0.57 | 0.98 | 0.39 | **0.97** |
| Statistic | ConcDiff | -0.29 | 0.12 | -0.11 | -0.03 | -0.02 | -0.29* | 0.57 | 0.96 | 0.81 | 0.18 | 0.27 | 0.53 | 0.35 | 0.32 | 0.43 | -0.27 | 0.96 | -0.78 |
| | MagDiff | -0.17 | -0.13 | -0.16 | 0.06 | 0.11 | -0.38* | 0.72 | 0.38 | **1.00** | 0.78 | 0.35 | 0.37 | 0.17 | 0.31 | 0.37 | 0.50 | 0.48 | -0.37 |
| | UnifDiff | 0.03 | **0.35*** | -0.18 | 0.02 | 0.38** | -0.34* | 0.90 | 0.60 | 0.21 | 0.50 | 0.37 | 0.75 | 0.24 | 0.33 | 0.17 | 0.89 | 0.77 | 0.18 |

Statistical significance for prediction-grounded tests evaluated with correlation is indicated by * (5%) and ** (1%).

# 4 BENCHMARK RESULTS

In the following, we illustrate selected benchmark results. Due to limited space, we focus on presenting (a) an aggregated overview and (b) an exemplary detailed result for a single dataset and a single architecture. We provide detailed results in Appendix C. We further present average runtimes of the similarity measures, which can vary by multiple magnitudes, in Appendix D.

For an aggregated result overview, we rank all measures for each combination of test, dataset, and architecture. The rank distribution across tests in each domain is shown in Figure 3. We observe that no measure outperforms the others consistently across the domains, with most measures having high variance in their rankings. However, some domain-specific trends can be identified, such as neighborhood-based second-order cosine similarity (2nd-Cos), Jaccard similarity, or rank similarity measures that perform well in the graph domain. In the language domain, the popular linear CKA measure performs best. Further, in the vision domain, various types of measures perform well—particularly the less-used Jaccard similarity. Orthogonal Procrustes, which has overall performed well in the analysis by Ding et al. (2021), consistently ranks among the top 50% of measures. By contrast, other prominent measures, such as RSA or SVCCA, do not appear to stand out in general.

The `ReSi` benchmark can also provide more detailed results for analysis. To provide some examples, Table 3 presents the outcomes for a single dataset and architecture in each domain. Higher values always indicate better adherence of measures to similarity groundings. Again, we observe that, even for a single dataset, there is no single measure that outperforms the others across all tests. Instead, most of the measures appear to perform well in some tests, but not in others. For instance, measures such as the eigenspace overlap score or GULP appear to be able to perfectly distinguish layers of a GraphSAGE model but appear incapable of identifying whether they have learned specific shortcuts, which most other measures appear to be capable of. One can also identify model-specific trends, such as neighborhood-based measures correlating particularly well with predictions from BERT models. In addition, considering the performance of descriptive statistics-based measures may provide additional geometric insights on representations: for instance, the concentricity of BERT representations appears to inform relatively well about the degree to which it was trained to memorize data.

## 5 TAKEAWAYS

**No Free Lunch.** Throughout our benchmark, we observe that ranks of individual measures vary significantly. For example, CKA (Kornblith et al., 2019) ranks first in Language, eighth in Vision, and 11th in Graphs when aggregating across all tests. This inconsistency extends beyond CKA, highlighting limitations in the general applicability of established measures. This implies that measures should be chosen carefully for a task at hand and that specific tasks may require the development of specialized similarity measures to appropriately capture relevant aspects of model behavior.

**Preprocessing Matters.** The similarity measures analyzed in this study include Orthogonal Procrustes (OrthProc) and Procrustes Size-and-Shape Distance (ProcDist), which differ only in the way the representations are preprocessed. Our results indicate that this seemingly small difference has a substantial impact on what properties of models similarity measures capture. This highlights avenues for future work: preprocessing implicitly assumes that representations are equivalent under the preprocessing transformation. If, in the case of OrthProc, results are consistently worse after normalizing representations to unit norm, this could be an indicator that the absolute magnitude of each axis carries semantic meaning that similarity measures need to pick up on. Therefore, a better understanding of what kinds of representation are truly equivalent is crucial, and the tests from `ReSi` could be used to empirically investigate the impact of different preprocessing techniques.

**Need of Best Practices.** Currently, similarity measures are often selected without deeper justification. Our results indicate that this practice leads to similarity scores that are difficult to interpret, as it is unclear whether the score indicates similarity on an interpretable axis and whether the score is specific to a measure. We argue that the community needs to develop a holistic set of best practices that enables robust and reliable analyses with similarity measures, which was also recently advocated for by Soni et al. (2024) in a neuroscience context. This further highlights potential for cross-domain pollination. With the following recommendations, we make a step towards such guidelines.

**General Recommendations.** Based on our results, from Figure 3, we can derive some general recommendations for applying similarity measures on specific domains. For graphs, neighborhood-based measures appear favorable, for language models, one should likely prefer CKA or distance correlation, and for vision models, Jaccard similarity appears to be the overall best choice. In general, the groups of alignment-, RSM- and neighborhood-based measures also appear to overall perform better than CCA-, topology- and descriptive statistics-based measures. However, this does not imply that measures from these families cannot be useful in specific applications. If one considers grounding to a specific test as particularly important, the tables in Appendix C inform on the ability of similarity measures in that test. If, for instance, one considers correlation with output difference the best grounding, Jaccard similarity would be preferable for the vision domain, distance correlation for the language domain, and RSA for the graph domain.

**Unexpected Efficacy of Overlooked Measures.** Within our benchmark, we implement a broad set of 24 measures, of which several measures have hardly been considered before in related literature and which sometimes performed surprisingly well. For example, Jaccard similarity emerged as the best-performing measure in the vision domain. This finding challenges popular opinion and suggests that rarely used measures, despite their limited prior application in these contexts, may possess unique properties that make them particularly well suited for certain types of data. This unexpected performance highlights the potential for revisiting and rigorously testing lesser known measures.

**Potential for Deeper Insights.** `ReSi` can also be used in different ways to obtain deeper insights about neural representations. One way comes from direct analysis of the given results. For instance, the finding that neighborhood-based measures perform well in the graph domain may indicate that neighborhoods within GNN representations are driven by training objectives. Similarly, the results of Test 5 indicate that augmentation of BERT models particularly affects uniformity of their representations. Although these findings can only be considered preliminary signals, they illustrate how `ReSi` can provide valuable pointers for future research. Additionally, the groundings provided by `ReSi`'s tests can be used to more systematically analyze the impact of preprocessing (as pointed out above), neural network design choices such as the type of normalization layer, or parameters of similarity measures such as similarity functions for RSMs, on representational similarity scores.

## 6 RELATED WORK

Despite the large amount of representational similarity measures that have been proposed in the literature (Klabunde et al., 2023), only few efforts have focused on a comparative analysis of existing

similarity measures. In the following, we briefly provide an overview of existing work.

**Grounding by Prediction.** Ding et al. (2021) first proposed grounding similarity of representations to performance differences of probes trained on them. They benchmarked three similarity measures, with tests focusing on identifying effects of seed variation or manually manipulating the representations. Compared to `ReSi`, they focus only on a single test in our framework. Their work was extended by Hayne et al. (2024) with a focus on measuring similarity of high-dimensional representations in CNNs. Similar to Test 2 in `ReSi`, Barannikov et al. (2022) tested the ability of their similarity measure to capture prediction disagreement.

**Grounding by Design.** Similar to Tests 3-5 in `ReSi`, some previous work has also tested whether similarity measures could distinguish representations based on differences in their underlying models. Boix-Adserà et al. (2022) considered representations of models from different architectures, and required similarity measures to distinguish representations by model family. Tang et al. (2020) tested whether measures could distinguish models based on differences in their training data. In a setting similar to Test 6, Kornblith et al. (2019) grouped representations of models that only differ in their training seed by layer depth, and tested whether measures assign more similarity to representations from corresponding layers. Finally, a few analyses have tested whether similarity measures can detect model-independent changes in representations. For instance, it has been explored whether measures distinguished synthetic representations with different cluster structure (Barannikov et al., 2022), levels of noise (Morcos et al., 2018), or subsampled dimensions (Shahbazi et al., 2021).

## 7 DISCUSSION AND CONCLUSIONS

We briefly summarize our contributions and discuss limitations as well as potential avenues for future research to be built on `ReSi`.

**Contributions.** In this paper, we have presented `ReSi`, the first comprehensive benchmark for representational similarity measures. By applying its six tests on a wide range of 24 similarity measures to evaluate their capabilities on graph, language, and vision representations, we have demonstrated that it (i) can serve as a test environment for new measures, (ii) provide guidance regarding which measures to choose for specific domains, architectures or application scenarios, and (iii) can be leveraged to obtain deeper understanding of representational similarity. We provide all code publicly online, and invite the machine learning community to use it to test (potentially new) measures, or to extend it to include more models or even additional tests.

**Limitations.** We have put considerable effort into properly grounding the benchmark tests. Specifically, when *grounding by design*, we have not only carefully set up distinct training conditions, but also verified from the validation performance of the models under consideration that these indeed follow the intended behavior and adapted the generation process if necessary. Still, we cannot fully control what models learn. Two behaviorally identical models could still differ in their internal process and, thereby, their representations, potentially confounding the resulting test scores. Nevertheless, our tests would inform whether the grounding property can be inferred from representations without supervision. When *grounding by predictions*, we have deliberately restricted the tests to models that only vary in their training seed, to reduce the potential for additional confounders as much as possible. Furthermore, in our tests, we solely focus on last-layer representations. The validity of the existing tests at other layers could be verified by evaluating whether the probes follow the same behavioral patterns as the full model.

**Future Work.** While `ReSi` implements 24 similarity measures, not all existing measures are included. Additionally, we did not explore the effects of different preprocessing approaches or parameter choices of individual measures. Given that such factors can lead to different results (Boix-Adserà et al., 2022; Timkey & van Schijndel, 2021), `ReSi` opens the way for a systematic evaluation of such steps in future work. Similarly, the six tests implemented in `ReSi` should be considered a comprehensive but extensible foundation of tests that users can build upon. We see a great potential in augmenting `ReSi` with additional tests, with the aim of a unified test suite that can inform the whole community of researchers and practitioners. For example, tests could be grounded via insights from interpretability, e.g., how similar reliance on specific input features is, as found via explanations (Eberle et al., 2022)—assuming the explanation is faithful—or how similar internal features are, as discovered via sparse autoencoders (Lan et al., 2024). Finally, we acknowledge that `ReSi` currently evaluates similarity measures within single-modality settings. However, these measures are also crucial for measuring the alignment of representations in multimodal contexts. Extending `ReSi` to encompass multimodal settings represents highly promising future work.

ACKNOWLEDGEMENTS

This work is supported by the Deutsche Forschungsgemeinschaft (DFG, German Research Foundation) under Grant No.: 453349072. The authors acknowledge support by the state of Baden-Württemberg through bwHPC and DFG through grant INST 35/1597-1 FUGG. This work was partly funded by Helmholtz Imaging (HI), a platform of the Helmholtz Incubator on Information and Data Science.

REPRODUCIBILITY STATEMENT

All our code and data as well as instructions how to run the benchmark are publicly available at `https://github.com/mklabunde/resi`.

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

# A  REPRESENTATIONAL SIMILARITY MEASURES

In this section, we first provide explicit descriptions of all similarity measures that we included in `ReSi`, and then give details on the hyperparameter choices.

## A.1  DEFINITIONS OF SIMILARITY MEASURES

In the following, we provide brief descriptions of all the representational similarity measures that we consider in this study, where we follow the categorization proposed in the recent survey by Klabunde et al. (2023). For more detailed descriptions and a broader overview of existing measures, we also point to this survey.

**CCA-based Measures.**  *Canonical Correlation Analysis (CCA)* Hotelling (1936) is based on the problem of finding weights $\boldsymbol{w_R} \in \mathbb{R}^D, \boldsymbol{w_{R'}} \in \mathbb{R}^{D'}$ for the columns in the representations, such that the linear combinations $\boldsymbol{Rw_R}$ and $\boldsymbol{R'w_{R'}} \in \mathbb{R}^N$ have maximal correlation. Assuming mean-centered representations, one can determine a set of canonical correlations $\rho_i$ that satisfy

$$\rho_i := \max_{\boldsymbol{w_R^{(i)}}, \boldsymbol{w_{R'}^{(i)}}} \frac{\langle \boldsymbol{Rw_R^{(i)}}, \boldsymbol{R'w_{R'}^{(i)}} \rangle}{\|\boldsymbol{Rw_R^{(i)}}\| \cdot \|\boldsymbol{R'w_{R'}^{(i)}}\|} \tag{9}$$
$$\text{s.t. } \boldsymbol{Rw_R^{(j)}} \perp \boldsymbol{Rw_R^{(i)}}, \ \ \boldsymbol{R'w_{R'}^{(j)}} \perp \boldsymbol{R'w_{R'}^{(i)}} \ \ \forall j < i.$$

A single similarity score $m(\boldsymbol{R}, \boldsymbol{R'})$ can then be obtained by aggregating the individual canonical correlations $\rho_i$, e.g., via taking their mean:

$$m_{\text{CCA}}(\boldsymbol{R}, \boldsymbol{R'}) = \frac{1}{D} \sum_{i=1}^{D} \rho_i. \tag{10}$$

In *Singular Value CCA (SVCCA)* (Raghu et al., 2017), this exact aggregation is used, but representations are first denoised by applying PCA on the mean-centered representations, removing those principal components, which explain less than a fixed percentage (usually 1%) of the variance.

For their *Projection-weighted CCA* (PWCCA) measure, Morcos et al. (2018) considered a weighted average of the canonical correlations, where the weighting coefficients $\alpha_i = \sum_{j=1}^{D} |\langle \boldsymbol{Rw_R^{(i)}}, \boldsymbol{R_{-,j}} \rangle|$ model the importance of each canonical correlation $\rho_i$.

**Alignment-based Measures.**  Several measures from this category are based on solving the orthogonal Procrustes problem, which intuitively is based on finding the best orthogonal mapping of to representations onto each other. Specifically, the *Orthogonal Procrustes (OrthProc)* measure is defined via

$$m_{\text{OrthProc}}(\boldsymbol{R}, \boldsymbol{R'}) = \min_{\boldsymbol{Q} \in \text{O}(D)} \|\boldsymbol{RQ} - \boldsymbol{R'}\|_F = (\|\boldsymbol{R}\|_F^2 + \|\boldsymbol{R'}\|_F^2 - 2\|\boldsymbol{R}^\mathsf{T}\boldsymbol{R'}\|_*)^{\frac{1}{2}}, \tag{11}$$

where it is assumed that columns of the input representations are centered, and scaled to unit norm. The *Procrustes Size-and-Shape-Distance* only differs in preprocessing, as it does not assume that the matrix is scaled to unit norm. For the *Permutation Procrustes* measure, the minimization in Equation (11) is restricted to permutation matrices. The *Orthogonal Angular Shape Metric* (Williams et al., 2021) follows a similar rationale, but optimizes minimizes a Frobenius norm:

$$m_{\text{AngShape}}(\boldsymbol{R}, \boldsymbol{R'}) = \min_{\boldsymbol{Q} \in \text{G}(D)} \arccos\langle \boldsymbol{RQ}, \boldsymbol{R'} \rangle_F. \tag{12}$$

The matrix $\boldsymbol{Q}_*$ that yields the solution of the Procrustes Problem in Equation (11) is also used for the *Aligned Cosine Similarity* measure. Specifically, it considers the cosine similarities between all instance representations after alignment, and uses their average as similarity score:

$$m_{\text{AlignCos}}(\boldsymbol{R}, \boldsymbol{R}') = \frac{1}{N} \sum_{i=1}^{N} \text{cos-sim}\left((\boldsymbol{R}\boldsymbol{Q}^*)_i, \boldsymbol{R}'_i\right). \tag{13}$$

In a slightly different approach, (Li et al., 2015) proposed to align representations by matching neurons based on their correlation. For the *Hard Correlation Match* measure, neurons are matched one-to-one in a greedy fashion. Letting $\boldsymbol{M}$ denote the matrix that matches the neurons, which is a permutation matrix in this strict matching, the measure then yields the average correlation between the matched neurons:

$$m_{\text{HardCorr}}(\boldsymbol{R}, \boldsymbol{R}') = \frac{1}{D} \sum_{j=1}^{D} \frac{\langle \boldsymbol{R}_{-,j}, (\boldsymbol{R}'\boldsymbol{M})_{-,j} \rangle}{\|\boldsymbol{R}_{-,j}\|_2 \|(\boldsymbol{R}'\boldsymbol{M})_{-,j}\|_2}. \tag{14}$$

For the *Soft Correlation Match*, this matching is relaxed, so that one neuron from $\boldsymbol{R}$ can be matched to multiple neurons from $\boldsymbol{R}'$.

Finally, the *Linear Regression* measure aligns representations by predicting one from the other via a linear transformation. The resulting $R^2$ score can then be used as similarity score:

$$m_{\text{LinReg}}(\boldsymbol{R}, \boldsymbol{R}') = 1 - \frac{\min_{\boldsymbol{W} \in \mathbb{R}^{D \times D}} \|\boldsymbol{R}' - \boldsymbol{R}\boldsymbol{W}\|_F^2}{\|\boldsymbol{R}'\|_F^2} = \frac{\left\|\left(\boldsymbol{R}'(\boldsymbol{R}'^{\top}\boldsymbol{R}')^{-1/2}\right)^{\top}\boldsymbol{R}\right\|_F^2}{\|\boldsymbol{R}\|_F^2}. \tag{15}$$

**RSM-based Measures.** To avoid issues in finding optimal alignments, several methods consider *representational similarity matrices (RSMs)*, which describe the similarity of each instance $\boldsymbol{R}_i$ to all other instances in a representation $\boldsymbol{R}$. Formally, the RSM $\boldsymbol{S} \in \mathbb{R}^{N \times N}$ of a representation $\boldsymbol{R}$ can be defined in terms of its elements via

$$\boldsymbol{S}_{i,j} := s(\boldsymbol{R}_i, \boldsymbol{R}_j). \tag{16}$$

where $s : \mathbb{R}^D \times \mathbb{R}^D \longrightarrow \mathbb{R}$ denotes a given instance-wise similarity function. Common choices for the similarity function $s$ include correlation Kriegeskorte et al. (2008) or kernel functions Kornblith et al. (2019). A direct way to obtain a representational similarity measure is then to consider the norm of the difference between two RSMs, as in the *RSM Norm Difference* Yin & Shen (2018):

$$m_{\text{RSMDiff}}(\boldsymbol{R}, \boldsymbol{R}') = \|\boldsymbol{S} - \boldsymbol{S}'\|. \tag{17}$$

More broadly, Kriegeskorte et al. (2008) proposed *Representational Similarity Analysis* (RSA), which considers an inner similarity function $s_{\text{in}}$ to compute RSMs, of which the lower triangles are then vectorized to a vector $\mathsf{v}(\boldsymbol{S}) \in \mathbb{R}^{N(N-1)/2}$ and compared via an outer similarity function $s_{\text{out}}$:

$$m_{\text{RSA}}(\boldsymbol{R}, \boldsymbol{R}') = s_{\text{out}}(\mathsf{v}(\boldsymbol{S}), \mathsf{v}(\boldsymbol{S}')). \tag{18}$$

One of the most commonly used similarity measures in related literature is *Centered Kernel Alignment (CKA)*, which has been proposed by Kornblith et al. (2019). CKA applies kernel functions as instance-wise similarity measures $s$ to compute the RSMs, which are then compared via the Hilbert-Schmidt Independence Criterion (HSIC) (Gretton et al., 2005). Specifically, the similarity score is computed via

$$m_{\text{CKA}}(\boldsymbol{R}, \boldsymbol{R}') = \frac{\text{HSIC}(\boldsymbol{S}, \boldsymbol{S}')}{\sqrt{\text{HSIC}(\boldsymbol{S}, \boldsymbol{S})\text{HSIC}(\boldsymbol{S}', \boldsymbol{S}')}}. \tag{19}$$

*Distance Correlation (DistCorr)* (Székely et al., 2007) is motivated from testing statistical dependence of two random variables, but can also be applied as representational similarity measure. Assuming that RSMs are mean-centered in both rows and columns, one can compute their squared sample distance covariance $\text{dCov}^2(\boldsymbol{S}, \boldsymbol{S}') = \frac{1}{N^2} \sum_{i=1}^{N} \sum_{j=1}^{N} \boldsymbol{S}_{i,j}\boldsymbol{S}'_{i,j}$, and then derive the distance correlation via

$$m_{\text{DistCorr}}(\boldsymbol{R}, \boldsymbol{R}') = \frac{\text{dCov}^2(\boldsymbol{S}, \boldsymbol{S}')}{\sqrt{\text{dCov}^2(\boldsymbol{S}, \boldsymbol{S}) \, \text{dCov}^2(\boldsymbol{S}', \boldsymbol{S}')}}. \tag{20}$$

*Eigenspace Overlap Score* (May et al., 2019) compares RSMs by comparing the spaces spanned from their eigenvectors. Letting $\boldsymbol{U} \in \mathbb{R}^{N \times D}, \boldsymbol{U}' \in \mathbb{R}^{N \times D'}$ denote the matrices of eigenvectors that correspond to the non-zero eigenvalues of $\boldsymbol{S}, \boldsymbol{S}'$, respectively, the measure is defined as

$$m_{\text{EOS}}(\boldsymbol{R}, \boldsymbol{R}') = \frac{1}{\max(D, D')} \|\boldsymbol{U}^{\top}\boldsymbol{U}'\|_F^2. \tag{21}$$

Finally, *GULP* (Boix-Adserà et al., 2022) aims to measure the extent of how differently linear (ridge) regression models that use either the representation $\boldsymbol{R}$ or the representation $\boldsymbol{R}'$ can generalize. Letting the RSMs $\boldsymbol{S} = \frac{1}{N}\boldsymbol{R}^{\mathsf{T}}\boldsymbol{R}$ denote the matrix of covariance within a representation, $\boldsymbol{S}_{\boldsymbol{R},\boldsymbol{R}'} = \frac{1}{N}\boldsymbol{R}^{\mathsf{T}}\boldsymbol{R}'$ the cross-covariance matrix, and $\boldsymbol{S}^{-\lambda} = (\boldsymbol{S} + \lambda\boldsymbol{I}_D)^{-1}$ the inverse of a regularized covariance matrix, they provide a closed-form definition of the GULP measure

$$m_{\mathrm{GULP}}^{\lambda}(\boldsymbol{R}, \boldsymbol{R}') = \Big( \operatorname{tr}(\boldsymbol{S}^{-\lambda}\boldsymbol{S}\boldsymbol{S}^{-\lambda}\boldsymbol{S}) + \operatorname{tr}(\boldsymbol{S}'^{-\lambda}\boldsymbol{S}'\boldsymbol{S}'^{-\lambda}\boldsymbol{S}') - 2\operatorname{tr}(\boldsymbol{S}^{-\lambda}\boldsymbol{S}_{\boldsymbol{R},\boldsymbol{R}'}\boldsymbol{S}'^{-\lambda}\boldsymbol{S}_{\boldsymbol{R},\boldsymbol{R}'}^{\mathsf{T}}) \Big)^{1/2}, \tag{22}$$

where the hyperparameter $\lambda \geq 0$ corresponds to the regularization weight of the ridge regression models.

**Neighborhood-based Measures.** The following set of measures is based on comparing the nearest neighbors of instances in the representation space. Each measure determines the sets of $k$ nearest neighbors $\mathcal{N}_{\boldsymbol{R}}^k(i)$ of each instance representation $\boldsymbol{R}_i$ from the full representation matrix $\boldsymbol{R}$ with respect to a given similarity function $s$, and then computes a vector of instance-wise neighborhood similarities $\big(v_{\mathrm{NN}}(\boldsymbol{R}, \boldsymbol{R}')_i\big)_{i \in \{1,\ldots,N\}}$, which are averaged over all instances to obtain similarity measures for the full representations $\boldsymbol{R}, \boldsymbol{R}'$:

$$m_{\mathrm{NN}}(\boldsymbol{R}, \boldsymbol{R}') = \frac{1}{N} \sum_{i=1}^{N} v_{\mathrm{NN}}(\boldsymbol{R}, \boldsymbol{R}')_i. \tag{23}$$

For the *k-NN Jaccard Similarity*, this vector simply contains the Jaccard similarities of the nearest neighbors of each pair of corresponding instance representations $\boldsymbol{R}_i$ and $\boldsymbol{R}_i'$:

$$\big(\boldsymbol{v}_{\mathrm{Jaccard}}^k(\boldsymbol{R}, \boldsymbol{R}')\big)_i := \frac{|\mathcal{N}_{\boldsymbol{R}}^k(i) \cap \mathcal{N}_{\boldsymbol{R}'}^k(i)|}{|\mathcal{N}_{\boldsymbol{R}}^k(i) \cup \mathcal{N}_{\boldsymbol{R}'}^k(i)|}. \tag{24}$$

*Second-Order Cosine Similarity* (Hamilton et al., 2016a) considers the union of the instance-wise nearest neighbors in terms of cosine similarity as an ordered set $\{j_1, \ldots, j_{K(i)}\} := \mathcal{N}_{\boldsymbol{R}}^k(i) \cup \mathcal{N}_{\boldsymbol{R}'}^k(i)$, and then compares these cosine similarities to the nearest neighbors via

$$\big(\boldsymbol{v}_{\mathrm{2nd\text{-}Cos}}^k(\boldsymbol{R}, \boldsymbol{R}')\big)_i := \quad \mathrm{cos\text{-}sim}\,\big((\boldsymbol{S}_{i,j_1}, \ldots, \boldsymbol{S}_{i,j_{K(i)}}), (\boldsymbol{S}'_{i,j_1}, \ldots, \boldsymbol{S}'_{i,j_{K(i)}})\big),$$

where $\boldsymbol{S}, \boldsymbol{S}'$ denote the RSMs w.r.t. cosine similarity.

*Rank Similarity* (Wang et al., 2022) not only considers the cardinality of the overlap between instance-wise neighborhoods $\mathcal{N}_{\boldsymbol{R}'}^k(i)$, but also factors in the order of common neighbors with respect to cosine similarity. To increase the importance of close neighbors, this measure defines distance-based ranks $r_{\boldsymbol{R}_i}(j)$ for all $j \in \mathcal{N}_{\boldsymbol{R}}^k(i)$, where $r_{\boldsymbol{R}_i}(j) = n$ if $\boldsymbol{R}_j$ is the $n$-th closest neighbor of $\boldsymbol{R}_i$. Based on these ranks, the instance-wise similarities are then defined as

$$\big(\boldsymbol{v}_{\mathrm{RankSim}}^k(\boldsymbol{R}, \boldsymbol{R}')\big)_i = \frac{1}{(\boldsymbol{v}_{\max})_i} \cdot \sum_{j \in \mathcal{N}_{\boldsymbol{R}}^k(i) \cap \mathcal{N}_{\boldsymbol{R}'}^k(i)} \frac{2}{(1 + |r_{\boldsymbol{R}_i}(j) - r_{\boldsymbol{R}_i'}(j)|)(r_{\boldsymbol{R}_i}(j) + r_{\boldsymbol{R}_i'}(j))}, \tag{25}$$

where $(\boldsymbol{v}_{\max})_i$ is a normalization factor that limits the maximum of the ranking similarity to one.

**Topology-based Measures.** Measures of this category aim to approximate and compare lower-dimensional data manifolds, which the high-dimensional representations are assumed to be concentrated on. This approximation is often done using discrete structures such as graphs, and this approach has also been chosen by Tsitsulin et al. (2020) for their *Multi-Scale Intrinsic Distance* (IMD) measure. Specifically, they considered $k$-NN graphs $\mathcal{G}(\boldsymbol{R})$ for each representation, which are then compared through their *heat kernel trace*, which is defined as $\mathrm{hkt}_{\mathcal{G}(\boldsymbol{R})}(t) = \sum_i e^{-t\lambda_i}$, with $\lambda_i$ denoting the eigenvalues of the normalized graph Laplacian of $\mathcal{G}(\boldsymbol{R})$. The IMD score is then defined as

$$m_{\mathrm{IMD}}(\boldsymbol{R}, \boldsymbol{R}') = \sup_{t>0} e^{-2(t+t^{-1})} |\mathrm{hkt}_{\mathcal{G}(\boldsymbol{R})}(t) - \mathrm{hkt}_{\mathcal{G}(\boldsymbol{R}')}(t)|. \tag{26}$$

Similarly, the *Representation Topology Divergence* (RTD) (Barannikov et al., 2022) considers graphs $\mathcal{G}^{\alpha}(\boldsymbol{R})$, in which nodes are connected if the Euclidean distance of the corresponding instance representations is lower than $\alpha$, and union graphs $\mathcal{G}^{\alpha}(\boldsymbol{R}, \boldsymbol{R}')$, where edges are formed if in at least one of the two representations this condition is satisfied. Based on these graphs, one collects a set $\mathcal{B}(\boldsymbol{R}, \boldsymbol{R}')$ of intervals $(\alpha_1, \alpha_2)$, in which these graphs differ in the number of their connected components. The total length of these intervals, denoted as $b(\boldsymbol{R}, \boldsymbol{R}') = \sum_{(\alpha_1,\alpha_2) \in \mathcal{B}(\boldsymbol{R},\boldsymbol{R}')} \alpha_2 - \alpha_1$,

then quantifies similarity between two representations. Given that this similarity score is not symmetric, one, however, further takes the average

$$m_{\text{RTD}}(\boldsymbol{R}, \boldsymbol{R}') = \tfrac{1}{2}(b(\boldsymbol{R}, \boldsymbol{R}') + b(\boldsymbol{R}', \boldsymbol{R})). \tag{27}$$

Instead of computing one score for the full representation matrices $\boldsymbol{R}, \boldsymbol{R}'$, Barannikov et al. (2022) further recommend sampling multiple subsets of instances based on which this score can be computed, and again averaging the resulting scores in the end.

**Descriptive Statistics.** The final set of measures that we test within our benchmark considers statistical properties of individual representations $\boldsymbol{R}$. Given a pair of representations $\boldsymbol{R}, \boldsymbol{R}'$, and a statistic $m_{\text{stat}}$, the individual statistics are then compared by taking their absolute difference:

$$m_{\text{StatDiff}} = |m_{\text{stat}}(\boldsymbol{R}) - m_{\text{stat}}(\boldsymbol{R}')|. \tag{28}$$

Toward that end, the first statistic we consider is *Magnitude* (Wang et al., 2022), which corresponds to the length of the mean instance representation:

$$m_{\text{Mag}}(\boldsymbol{R}) := \|\tfrac{1}{N} \sum_{i=1}^{N} \boldsymbol{R}_i\|_2. \tag{29}$$

*Concentricity* (Wang et al., 2022), by contrast, considers the average distance of each instance representation to the mean representation:

$$m_{\text{Conc}}(\boldsymbol{R}) := \tfrac{1}{N} \sum_{i=1}^{N} \text{cos-sim}(\boldsymbol{R}_i, \tfrac{1}{N} \sum_{j=1}^{N} \boldsymbol{R}_j). \tag{30}$$

Finally, *Uniformity* (Wang & Isola, 2020) measures how close the distribution of instance representations is to a uniform distribution on the unit hypersphere, and is defined as

$$m_{\text{Unif}}(\boldsymbol{R}) = \log \left( \tfrac{1}{N^2} \sum_{i=1}^{N} \sum_{j=1}^{N} e^{-t\|\boldsymbol{R}_i - \boldsymbol{R}_j\|_2^2} \right), \tag{31}$$

where $t$ is a hyperparameter Wang & Isola (2020).

## A.2 HYPERPARAMETER CHOICES

Regarding the hyperparameters of the measures under study, we largely followed recommendations made in the original references. Specifically, we made the following choices:

- For SVCCA, we included as many principal components as necessary to explain 99% of the variance in the training data.
- For RSMDiff, we computed the RSMs based on Euclidean distance
- For RSA, we applied Pearson correlation as inner and Spearman correlation as outer similarity function.
- For CKA, we used the original, non-batched implementation, and chose a linear kernel as similarity function.
- For the distance correlation (DistCorr), we computed RSMs based on Euclidean distance.
- In GULP, we set the regularization weight to $\lambda = 0$.
- For all neighborhood-based measures (Jaccard, RankSim, 2nd-Cos), we set the neighborhood size to $k = 10$ and determined nearest neighbors based on cosine similarity.
- For the IMD score, we used 8,000 approximation steps and five repetitions.
- For RTD, we always sampled 10 subsets of 500 instances.
- For the uniformity difference (UnifDiff), we set $t = 2$.

All other measures did not provide any hyperparameters.

## B EXPERIMENT DETAILS

Complementing the higher level experiment descriptions in the main manuscript, a more detailed description of them is provided in this section. This includes general information about the chosen model architectures, datasets and hyperparameters for each domain, with additional details that are unique to individual experiments provided later.

## B.1 REPRESENTATION EXTRACTION

In order to measure representational similarity, representations need to be extracted from the trained architectures. As noted in the main manuscript, we use the representations of the last hidden layer for all experiments, except the monotonicity experiment, where we use additional hidden layers of the architectures as well. We chose to do so as we can verify our assumptions of a model learning shortcuts or learning to be more robust to augmentations with the predictions directly derived from these representations. In an earlier layer, for example, we would not be able to verify whether a model learned the shortcut or not. For the output correlation, a similar argument can be made, as predictions are directly derived from these representations. Only for the layer monotonicity test, intermediate representations were used. Moreover, we carefully extract representations at locations where no residual connection bypasses the location of representation extraction to ensure that we measure the entirety of the representation and not just a part of it, i.e., after transformer and residual blocks. This is especially important for the monotonicity experiment where a disregard for this may invalidate the setting assumptions. Architectures, where this was important to consider, were all Transformer architectures, as well as the vision-specific ResNets.

**Graphs.** For the graph neural networks, the extracted representations naturally follow the desired matrix format $\boldsymbol{R} \in \mathbb{R}^{N \times D}$, where each row corresponds to the representation of a specific node in the network. Thus, no additional processing was necessary.

**Language.** Transformers produce one representation per token of each input. As we focus on the last-layer representations, we focus on the representation of the CLS token as the representation for the whole input for BERT and ALBERT. As this token is used by the final classifier, we argue that all relevant information for that input will be contained in this representation making this token representation the most important one. However, we also compare models via representations mean-pooled over all tokens of an input to give representations of other tokens more direct influence. For SmolLM2, we use the final token of the input as it decides what the next prediction is, similar to the CLS token for BERT and ALBERT.

While it is possible to compare representations of all tokens, inputs with many tokens would take up more rows in the final representation matrix, which could bias similarity estimates towards similarity of long inputs. Additionally, the runtime of many representational similarity measures scale quadratically in the number of rows of the representations, which practically limits the size of the representations given the many comparisons that we had to run in our benchmark.

**Vision.** Vision models vary in their representations. CNNs generally conduct a global average pooling operation before their classifier, leading to representations being in the desired $\boldsymbol{R} \in \mathbb{R}^{N \times D}$ format. Transformer-based vision models yield a CLS token in conjunction to additional tokens, hence we follow the language domain and only use the CLS token when comparing ViT representations.

In the monotonicity experiment, CNN representations were extracted prior to the global average pooling layer, resulting in representations with varying spatial extents. This poses challenges for measures that leverage spatial extent to increase sample size $N$, as such measures often assume correspondence between features at identical spatial positions. Furthermore, for spatial dimensions of $w = 112$ and $h = 112$, the number of values in the representation exceeds 10,000, leading to potential memory issues. To address these concerns, we restricted representation extraction to layers with a global stride $\geq 8$, yielding $w = h = 28$ in the ImageNet experiments. Additionally, we limited the extraction to the final six layers of the architecture. The extracted representations were subsequently scaled to a uniform $7 \times 7$ spatial extent via average pooling, ensuring consistency across all layers except the classifier layer, which lacks spatial dimensions. For spatial representation comparisons, the spatial dimensions $w$ and $h$ were folded into the channel dimension $c$, converting representations of shape $N \times c \times h \times w$ to $N \times (chw)$, unless the measure required $N > D$ or exhibited prohibitively high computational complexity. In such cases, the spatial extent was moved into the sample dimension, reshaping $N \times c \times h \times w$ to $(Nhw) \times c$. To mitigate memory and time constraints, the sample size was reduced from $Nhw$ to $N$ by subsampling every 49th value.

For comparisons between spatial and non-spatial representations, spatial dimensions were removed entirely through average pooling.

We note that this resampling and subsampling of the representations in the *CNN* case is suboptimal, yet this issue originates from the limitations of the similarity measure, namely requiring $N > D$ or scaling prohibitively.

Moreover, in all vision experiments, representations were extracted from test images that models had not seen during training. For ImageNet100, the 50 validation cases per class are used, resulting in $N = 5000$ samples, and for CIFAR100 we use the full test dataset.

## B.2 GENERAL EXPERIMENT DETAILS

Across all modalities and experiments, efforts were made to keep architectures, training hyperparameters or dataset choices as static as possible, while minimizing the overall compute effort that was necessary to compare representations with all these measures. In the following, we describe general experimental settings which apply to all tests, separately for each domain.

### B.2.1 GRAPH SETTINGS

**Models and Parametrization.**    As graph neural network architectures, we chose the *GCN* (Kipf & Welling, 2017), *GraphSAGE* (Hamilton et al., 2017), and *GAT* (Veličković et al., 2018) models due to their widespread prominence. In our implementation, we used the respective model classes as provided in the `PyTorch Geometric` (Fey & Lenssen, 2019) package. We further applied the *P-GNN* (You et al., 2019) architecture, since its position-aware approach to aggregation of node information provides a contrast to the other three models, which only aggregate information of neighboring nodes in their convolutions. We applied the reference implementation by the authors, which was adapted to the given node classification tasks. Due to its large memory consumption, it was only feasible to apply this model on the Cora dataset. Further, we did not use this model for the augmentation test, because the *DropEdge* approach that we use for this test is not appropriate for position-aware embeddings – dropping edges would alter the distances of nodes to the corresponding anchor sets, and these distances are the key element of the message passing of P-GNNs. Overall, we also think that the consideration of datasets, where the corresponding task fits better to the strength of these position-aware GNNs, may be a good extension for future studies that build on our benchmark.

For each experiment and dataset, we trained these models from scratch. Regarding the hyperparameters, we roughly followed values that were used for these models in existing benchmarks. An overview of the hyperparameter choices for each algorithm and dataset are presented in Table 4. The layer count also includes the classification layer, thus the number of inner layers is one less than the number given in the table. For the GAT model, we always used $h = 8$ attention heads. We always used the *Adam* optimizer (Kingma & Ba, 2015) as implemented in `PyTorch` (Paszke et al., 2019) to optimize a cross-entropy loss on the given multiclass datasets. To validate the performance, we plot the training, validation and test accuracies of the final models in Figure 4.

All these parameter choices were consistently applied in all our benchmark tests, except for Test 6 (layer monotonicity), where we increased the number of layers to six, yielding five inner layers.

Table 4: Hyperparameters for all architectures on the respective datasets in the graph domain.

| Dataset | Architecture | Dimension | Layers | Activation | Dropout Rate | Learning Rate | Weight Decay | Epochs |
|---|---|---|---|---|---|---|---|---|
| Cora | GCN | 64 | 3 | relu | 0.5 | 0.01 | 0.0 | 200 |
| | GraphSAGE | 64 | 3 | relu | 0.5 | 0.005 | 0.0005 | 200 |
| | GAT | 64 | 2 | elu | 0.6 | 0.005 | 0.0005 | 200 |
| | P-GNN | 32 | 2 | relu | 0.5 | 0.01 | 0.0 | 200 |
| Flickr | GCN | 256 | 3 | relu | 0.5 | 0.01 | 0.0 | 200 |
| | GraphSAGE | 256 | 3 | relu | 0.5 | 0.01 | 0.0 | 200 |
| | GAT | 256 | 3 | relu | 0.5 | 0.01 | 0.0 | 200 |
| OGBN-Arxiv | GCN | 256 | 3 | relu | 0.5 | 0.01 | 0.0 | 500 |
| | GraphSAGE | 256 | 3 | relu | 0.5 | 0.005 | 0.0005 | 500 |
| | GAT | 256 | 3 | relu | 0.5 | 0.01 | 0.0 | 500 |

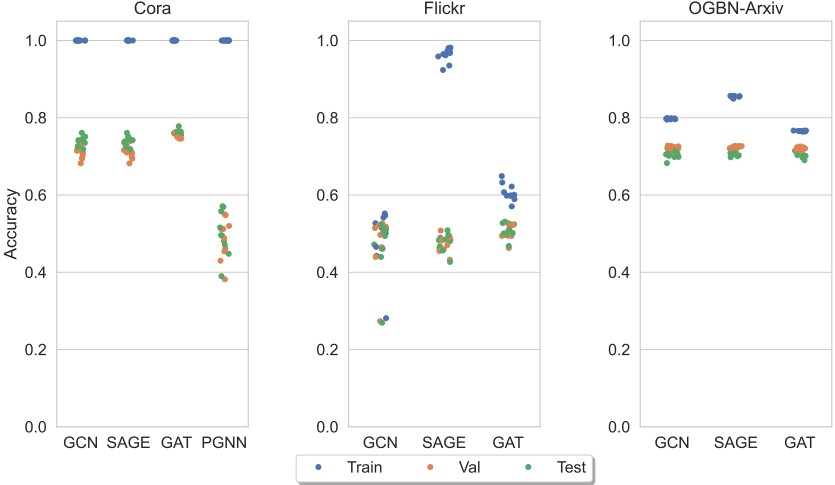

Figure 4: *Validation accuracies of GNN models trained with standard parametrization.* We used standard train/validation/test-splits provided with the given datasets. Test accuracies largely correspond to those obtained in common benchmarks.

**Datasets.**    We focus on graph datasets that provide multiclass labels for node classification, and for which dataset splits into training, validation and test sets are already available. Thus, we selected the following networks.

1. *Cora* (Yang et al., 2016): in this citation network, nodes represent documents and edges represent citation links. Node features are given by bag-of-words representations of the corresponding documents, and node labels correspond to topical categories of the papers.

2. *Flickr* (Zeng et al., 2020): in this network, nodes represent images uploaded to Flickr, and edges are formed if two images share some common properties, such as geographic location, gallery, or users who have commented on it. Node features correspond to bag-of-word representations of the images, classes were formed based on image tags.

3. *OGBN-Arxiv* (Hu et al., 2020): this dataset represents a citation network between Computer Science papers on *arXiv*. Each node corresponds to a paper, and directed edges indicate that one paper cites another one. Node features are given by aggregated word embeddings of paper titles and abstracts, labels correspond to the subject area of each paper.

Statistics of the datasets can be found in Table 5.

### B.2.2  LANGUAGE SETTINGS.

**Models.**    We use BERT (Devlin et al., 2019), ALBERT (Lan et al., 2020) and SmolLM2 (Allal et al., 2025) models. BERT and ALBERT are encoder-only transformer models, whereas SmolLM2 is a decoder-only transformer. Pretraining of these models is expensive, so we rely on publicly available models, which we will fine-tune for our tests. To make the tests challenging, the fine-tuning must induce non-negligible differences in behavior and representations. Otherwise, it would be trivial to find models of the same group in the design-based grounding tests, for instance. We generally fine-tune with different seeds, but BERT models in their whole are relatively little affected by this

Table 5: Statistics of the graph datasets used within the `ReSi` benchmark

| Dataset | Nodes | Edges | Features | Classes |
|---|---|---|---|---|
| Cora | 2,708 | 10,556 | 1,433 | 7 |
| Flickr | 89,250 | 899,756 | 500 | 7 |
| OGBN-Arxiv | 169,343 | 1,166,243 | 128 | 40 |

approach (Merchant et al., 2020). Thus, we start fine-tuning of BERT from different pretrained models, provided by Sellam et al. (2022). For ALBERT, we fine-tune the same pretrained model[3] as the weights are shared across layers, which should increase the changes induced in the model by fine-tuning. Finally, for SmolLM2, we finetune the 1.7B base model[4].

**Datasets.** We used two classification datasets: SST2 (Socher et al., 2013) is a collection of sentences extracted from movie reviews labeled with positive or negative sentiment. MNLI (Williams et al., 2018) is a dataset of premise-hypothesis pairs. They are labeled according to whether the hypothesis follows from the premise. We used the validation and validation-matched subsets to extract representations for SST2 and MNLI, respectively.

**Training Hyperparameters.** As we used the pretrained models from Sellam et al. (2022) for BERT, we only fine-tuned the models for our experiments. While the number of epochs varies between 3 and 10 depending on the experiment, we always used the checkpoint with the best validation performance. We generally used a linear learning rate schedule with 10% warm up to a maximum of 5e-5, evaluate every 1000 steps, and used a batch size of 64. Otherwise, we used default hyperparameters of the `transformers` library[5]. For ALBERT, the training is identical with the exception of always training for 10 epochs. For SmolLM2, we use full finetuning for 500 steps with a batchsize of 16. Otherwise, we use the default hyperparameters as released with the finetuning script of the SmolLM2 release.

### B.2.3 VISION SETTINGS

**Architecture Choices.** In the vision domain, a plethora of architectures exist which could have been used in the scope of the benchmark. To narrow the scope and to keep computational overhead manageable, only architectures for classification were considered. Moreover, a subset of architecture of prominent classification architecture families were evaluated, namely ResNets (He et al., 2016), ViTs Dosovitskiy et al. (2021) and older VGG's Simonyan & Zisserman (2015). To capture whether architecture size influences the measures, different sizes for the architectures were evaluated, resulting in 1. *ResNet18*, 2. *ResNet34*, 3. *ResNet101*, 4. *VGG19*, 5. *ViT-B/32* and 6. *ViT-L/32* as the final architecture choice. Pre-trained checkpoints for ViT-B/32 and ViT-L/32 use are taken from `huggingface` provided under the Apache 2.0. Specific checkpoints are *'google/vit-base-patch32-224-in21k'* and *'google/vit-large-patch32-224-in21k'*.

**Datasets.** In the vision domain, we use the ImageNet100 (IN100) and the CIFAR-100 (Krizhevsky & Hinton, 2009) datasets. IN100 is derived from ImageNet1k (Russakovsky et al., 2015) by subsampling 100 randomly chosen classes. This reduces training time while keeping image resolution and content similar to ImageNet1k. When training the models on the IN100 dataset, the image resolution was fixed to 224x224, while the resolution was set to 32x32 when training

---

[3]`https://huggingface.co/albert/albert-base-v2`

[4]`https://huggingface.co/HuggingFaceTB/SmolLM2-1.7B`

[5]`https://huggingface.co/docs/transformers/v4.40.2/en/main_classes/trainer#transformers.TrainingArguments`

Table 6: Vision domain: Training hyperparameters for all architectures on the ImageNet100 dataset. Aside from the listed parameters, we note that all models shared more hyperparameters, namely *label smoothing* of 0.1 and a cosine annealing *learning rate* schedule.

| Dataset | Architecture | Batch Size | Learning Rate | Weight Decay | Optimizer | Epochs |
|---------|-------------|-----------|---------------|--------------|-----------|--------|
| ImageNet100 | ResNet18 | 128 | 0.1 | 4e-5 | SGD | 200 |
| | ResNet34 | 128 | 0.1 | 4e-5 | SGD | 200 |
| | ResNet101 | 128 | 0.1 | 4e-5 | SGD | 200 |
| | VGG11 | 128 | 0.1 | 4e-5 | SGD | 200 |
| | VGG19 | 128 | 0.1 | 4e-5 | SGD | 200 |
| | ViT-B32 | 512 | 3e-3 | 0.1 | AdamW | 300 |
| | ViT-L32 | 512 | 3e-3 | 0.1 | AdamW | 300 |

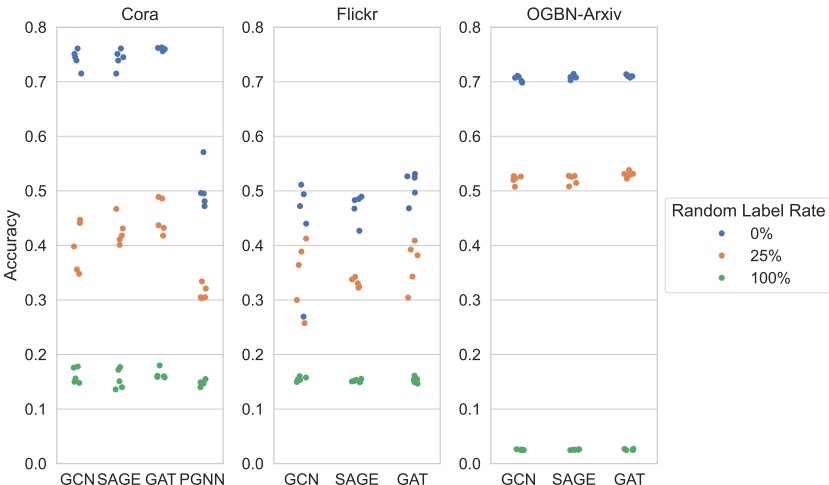

Figure 5: *Validation accuracies of GNN models trained for Test 3 (Label Randomization).* Accuracies were computed on test sets with regular labels. With increasing degree of randomization of target labels, performance degraded strongly. Clusters of accuracies per group are clearly separated.

models on the CIFAR-100 dataset. A notable exception are the ViT models trained on CIFAR-100, which used images CIFAR-100 images that were upsampled from 32x32 to 224x224, as proposed by Touvron et al. (2021). This intervention was necessary to remain compatibility to the pre-trained weights of the patch encoder.

**Training Hyperparameters.** The `ReSi` benchmark requires models that can be differentiated by their training setting. Hence, it is mandatory to either train models from scratch or to fine-tune them accordingly. For all models of the vision domain, the choice was made to train the models from scratch for each respective dataset, except for ViTs whose encoders were initialized from a pre-trained IN21k checkpoint. The chosen training hyperparameters are kept static across tests. A selection of hyperparameters is displayed in Table 6. While the parameter choices worked well for ResNets and VGGs, the ViT performance was lower than the ResNets, despite attempts to optimize their training settings. We assume this may originate from the fewer overall steps taken due to the 90% lower dataset size relative to full ImageNet1k.

### B.2.4 COMPUTE RESOURCES

To conduct the experiments, a broad spectrum of hardware was used: For the model training, GPU nodes with up to 80GB VRAM were employed. Depending on domain, representations were either extracted on GPU nodes and saved to disk for later processing, or extracted on demand on CPU nodes. Lastly, the representational similarity measures were calculated between representations on CPU nodes with 6-256 CPU cores and working memory between 80 and 1024 GB.

To extend the current tests with additional similarity measures, practitioners need to have CPU compute resources to compute their own similarity measures on the existing model's representations. To introduce novel tests, practitioners would need to provide GPU compute resources for network training and subsequent CPU resources to evaluate all similarity measures on the new models.

### B.3 TEST 3: LABEL RANDOMIZATION - DETAILS

**Graphs.** We trained groups of five models each, for which $r \in \{25\%, 50\%, 75\%, 100\%\}$ of all labels are randomized during training. The models in each group were initialized with different training seeds, and different seeds also imply different randomization of labels. Given that we exclusively consider predictions on datasets with $L \geq 7$ classes, for all labels that were drawn to be randomized, we randomly drew a new label from the set of all existing labels, excluding the actual ground-truth label, with uniform probability. After observing the performance of the models

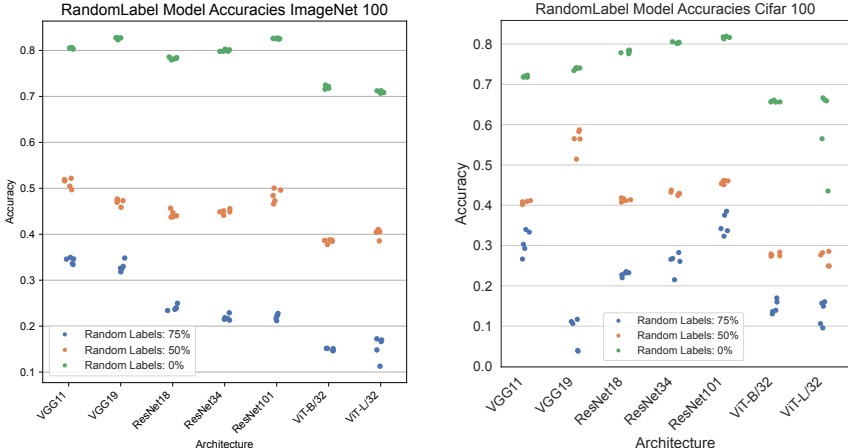

Figure 6: *Validation accuracies of vision models trained for Test 3 (Label Randomization).* Training with varying degrees of random labels resulted in models of highly different accuracies. We highlight the three clusters used in the Label Randomization test for the vision domain.

with randomized labels on validation and test data, we decided to only include models for which $r \in \{25\%, 100\%\}$ were altered during training for this test, along with five models trained on the regular data. This is mainly due to the models already having shown behavior indicating that for higher degrees of randomization than 25%, there was hardly any relationship left between features and true labels that the models would have learned. We illustrate the performance of the models trained under varying degrees of label randomization in Figure 5, where we can see that the models are strongly affected by such randomization.

**Language.** We create datasets where $r \in \{25\%, 50\%, 75\%, 100\%\}$ of all labels are randomized so they can only be learned by memorization. If an instance should get a randomized label, we pick uniformly at random from a new set of five labels that are entirely unrelated to the data. After validating behavior, we use models without randomization of 0%, 75%, and 100% for BERT and ALBERT. For SmolLM2, we use 0% and 100%.

**Vision.** We create datasets where $r \in \{25\%, 50\%, 75\%, 100\%\}$ of all labels in the training set are randomized, forcing models to memorize. Whenever a label is chosen to be randomized, we randomly sample from all 100 existing labels, and replace the true label with the sampled one. Moreover, we re-use the first five models from Test 1 and 2 which represent a dataset with $r = 0$. After validating behavior, we use models with 0%, 50%, 75%, with clusters visualized in Figure 6.

### B.4 TEST 4: SHORTCUT AFFINITY - DETAILS

**Graphs.** For all graph datasets, shortcuts were added by appending a one-hot encoding of the corresponding labels to the node features. Nodes which were not assigned a true level shortcut obtain a one-hot encoding of a random label instead. We trained models with a ratio of $\rho \in \{0\%, 25\%, 50\%, 75\%, 100\%\}$ true shortcuts in the training data, but only considered the models trained on with $\rho \in \{0\%, 50\%, 100\%\}$ shortcuts in the training data for the benchmark, as otherwise there would not have been a very clear separation of the models. We illustrate the performance of models influenced by the remaining degrees of shortcut likelihood in the training data in Figure 7, where we can see that most models are indeed affected. In particular, GraphSAGE appears very sensitive to shortcuts, whereas other methods appear more robust - this may also be due to differences in the way node information is updated in the convolutional layers.
Overall, on Cora the effect was the smallest, which is likely due to the models fitting easily to the training labels even without a shortcut.

For all groups and seeds, we extracted representations based on the same fixed dataset, which contained one-hot encodings of random labels.

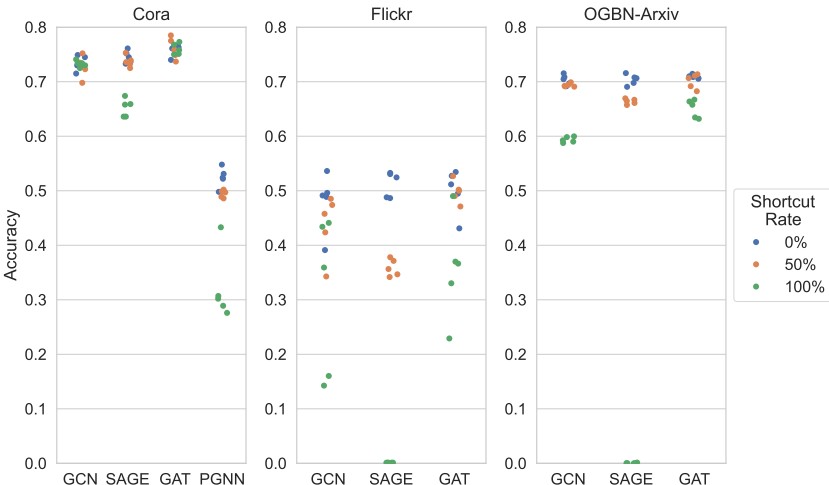

Figure 7: *Validation accuracies of GNN models trained for Test 4 (Shortcut Affinity).* Accuracies were computed on test sets with random shortcuts, but regular labels. With increasing degree of shortcut correlation with target labels, performance generally degraded strongly. Only for Cora, the effect was weaker, likely due to the models not requiring the shortcuts for strong performance. Still, clusters of accuracies per group are mostly well-separated.

**Language.** We added shortcuts to the data by prepending a new special token to each sentence for BERT and ALBERT. Their embedding matrix was extended with one token per class and trained during fine-tuning. We created datasets with five levels of shortcut strength, i.e., the rate at which the shortcut leaks the correct label. For SmolLM2, we added the correct answer token as a shortcut during the task prompt. As completely incorrect shortcut tokens would perfectly leak the label in binary problems like SST2, the lowest rate is the relative frequency of the majority class. Then we made four equally sized steps up to perfect shortcuts. This means that SST2 datasets have shortcut rates $r \in \{55.8\%, 66.8\%, 77.9\%, 88.9\%, 100\%\}$ and for MNLI we have $r \in \{35.4\%, 67.7\%, 51.55\%, 83.85\%, 100\%\}$. We trained models with five different seeds for each shortcut dataset, but after validating sufficiently different behavior (see Figure 13, Figure 14, and Figure 15), we only used the models with the lowest and the two highest rates.

**Vision.** In the vision domain, shortcuts were created through insertion of a colored dot. Each dot has a static color, a diameter of $5$, and is placed randomly in the image. To turn this into a shortcut, each class is assigned a unique color code, which is uniformly drawn once and shared across all experiments. By varying how often the pre-determined color appears on an image containing the corresponding class label, one can control how useful the color dot feature is to the class in the image. A few exemplary images of the shortcuts are shown in Figure 8.

We trained models under five degrees of correlation $\rho \in \{0\%, 25\%, 50\%, 75\%, 100\%\}$, providing a more or less potent shortcut. To validate that these different ratios lead to sufficiently different behavior that our grouping assumption holds, we measure the test set accuracy of all models with $r = 0\%$. Models that learned to utilize the shortcut will show a stronger decrease in accuracy as the shortcut does not provide any meaningful signal anymore. In order to provide clearly distinguishable clusters, we discarded models trained with $r \in \{25\%, 50\%\}$. The accuracy clusters after removing the two groups are shown in Figure 9.

## B.5   TEST 5: AUGMENTATION - DETAILS

**Graphs.** To augment the graph data during training, we applied the *DropEdge* approach (Rong et al., 2020), which, at each training epoch, samples a fixed fraction of edges that are removed from the network for this specific epoch. Based on this approach, we trained four groups of augmented models, for which $r \in \{20\%, 40\%, 60\%, 80\%\}$ of all edges were removed at each epoch. Again, after considering the performance of the models on validation and test data, we chose to only include

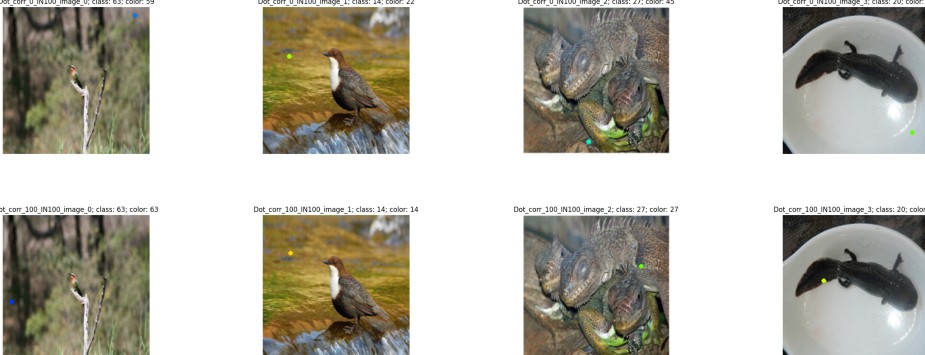

Figure 8: *Visualization of the shortcuts introduced in the vision domain.* We statically assign each class label a color code representing the class. In our benchmark, we control the extent as to which the color dot correlates with the true label, thereby making the shortcut more or less useful. In the top row, the cases where color dots are placed randomly are displayed, while at the bottom the color labels of the dot always correspond to the class label.

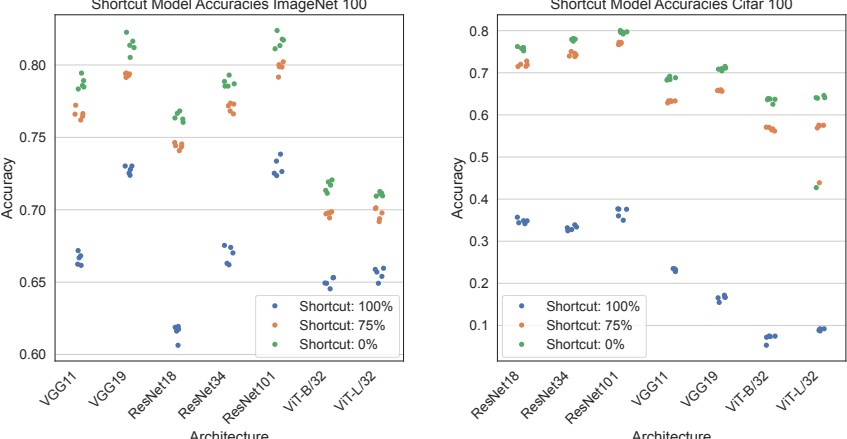

Figure 9: *Validation accuracies of vision models trained for Test 4 (Shortcut Affinity).* Models trained with highly correlated shortcuts show lower accuracy when the correlation of the shortcut feature with the image label is removed. After evaluating all trained models under this setting, we removed groups that were not well separated from others.

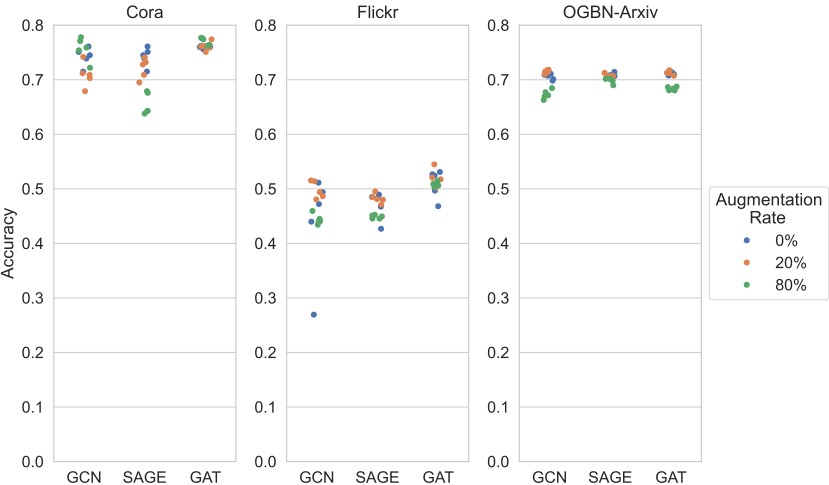

Figure 10: *Validation accuracies of GNN models trained for Test 5 (Augmentation).* Accuracies were computed on unaltered test sets. In general, augmentation impacts the performance of models, though the these effect differ over models and datasets. In general, clusters of accuracies per group are identifiable.

the models with $r \in \{20\%, 80\%\}$ for this test along with a group of models trained with standard parameters, which was due to the overall weak effect impact that the intermediate steps appeared to have. The performance of these remaining models is shown in Figure 10, where we overall observe weak, but visible effect from augmentation.

**Language.** BERT has been trained on large-scale data of varying quality and, as preliminary experiments showed, highly robust to augmentation like word casing, synonyms, and typos[6]. Therefore, we chose more aggressive augmentation that can also scale to large amounts of data. We randomly replace words with synonyms, delete and swap words, and randomly insert a synonym of a word in the sentence in a random position, as described by Wei & Zou (2019) and implemented in the `textattack` library[7] (Morris et al., 2020). To create datasets with different level of augmentation, we augmented $r \in \{0\%, 25\%, 50\%, 75\%, 100\%\}$ of all words per sentence. We train five models with different seeds per dataset. For SST2, we use the 0% and 100% models; for MNLI, we use 0%, 25%, and 100%for BERT models and 0% and 100% for ALBERT models. We do not implement this test for SmolLM2.

**Vision.** For the augmentation test, we utilized increasing levels of *additive Gaussian noise* with five different noise variance intervals $[0, v]$, $v \in \{0, 1000, 2000, 3000, 4000\}$. We refer to these levels as **Off**, **S**, **M**, **L** and **Max**. Exemplary images of the augmented images are displayed in Figure 11. Aside from the noise, only *RandomResizeCrop* and *HorizontalFlip* augmentations are used. After validating model behavior, we decided to use **Max**, **S** and **M** for our benchmark, see Figure 12 for the resulting clusters.

## B.6 TEST 6: LAYER MONOTONICITY - DETAILS

**Graphs.** Since for the graph neural network models, we never used more than three layers in the previous experiments, for this test we trained a new set of five graph neural network models, each with five inner layers. That way, we can obtain a sufficient number of comparisons, while still preserving decent performance—with higher number of layers, the performance began to strongly deteriorate. As all inner layer shared the same number of channels, no preprocessing was necessary to compare the layer-wise representations.

---

[6]We tested most of the augmentations from `https://langtest.org/docs/pages/tests/robustness`.

[7]`https://textattack.readthedocs.io/`

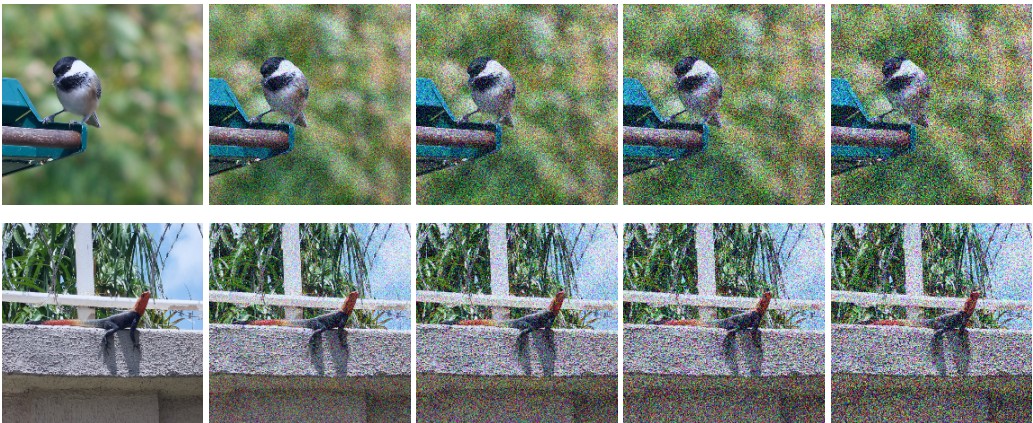

Figure 11: *Visualization of the additive Gaussian noise used in the vision augmentation test.* We show an exemplary image to highlight the gradually increasing level of noise introduced. From left to right (**Off**, **S**, **M**, **L**, **Max**).

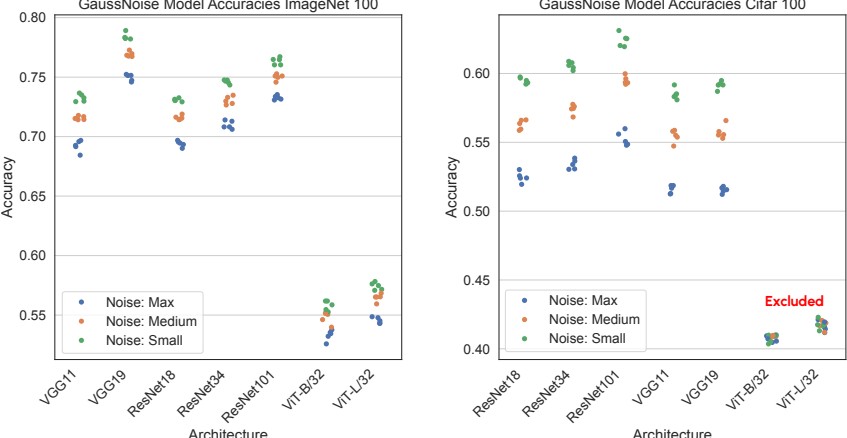

Figure 12: *Validation accuracies of vision models trained for Test 5 (Augmentation).* Models trained with varying degrees of additive Gaussian noise show lower accuracy on noise-free samples. Two groups were removed to improve separation. Further, ViT-B/32 and ViT-L/32 did not qualify the group-separation requirement and were excluded.

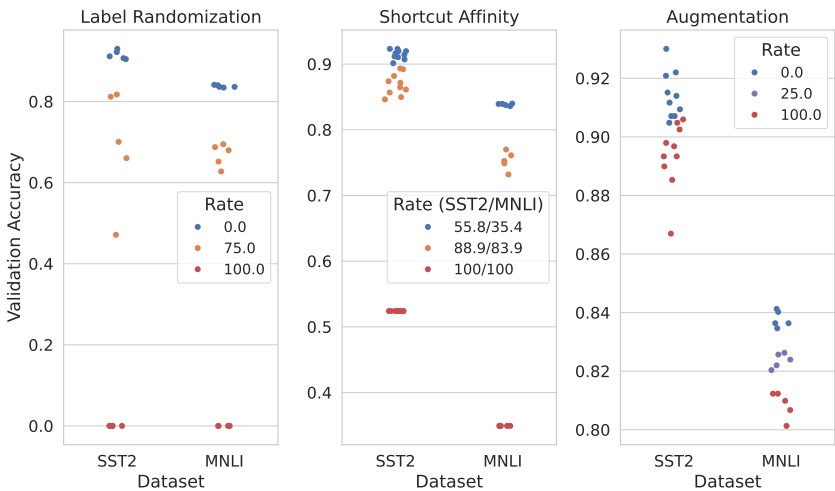

Figure 13: *Validation accuracies of BERT models trained for tests 3-5.* Across all tests, groups are clearly separated.

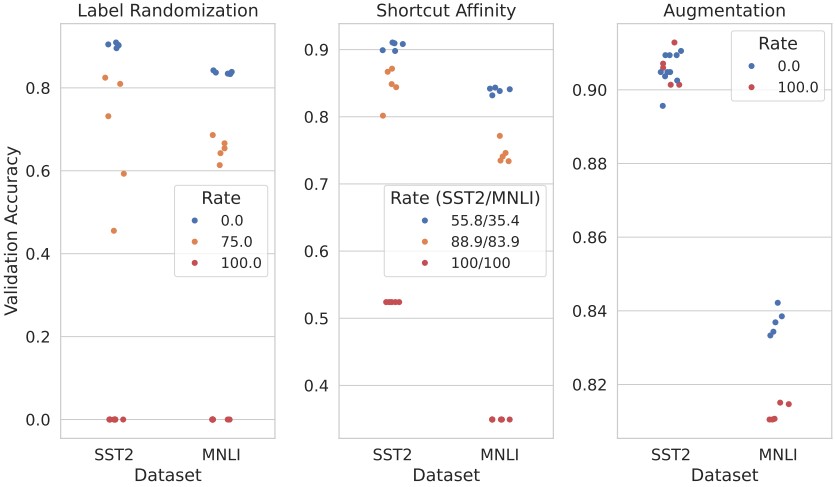

Figure 14: *Validation accuracies of ALBERT models trained for tests 3-5.* For SST2, there is no separation between groups. Thus, we exclude the augmentation test on SST2 for ALBERT.

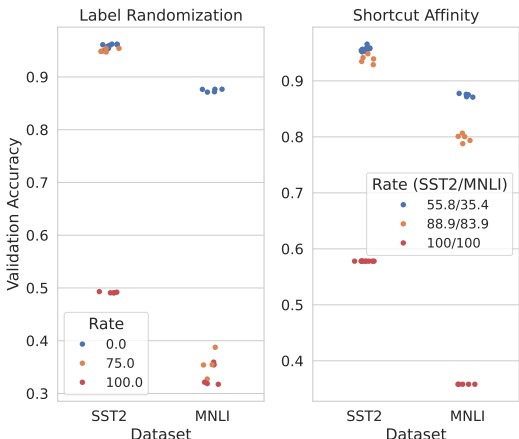

Figure 15: *Validation accuracies of SmolLM2 models trained for tests 3 and 4.* For the label randomization test, we only use two groups as the three different training setups cannot be distinguished.

**Language.** We used ten models trained on standard data with varying seeds for each dataset. We used the CLS token representations or mean-pooled representations across all 12 transformer blocks and the embedding layer for BERT and ALBERT. For SmolLM2, we again use the final token of the prompt. As the skip-connections only skip layers inside the block, we argue that representations after the blocks should follow our assumptions.

**Vision.** For the layer monotonicity test, we used the 10 models from Test 1 and 2. Differently to those tests, we extracted representations not only before the classifier, but we also needed to extract representations in intermediate layers.

## C   FULL BENCHMARK RESULTS

In the following, we provide the full results of all experiments within our benchmark. For the tests grounded by prediction, we report statistical significance of the shown Spearman correlations on the 5% (*) and the 1% (**) level.

### C.1   GRAPH RESULTS

We begin with presenting the results from the graph domain. Results are shown in Tables 7-12, with each table presenting results of a single test. For the PWCCA measure, we report some `NaN` values—in these cases, the reference implementation consistently yields negative eigenvalues for a covariance matrix, which then results in undefined values when the square root of these eigenvalues is taken to compute the pseudo-inverse of this matrix.

Table 7: Results of Test 1 (*Correlation to Accuracy Difference*) for the graph domain.

| | Eval. Dataset Arch. | Spearman | | | | | | | | | |
| | | Cora | | | | Flickr | | | OGBN-Arxiv | | |
| | | GCN | SAGE | GAT | PGNN | GCN | SAGE | GAT | GCN | SAGE | GAT |
|---|---|---|---|---|---|---|---|---|---|---|---|
| CCA | PWCCA | -0.05 | 0.07 | -0.26* | -0.09 | nan | -0.05 | -0.16 | -0.09 | 0.06 | -0.30** |
| | SVCCA | -0.02 | -0.13 | -0.19 | -0.33* | 0.01 | 0.01 | -0.18 | -0.27 | 0.10 | -0.10 |
| Alignment | AlignCos | -0.33* | 0.13 | -0.29 | -0.08 | 0.35* | 0.24 | -0.07 | -0.08 | 0.17 | -0.17 |
| | AngShape | 0.15 | -0.29 | -0.12 | -0.02 | 0.39** | 0.28 | -0.15 | -0.04 | 0.09 | -0.09 |
| | HardCorr | 0.11 | -0.11 | -0.14 | -0.12 | 0.31* | 0.35* | 0.06 | **0.37*** | 0.02 | 0.04 |
| | LinReg | 0.06 | -0.21* | -0.11 | -0.14 | -0.04 | 0.17 | -0.18 | 0.07 | -0.01 | -0.19 |
| | OrthProc | 0.15 | -0.29 | -0.12 | -0.02 | 0.39** | 0.28 | -0.15 | -0.04 | 0.09 | -0.09 |
| | PermProc | -0.12 | **0.18** | -0.26 | **0.29** | 0.20 | -0.19 | 0.15 | -0.09 | 0.03 | **0.43**** |
| | ProcDist | 0.08 | 0.01 | -0.19 | **0.29** | 0.02 | -0.06 | 0.11 | -0.17 | 0.07 | **0.43**** |
| | SoftCorr | **0.18** | -0.05 | 0.03 | -0.02 | 0.30* | 0.33* | -0.07 | 0.35* | 0.12 | 0.12 |
| RSM | CKA | 0.16 | -0.17 | 0.02 | -0.08 | 0.03 | 0.27 | -0.16 | -0.17 | 0.11 | -0.05 |
| | DistCorr | 0.01 | -0.17 | 0.03 | 0.17 | 0.41** | **0.42**** | -0.19 | -0.10 | 0.15 | -0.06 |
| | EOS | -0.24 | 0.08 | -0.05 | -0.04 | 0.15 | -0.27 | **0.29** | -0.21 | 0.05 | -0.32* |
| | GULP | -0.43** | 0.08 | -0.12 | -0.02 | 0.04 | -0.27 | -0.27 | -0.08 | 0.06 | -0.34* |
| | RSA | -0.27 | 0.04 | -0.28 | 0.21 | 0.53** | 0.32* | -0.08 | -0.07 | 0.25 | 0.32* |
| | RSMDiff | -0.19 | 0.07 | -0.14 | 0.24 | -0.18 | -0.16 | 0.13 | -0.05 | -0.19 | 0.02 |
| Neighbors | 2nd-Cos | -0.10 | 0.04 | -0.11 | 0.13 | **0.54**** | -0.19 | 0.01 | **-0.47**** | 0.22 | -0.19 |
| | Jaccard | 0.02 | -0.11 | -0.16 | 0.07 | 0.32* | 0.28 | -0.18 | -0.32* | -0.13 | -0.14 |
| | RankSim | -0.00 | -0.10 | **0.32*** | 0.23 | 0.35* | 0.31* | -0.20 | -0.28 | 0.05 | -0.09 |
| Topology | IMD | -0.10 | 0.00 | -0.03 | 0.04 | -0.10 | 0.37* | -0.09 | -0.21 | -0.02 | -0.15 |
| | RTD | 0.06 | -0.26 | 0.13 | -0.44** | 0.19 | 0.13 | -0.17 | -0.23 | 0.15 | -0.33* |
| Statistic | ConcDiff | 0.15 | -0.25 | -0.20 | 0.13 | -0.08 | -0.29 | -0.07 | -0.07 | -0.13 | -0.12 |
| | MagDiff | 0.08 | -0.13 | -0.19 | 0.18 | 0.02 | -0.17 | 0.14 | -0.18 | -0.20 | 0.11 |
| | UnifDiff | -0.10 | -0.04 | -0.09 | -0.13 | -0.18 | 0.03 | -0.18 | -0.19 | -0.20 | -0.25 |

Table 8: Results of Test 2 (*Correlation to Output Difference*) for the graph domain.

| Type Test Eval. Dataset Arch. | | Grounding by Prediction | | | | | | | | | | | | | | | | | | | |
| | | JSD Corr. Spearman | | | | | | | | | | Disagr. Corr. Spearman | | | | | | | | | |
| | | Cora | | | | Flickr | | | OGBN-Arxiv | | | Cora | | | | Flickr | | | OGBN-Arxiv | | |
| | | GCN | SAGE | GAT | PGNN | GCN | SAGE | GAT | GCN | SAGE | GAT | GCN | SAGE | GAT | PGNN | GCN | SAGE | GAT | GCN | SAGE | GAT |
|---|---|---|---|---|---|---|---|---|---|---|---|---|---|---|---|---|---|---|---|---|---|
| CCA | PWCCA | 0.08 | 0.27* | 0.02 | -0.37** | nan | 0.38** | 0.27* | -0.22* | -0.04 | 0.29** | 0.08 | 0.36** | -0.14 | 0.06 | nan | 0.32** | 0.16 | -0.24* | -0.08 | 0.03 |
| | SVCCA | -0.09 | 0.02 | 0.26 | -0.06 | -0.04 | 0.23 | 0.13 | 0.19 | 0.16 | -0.17 | -0.06 | 0.04 | 0.17 | 0.02 | -0.27 | 0.21 | 0.03 | 0.08 | 0.11 | -0.15 |
| Alignment | AlignCos | 0.02 | 0.38** | 0.17 | -0.15 | 0.31* | 0.44** | -0.01 | -0.03 | 0.05 | 0.28 | -0.08 | 0.27 | -0.05 | 0.01 | 0.15 | 0.37* | -0.08 | -0.10 | 0.00 | 0.17 |
| | AngShape | 0.70** | 0.27 | 0.28 | -0.15 | 0.43** | 0.63** | 0.13 | 0.04 | -0.15 | 0.44** | 0.62** | 0.16 | 0.05 | 0.20 | 0.19 | 0.57** | 0.03 | 0.02 | -0.16 | 0.27 |
| | HardCorr | 0.48** | 0.16 | 0.52** | -0.15 | 0.53** | 0.50** | 0.09 | -0.05 | -0.28 | 0.46** | 0.45** | 0.10 | 0.26 | 0.27 | 0.40** | 0.46** | -0.09 | 0.02 | -0.24 | 0.24 |
| | LinReg | 0.58** | 0.33** | 0.19 | -0.27* | 0.15 | 0.48** | 0.18 | -0.09 | -0.17 | 0.47** | 0.57** | 0.09 | -0.06 | 0.10 | 0.12 | 0.46** | 0.06 | -0.10 | -0.19 | 0.22* |
| | OrthProc | 0.70** | 0.27 | 0.28 | -0.15 | 0.43** | 0.63** | 0.13 | 0.04 | -0.15 | 0.44** | 0.62** | 0.16 | 0.05 | 0.20 | 0.19 | 0.57** | 0.03 | 0.02 | -0.16 | 0.27 |
| | PermProc | 0.29 | 0.10 | 0.13 | 0.22 | 0.29 | -0.10 | -0.42** | 0.34* | -0.55** | 0.06 | 0.24 | 0.33* | -0.14 | 0.46** | 0.18 | -0.10 | -0.27 | 0.25 | -0.41** | 0.22 |
| | ProcDist | 0.52** | 0.08 | 0.05 | 0.22 | 0.02 | -0.18 | -0.38* | 0.36* | -0.50** | 0.13 | 0.45** | 0.24 | -0.17 | 0.46** | -0.13 | -0.15 | -0.27 | 0.26 | -0.38** | 0.30* |
| | SoftCorr | 0.53** | -0.01 | 0.21 | -0.13 | 0.53** | 0.53** | 0.23 | 0.02 | -0.28 | 0.56** | 0.49** | 0.06 | 0.07 | 0.23 | 0.39** | 0.49** | 0.02 | 0.03 | -0.26 | 0.35* |
| RSM | CKA | 0.43** | -0.03 | 0.53** | -0.25 | 0.03 | 0.58** | 0.17 | 0.12 | -0.02 | 0.11 | 0.36* | 0.00 | 0.45** | 0.07 | -0.21 | 0.53** | 0.06 | 0.03 | -0.04 | 0.23 |
| | DistCorr | 0.60** | 0.05 | 0.60** | -0.28 | 0.46** | 0.43** | 0.03 | 0.16 | 0.12 | 0.36* | 0.53** | -0.08 | 0.46** | 0.10 | 0.17 | 0.40** | -0.03 | 0.11 | 0.08 | 0.20 |
| | EOS | 0.17 | 0.22 | -0.04 | -0.26 | -0.03 | 0.38** | 0.11 | -0.17 | 0.11 | 0.37* | 0.13 | 0.02 | -0.09 | 0.17 | 0.02 | 0.33* | 0.23 | -0.25 | -0.02 | 0.12 |
| | GULP | -0.10 | 0.18 | -0.04 | 0.19 | 0.15 | 0.38** | 0.13 | 0.10 | 0.11 | 0.35* | -0.17 | -0.09 | -0.13 | 0.53** | 0.03 | 0.33* | -0.01 | 0.15 | -0.04 | 0.10 |
| | RSA | 0.12 | 0.13 | 0.20 | 0.21 | 0.52** | 0.63** | 0.14 | 0.20 | 0.14 | 0.46** | 0.06 | 0.33* | -0.02 | 0.38* | 0.35* | 0.59** | 0.06 | 0.17 | 0.15 | 0.45** |
| | RSMDiff | 0.11 | -0.01 | -0.07 | 0.18 | -0.22 | -0.04 | -0.37* | -0.20 | 0.11 | 0.33* | 0.05 | 0.14 | -0.15 | 0.43** | -0.30* | -0.00 | -0.26 | -0.14 | 0.09 | 0.36* |
| Neighbors | 2nd-Cos | 0.43** | 0.54** | 0.53** | -0.19 | 0.47** | 0.15 | 0.07 | 0.11 | 0.34* | 0.25 | 0.40** | 0.25 | 0.30* | 0.02 | 0.30* | 0.14 | -0.10 | 0.04 | 0.27 | 0.06 |
| | Jaccard | 0.44** | 0.45** | 0.38* | -0.02 | 0.33* | 0.42** | 0.11 | 0.20 | 0.09 | 0.37* | 0.44** | 0.33* | 0.12 | 0.24 | 0.04 | 0.42** | -0.00 | 0.14 | 0.01 | 0.22 |
| | RankSim | 0.36* | 0.56** | 0.51** | 0.06 | 0.33* | 0.30* | 0.21 | 0.02 | 0.36* | 0.13 | 0.34* | 0.23 | 0.44** | 0.24 | 0.06 | 0.24 | 0.05 | 0.03 | 0.27 | 0.10 |
| Topology | IMD | 0.24 | -0.38* | -0.09 | 0.05 | -0.08 | 0.29 | 0.04 | -0.13 | -0.03 | 0.01 | 0.20 | -0.11 | -0.13 | 0.01 | -0.10 | 0.33* | -0.02 | -0.13 | 0.01 | 0.04 |
| | RTD | 0.54** | 0.49** | 0.28 | 0.07 | 0.19 | -0.14 | 0.04 | -0.06 | 0.16 | -0.11 | 0.48** | 0.25 | 0.24 | -0.11 | -0.01 | -0.12 | 0.01 | -0.11 | 0.07 | -0.19 |
| Statistic | ConcDiff | 0.03 | 0.04 | 0.01 | 0.34* | -0.17 | -0.03 | 0.03 | -0.13 | 0.02 | -0.22 | -0.04 | -0.10 | 0.03 | 0.13 | -0.21 | -0.04 | 0.03 | -0.25 | 0.07 | -0.16 |
| | MagDiff | 0.00 | 0.06 | -0.15 | 0.24 | 0.03 | 0.06 | -0.26 | -0.13 | -0.13 | 0.08 | -0.07 | -0.25 | -0.07 | 0.36* | 0.06 | 0.07 | -0.20 | -0.25 | -0.19 | 0.22 |
| | UnifDiff | 0.13 | 0.27 | -0.11 | -0.20 | -0.32* | 0.02 | 0.21 | -0.34* | 0.12 | 0.12 | 0.07 | 0.16 | -0.15 | 0.15 | -0.34* | 0.04 | 0.18 | -0.33* | 0.10 | 0.06 |

Table 9: Results of Test 3 (*Label Randomization*) for the graph domain.

| | Eval. Dataset Arch. | AUPRC | | | | | | | | | | Conformity Rate | | | | | | | | | |
| | | Cora | | | | Flickr | | | OGBN-Arxiv | | | Cora | | | | Flickr | | | OGBN-Arxiv | | |
| | | GCN | SAGE | GAT | PGNN | GCN | SAGE | GAT | GCN | SAGE | GAT | GCN | SAGE | GAT | PGNN | GCN | SAGE | GAT | GCN | SAGE | GAT |
|---|---|---|---|---|---|---|---|---|---|---|---|---|---|---|---|---|---|---|---|---|---|
| CCA | PWCCA | 0.26 | 0.33 | 0.28 | 0.25 | nan | 0.44 | 0.24 | 0.24 | 0.44 | 0.36 | 0.60 | 0.52 | 0.64 | 0.58 | nan | 0.64 | 0.53 | 0.43 | 0.61 | 0.56 |
| | SVCCA | 0.35 | 0.31 | 0.42 | 0.20 | 0.33 | 0.80 | 0.27 | 1.00 | 0.56 | 0.93 | 0.62 | 0.56 | 0.59 | 0.39 | 0.69 | 0.91 | 0.54 | 1.00 | 1.00 | 0.96 |
| Alignment | AlignCos | 0.46 | 0.48 | 0.43 | 0.23 | 0.84 | 0.42 | 0.29 | 0.98 | 0.70 | 0.93 | 0.66 | 0.66 | 0.60 | 0.60 | 0.94 | 0.50 | 0.58 | 1.00 | 0.67 | 0.97 |
| | AngShape | 0.43 | 0.42 | 0.42 | 0.23 | **0.88** | 0.43 | 0.27 | 0.83 | 0.73 | 0.77 | 0.54 | 0.51 | 0.52 | 0.60 | 0.95 | 0.60 | 0.53 | 0.97 | 0.84 | 0.88 |
| | HardCorr | 0.42 | 0.42 | 0.42 | 0.21 | 0.77 | 0.46 | 0.33 | 0.83 | 0.83 | 0.54 | 0.53 | 0.51 | 0.51 | 0.54 | 0.94 | 0.68 | 0.67 | 0.97 | 0.97 | 0.77 |
| | LinReg | 0.49 | 0.43 | 0.46 | 0.24 | 0.22 | 0.45 | 0.23 | 0.44 | 0.68 | 0.47 | 0.71 | 0.58 | 0.69 | 0.54 | 0.49 | 0.66 | 0.52 | 0.63 | 0.81 | 0.63 |
| | OrthProc | 0.43 | 0.42 | 0.42 | 0.23 | **0.88** | 0.43 | 0.27 | 0.83 | 0.73 | 0.77 | 0.54 | 0.51 | 0.52 | 0.60 | 0.95 | 0.60 | 0.53 | 0.97 | 0.84 | 0.88 |
| | PermProc | 0.44 | 0.39 | 0.44 | 0.27 | 0.77 | 0.90 | 0.19 | 0.68 | 0.72 | 0.93 | 0.64 | 0.71 | 0.61 | 0.52 | 0.94 | **0.97** | 0.50 | 0.84 | 0.88 | 0.98 |
| | ProcDist | 0.45 | 0.45 | 0.43 | 0.27 | 0.79 | 0.62 | 0.19 | 0.92 | 0.82 | **1.00** | 0.67 | 0.68 | 0.60 | 0.51 | 0.93 | 0.88 | 0.57 | 0.98 | **1.00** | 0.98 |
| | SoftCorr | 0.42 | 0.42 | 0.42 | 0.23 | 0.60 | 0.45 | 0.33 | 0.83 | 0.82 | 0.55 | 0.53 | 0.51 | 0.50 | 0.55 | 0.85 | 0.67 | 0.65 | 0.97 | 0.96 | 0.71 |
| RSM | CKA | 0.43 | 0.42 | 0.42 | 0.24 | 0.73 | 0.66 | 0.27 | **1.00** | **1.00** | **1.00** | 0.55 | 0.51 | 0.51 | 0.58 | 0.91 | 0.86 | 0.54 | **1.00** | **1.00** | **1.00** |
| | DistCorr | 0.43 | 0.44 | 0.42 | 0.25 | 0.86 | 0.43 | 0.22 | **1.00** | **1.00** | **1.00** | 0.59 | 0.63 | 0.52 | **0.66** | 0.95 | 0.56 | 0.52 | **1.00** | **1.00** | **1.00** |
| | EOS | 0.30 | 0.26 | 0.28 | 0.26 | 0.41 | 0.42 | 0.22 | 0.25 | 0.43 | 0.34 | 0.65 | 0.55 | 0.66 | 0.54 | 0.50 | 0.52 | 0.58 | 0.50 | 0.54 | 0.51 |
| | GULP | 0.28 | 0.26 | 0.28 | 0.26 | 0.19 | 0.42 | 0.23 | 0.30 | 0.43 | 0.30 | 0.63 | 0.56 | 0.66 | 0.54 | 0.50 | 0.51 | 0.57 | 0.49 | 0.55 | 0.56 |
| | RSA | 0.46 | 0.44 | 0.43 | 0.23 | 0.74 | 0.42 | 0.33 | 0.96 | 0.43 | 0.49 | 0.69 | 0.61 | 0.54 | 0.58 | 0.89 | 0.52 | 0.64 | 0.99 | 0.58 | 0.63 |
| | RSMDiff | 0.52 | 0.53 | 0.78 | **0.29** | 0.71 | **0.92** | 0.19 | **1.00** | **1.00** | **1.00** | 0.73 | 0.83 | 0.86 | 0.61 | 0.91 | **0.97** | 0.51 | **1.00** | **1.00** | **1.00** |
| Neighbors | 2nd-Cos | 0.56 | 0.73 | 0.73 | 0.21 | 0.61 | 0.42 | 0.36 | 0.99 | 0.96 | 0.95 | 0.80 | 0.87 | 0.86 | 0.51 | 0.82 | 0.50 | 0.66 | **1.00** | 0.99 | 0.99 |
| | Jaccard | 0.42 | 0.42 | 0.43 | 0.24 | 0.56 | 0.43 | 0.29 | 0.83 | 0.43 | 0.78 | 0.51 | 0.55 | 0.55 | 0.57 | 0.77 | 0.58 | 0.57 | 0.97 | 0.63 | 0.93 |
| | RankSim | 0.45 | 0.42 | 0.47 | 0.22 | 0.47 | 0.43 | 0.33 | 0.84 | 0.55 | 0.78 | 0.65 | 0.54 | 0.64 | 0.54 | 0.66 | 0.53 | 0.58 | 0.97 | 0.63 | 0.93 |
| Topology | IMD | **0.73** | **0.97** | 0.84 | 0.23 | 0.24 | 0.37 | 0.23 | **1.00** | 0.90 | 0.90 | 0.93 | **0.99** | 0.96 | 0.54 | 0.47 | 0.70 | 0.54 | **1.00** | **1.00** | 0.97 |
| | RTD | 0.56 | 0.82 | **0.92** | 0.20 | 0.86 | 0.59 | 0.22 | 0.93 | 0.87 | 0.93 | 0.80 | 0.96 | **0.98** | 0.50 | **0.96** | 0.83 | 0.49 | 0.99 | 0.97 | 0.99 |
| Statistic | ConcDiff | 0.20 | 0.39 | 0.39 | 0.18 | 0.37 | 0.57 | 0.21 | 0.96 | **1.00** | **1.00** | 0.54 | 0.78 | 0.74 | 0.46 | 0.72 | 0.85 | 0.48 | 0.99 | **1.00** | **1.00** |
| | MagDiff | 0.20 | 0.24 | 0.37 | 0.28 | 0.66 | 0.72 | 0.18 | 0.55 | 0.49 | 0.34 | 0.51 | 0.60 | 0.74 | 0.61 | 0.89 | 0.93 | 0.47 | 0.83 | 0.86 | 0.75 |
| | UnifDiff | 0.33 | 0.50 | 0.32 | **0.29** | 0.53 | 0.90 | **0.40** | 0.53 | 0.54 | 0.33 | 0.70 | 0.81 | 0.67 | 0.59 | 0.84 | 0.96 | **0.75** | 0.78 | 0.81 | 0.66 |

Table 10: Results of Test 4 (*Shortcut Affinity*) for the graph domain.

| | Eval. Dataset Arch. | AUPRC | | | | | | | | | | Conformity Rate | | | | | | | | | |
| --- | --- | --- | --- | --- | --- | --- | --- | --- | --- | --- | --- | --- | --- | --- | --- | --- | --- | --- | --- | --- | --- |
| | | Cora | | | | Flickr | | | OGBN-Arxiv | | | Cora | | | | Flickr | | | OGBN-Arxiv | | |
| | | GCN | SAGE | GAT | PGNN | GCN | SAGE | GAT | GCN | SAGE | GAT | GCN | SAGE | GAT | PGNN | GCN | SAGE | GAT | GCN | SAGE | GAT |
| CCA | PWCCA | 0.33 | 0.47 | 0.22 | 0.38 | nan | 0.43 | 0.33 | 0.99 | 0.72 | **1.00** | 0.73 | 0.71 | 0.56 | 0.50 | nan | 0.60 | 0.73 | **1.00** | 0.83 | **1.00** |
| | SVCCA | 0.23 | 0.36 | 0.46 | 0.24 | 0.24 | 0.93 | 0.32 | **1.00** | 0.97 | 0.83 | 0.46 | 0.60 | 0.69 | 0.52 | 0.57 | 0.97 | 0.66 | **1.00** | 0.99 | 0.91 |
| Alignment | AlignCos | 0.64 | 0.60 | 0.59 | 0.40 | 0.89 | **1.00** | 0.48 | **1.00** | **1.00** | **1.00** | 0.84 | 0.88 | 0.89 | 0.63 | 0.98 | **1.00** | 0.77 | **1.00** | **1.00** | **1.00** |
| | AngShape | 0.63 | 0.79 | 0.62 | 0.35 | 0.58 | **1.00** | 0.31 | **1.00** | **1.00** | **1.00** | 0.79 | 0.91 | 0.90 | 0.63 | 0.80 | **1.00** | 0.72 | **1.00** | **1.00** | **1.00** |
| | HardCorr | 0.28 | 0.35 | 0.29 | 0.35 | 0.55 | **1.00** | 0.52 | 0.80 | 0.72 | 0.83 | 0.65 | 0.77 | 0.70 | 0.69 | 0.83 | **1.00** | 0.81 | 0.96 | 0.83 | 0.97 |
| | LinReg | 0.70 | 0.74 | 0.50 | 0.34 | 0.37 | 0.61 | 0.36 | **1.00** | **1.00** | **1.00** | 0.88 | 0.95 | 0.75 | 0.62 | 0.70 | 0.81 | 0.72 | **1.00** | **1.00** | **1.00** |
| | OrthProc | 0.63 | 0.79 | 0.62 | 0.35 | 0.58 | **1.00** | 0.31 | **1.00** | **1.00** | **1.00** | 0.79 | 0.91 | 0.90 | 0.63 | 0.80 | **1.00** | 0.72 | **1.00** | **1.00** | **1.00** |
| | PermProc | 0.23 | 0.27 | 0.25 | 0.25 | 0.42 | **1.00** | 0.27 | 0.65 | 0.43 | 0.77 | 0.57 | 0.67 | 0.64 | 0.57 | 0.72 | **1.00** | 0.58 | 0.86 | 0.60 | 0.91 |
| | ProcDist | 0.60 | 0.81 | 0.68 | 0.25 | 0.60 | **1.00** | 0.33 | **1.00** | **1.00** | **1.00** | 0.77 | 0.91 | 0.92 | 0.58 | 0.82 | **1.00** | 0.61 | **1.00** | **1.00** | **1.00** |
| | SoftCorr | 0.31 | 0.40 | 0.30 | 0.39 | 0.66 | **1.00** | **0.57** | 0.82 | 0.72 | 0.83 | 0.68 | 0.79 | 0.71 | 0.67 | 0.87 | **1.00** | 0.82 | 0.96 | 0.83 | 0.97 |
| RSM | CKA | 0.61 | 0.78 | 0.78 | 0.34 | 0.28 | **1.00** | 0.33 | **1.00** | 0.98 | **1.00** | 0.75 | 0.90 | 0.94 | 0.63 | 0.57 | **1.00** | 0.65 | **1.00** | **1.00** | **1.00** |
| | DistCorr | 0.66 | 0.82 | 0.80 | 0.42 | 0.33 | **1.00** | 0.32 | **1.00** | 0.99 | **1.00** | 0.83 | 0.92 | 0.95 | 0.62 | 0.69 | **1.00** | 0.72 | **1.00** | **1.00** | **1.00** |
| | EOS | 0.30 | 0.50 | 0.20 | 0.37 | 0.46 | 0.43 | 0.54 | **1.00** | 0.72 | 0.97 | 0.71 | 0.74 | 0.46 | 0.50 | 0.87 | 0.60 | 0.81 | **1.00** | 0.83 | 0.99 |
| | GULP | 0.31 | 0.50 | 0.20 | 0.37 | 0.20 | 0.43 | 0.45 | 0.99 | 0.72 | 0.96 | 0.72 | 0.74 | 0.46 | 0.50 | 0.49 | 0.60 | 0.81 | **1.00** | 0.83 | 0.99 |
| | RSA | 0.49 | 0.43 | 0.73 | 0.42 | 0.89 | **1.00** | 0.52 | 0.97 | 0.98 | 0.90 | 0.72 | 0.78 | 0.91 | **0.70** | 0.97 | **1.00** | 0.81 | 0.99 | 0.99 | 0.98 |
| | RSMDiff | 0.46 | 0.50 | 0.69 | 0.37 | 0.48 | 0.92 | 0.36 | **1.00** | **1.00** | **1.00** | 0.70 | 0.76 | 0.91 | 0.64 | 0.75 | 0.97 | 0.59 | **1.00** | **1.00** | **1.00** |
| Neighbors | 2nd-Cos | **0.85** | **0.91** | 0.81 | 0.34 | **1.00** | **1.00** | 0.52 | **1.00** | **1.00** | **1.00** | 0.95 | **0.97** | 0.95 | 0.68 | **1.00** | **1.00** | 0.82 | **1.00** | **1.00** | **1.00** |
| | Jaccard | 0.74 | 0.78 | **0.86** | 0.42 | **1.00** | 0.83 | 0.54 | **1.00** | **1.00** | **1.00** | 0.91 | 0.93 | **0.97** | 0.53 | **1.00** | 0.96 | **0.84** | **1.00** | **1.00** | **1.00** |
| | RankSim | 0.73 | 0.64 | 0.85 | **0.43** | **1.00** | 0.77 | 0.54 | **1.00** | **1.00** | **1.00** | 0.87 | 0.89 | 0.95 | 0.58 | **1.00** | 0.92 | 0.83 | **1.00** | **1.00** | **1.00** |
| Topology | IMD | 0.71 | 0.82 | 0.60 | 0.25 | 0.78 | 0.97 | 0.36 | 0.61 | 0.92 | 0.92 | 0.92 | 0.96 | 0.88 | 0.68 | 0.94 | 0.99 | 0.62 | 0.84 | 0.98 | 0.97 |
| | RTD | 0.71 | 0.89 | 0.71 | 0.35 | 0.77 | **1.00** | 0.25 | **1.00** | **1.00** | **1.00** | 0.87 | **0.97** | 0.93 | 0.64 | 0.90 | **1.00** | 0.63 | **1.00** | **1.00** | **1.00** |
| Statistic | ConcDiff | 0.51 | 0.17 | 0.20 | 0.17 | 0.18 | 0.18 | 0.32 | 0.81 | 0.96 | **1.00** | 0.77 | 0.43 | 0.54 | 0.44 | 0.50 | 0.46 | 0.61 | 0.96 | 0.99 | **1.00** |
| | MagDiff | 0.37 | 0.15 | 0.18 | 0.21 | 0.27 | 0.78 | 0.20 | 0.55 | 0.53 | **1.00** | 0.74 | 0.41 | 0.49 | 0.58 | 0.67 | 0.89 | 0.48 | 0.82 | 0.82 | **1.00** |
| | UnifDiff | 0.30 | 0.64 | 0.71 | 0.29 | 0.59 | 0.50 | 0.48 | 0.99 | 0.72 | 0.97 | 0.62 | 0.91 | 0.91 | 0.65 | 0.83 | 0.77 | 0.67 | **1.00** | 0.86 | 0.99 |

Table 11: Results of Test 5 (*Augmentation*) for the graph domain.

| | Eval. Dataset Arch. | AUPRC | | | | | | | | | Conformity Rate | | | | | | | | |
| --- | --- | --- | --- | --- | --- | --- | --- | --- | --- | --- | --- | --- | --- | --- | --- | --- | --- | --- | --- |
| | | Cora | | | Flickr | | | OGBN-Arxiv | | | Cora | | | Flickr | | | OGBN-Arxiv | | |
| | | GCN | SAGE | GAT | GCN | SAGE | GAT | GCN | SAGE | GAT | GCN | SAGE | GAT | GCN | SAGE | GAT | GCN | SAGE | GAT |
| CCA | PWCCA | 0.60 | 0.65 | 0.42 | nan | 0.57 | 0.51 | nan | 0.48 | 0.82 | 0.86 | 0.89 | 0.83 | nan | 0.83 | 0.75 | nan | 0.74 | 0.96 |
| | SVCCA | 0.27 | 0.29 | 0.34 | 0.64 | 0.67 | 0.49 | 0.89 | 0.81 | 0.82 | 0.54 | 0.55 | 0.58 | 0.84 | 0.86 | 0.68 | 0.89 | 0.85 | 0.91 |
| Alignment | AlignCos | 0.51 | 0.63 | 0.91 | 0.69 | 0.70 | 0.54 | 0.74 | 0.43 | **1.00** | 0.78 | 0.88 | 0.98 | 0.87 | 0.90 | 0.81 | 0.87 | 0.54 | **1.00** |
| | AngShape | 0.81 | 0.78 | 0.83 | 0.63 | 0.76 | 0.54 | 0.72 | 0.72 | 0.74 | 0.95 | 0.93 | 0.95 | 0.85 | 0.94 | 0.77 | 0.83 | 0.83 | 0.89 |
| | HardCorr | 0.51 | 0.63 | 0.51 | 0.71 | 0.72 | 0.57 | 0.47 | 0.51 | 0.54 | 0.85 | 0.89 | 0.81 | 0.89 | 0.92 | **0.85** | 0.59 | 0.77 | 0.79 |
| | LinReg | 0.94 | 0.85 | 0.87 | 0.30 | 0.81 | 0.34 | 0.72 | 0.72 | 0.81 | 0.98 | 0.96 | 0.95 | 0.53 | 0.93 | 0.69 | 0.83 | 0.83 | 0.93 |
| | OrthProc | 0.81 | 0.78 | 0.83 | 0.63 | 0.76 | 0.54 | 0.72 | 0.72 | 0.74 | 0.95 | 0.93 | 0.95 | 0.85 | 0.94 | 0.77 | 0.83 | 0.83 | 0.89 |
| | PermProc | 0.70 | 0.61 | 0.42 | 0.65 | 0.69 | 0.40 | 0.88 | 0.73 | 0.94 | 0.78 | 0.89 | 0.51 | 0.87 | 0.90 | 0.62 | 0.97 | 0.87 | 0.98 |
| | ProcDist | 0.76 | 0.69 | 0.75 | 0.70 | 0.81 | 0.40 | **1.00** | **1.00** | **1.00** | 0.88 | 0.90 | 0.92 | 0.89 | 0.95 | 0.63 | 0.51 | **1.00** | **1.00** |
| | SoftCorr | 0.63 | 0.70 | 0.50 | 0.73 | 0.58 | 0.55 | 0.43 | 0.46 | 0.45 | 0.89 | 0.92 | 0.80 | 0.87 | 0.85 | 0.84 | 0.51 | 0.71 | 0.66 |
| RSM | CKA | 0.70 | 0.73 | 0.91 | 0.57 | 0.75 | 0.51 | **1.00** | **1.00** | 0.99 | 0.90 | 0.92 | 0.97 | 0.83 | 0.91 | 0.74 | **1.00** | **1.00** | **1.00** |
| | DistCorr | 0.79 | 0.73 | 0.89 | 0.60 | 0.79 | 0.61 | **1.00** | **1.00** | 0.99 | 0.93 | 0.91 | 0.97 | 0.84 | 0.94 | 0.81 | **1.00** | **1.00** | **1.00** |
| | EOS | 0.53 | 0.62 | 0.34 | 0.68 | 0.53 | 0.46 | 0.82 | 0.49 | **1.00** | 0.86 | 0.90 | 0.75 | 0.88 | 0.80 | 0.70 | 0.96 | 0.75 | **1.00** |
| | GULP | 0.45 | 0.61 | 0.35 | 0.22 | 0.54 | 0.55 | 0.56 | 0.49 | **1.00** | 0.79 | 0.89 | 0.76 | 0.52 | 0.81 | 0.84 | 0.80 | 0.75 | **1.00** |
| | RSA | 0.39 | 0.57 | 0.78 | 0.68 | 0.72 | 0.59 | 0.75 | 0.49 | 0.67 | 0.68 | 0.84 | 0.94 | 0.86 | 0.92 | 0.84 | 0.89 | 0.78 | 0.85 |
| | RSMDiff | **0.99** | 0.79 | **1.00** | 0.64 | 0.93 | 0.40 | **1.00** | **1.00** | **1.00** | **1.00** | 0.93 | **1.00** | 0.87 | 0.97 | 0.62 | **1.00** | **1.00** | **1.00** |
| Neighbors | 2nd-Cos | 0.73 | **0.86** | 0.97 | **0.78** | 0.92 | **0.62** | **1.00** | 0.99 | **1.00** | 0.88 | **0.97** | 0.99 | **0.91** | 0.98 | 0.84 | **1.00** | **1.00** | **1.00** |
| | Jaccard | 0.78 | 0.81 | 0.95 | 0.74 | 0.88 | 0.58 | **1.00** | 0.99 | **1.00** | 0.90 | 0.95 | 0.98 | 0.88 | 0.97 | 0.84 | **1.00** | **1.00** | **1.00** |
| | RankSim | 0.54 | 0.62 | 0.95 | 0.75 | 0.88 | 0.55 | **1.00** | **1.00** | **1.00** | 0.72 | 0.88 | 0.98 | 0.89 | 0.96 | 0.83 | **1.00** | **1.00** | **1.00** |
| Topology | IMD | 0.74 | 0.34 | 0.20 | 0.59 | 0.57 | 0.28 | **1.00** | **1.00** | **1.00** | 0.93 | 0.64 | 0.46 | 0.87 | 0.86 | 0.53 | **1.00** | **1.00** | **1.00** |
| | RTD | 0.91 | 0.79 | 0.98 | 0.72 | **1.00** | 0.54 | **1.00** | **1.00** | **1.00** | 0.98 | 0.95 | 0.99 | 0.90 | **1.00** | 0.75 | **1.00** | **1.00** | **1.00** |
| Statistic | ConcDiff | 0.17 | 0.50 | 0.62 | 0.41 | 0.35 | 0.53 | 0.43 | 0.53 | 0.51 | 0.49 | 0.78 | 0.87 | 0.78 | 0.68 | 0.74 | 0.76 | 0.80 | 0.84 |
| | MagDiff | 0.20 | 0.39 | 0.54 | 0.59 | 0.17 | 0.43 | 0.77 | **1.00** | 0.69 | 0.51 | 0.74 | 0.83 | 0.87 | 0.45 | 0.66 | 0.93 | **1.00** | 0.91 |
| | UnifDiff | 0.75 | 0.53 | **1.00** | 0.68 | 0.24 | 0.25 | 0.76 | 0.53 | 0.45 | 0.91 | 0.86 | **1.00** | 0.90 | 0.52 | 0.66 | 0.88 | 0.81 | 0.78 |

Table 12: Results of Test 6 (*Layer Monotonicity*) for the graph domain.

| | Eval. Dataset Arch. | Conformity Rate | | | | | | | | | | Spearman | | | | | | | | | |
| --- | --- | --- | --- | --- | --- | --- | --- | --- | --- | --- | --- | --- | --- | --- | --- | --- | --- | --- | --- | --- | --- |
| | | Cora | | | | Flickr | | | OGBN-Arxiv | | | Cora | | | | Flickr | | | OGBN-Arxiv | | |
| | | GCN | SAGE | GAT | PGNN | GCN | SAGE | GAT | GCN | SAGE | GAT | GCN | SAGE | GAT | PGNN | GCN | SAGE | GAT | GCN | SAGE | GAT |
| CCA | PWCCA | **1.00** | **1.00** | **1.00** | 0.99 | **1.00** | **1.00** | 0.75 | **1.00** | **1.00** | **1.00** | **1.00** | **1.00** | **1.00** | **1.00** | nan | **1.00** | 0.37 | **1.00** | **1.00** | **1.00** |
| | SVCCA | 0.63 | 0.80 | 0.81 | **1.00** | 0.75 | 0.69 | 0.57 | 0.49 | 0.41 | 0.56 | 0.41 | 0.78 | 0.61 | **1.00** | 0.53 | 0.44 | 0.47 | -0.17 | -0.10 | 0.33 |
| Alignment | AlignCos | 0.97 | **1.00** | **1.00** | 0.64 | 0.59 | 0.77 | 0.46 | 0.83 | 0.93 | 0.89 | 0.98 | **1.00** | **1.00** | 0.45 | 0.33 | 0.68 | 0.04 | 0.84 | 0.93 | 0.84 |
| | AngShape | **1.00** | **1.00** | **1.00** | 0.97 | 0.80 | 0.99 | 0.66 | 0.95 | 0.99 | 0.99 | **1.00** | **1.00** | **1.00** | 0.91 | 0.62 | 0.98 | 0.45 | 0.97 | 0.99 | 0.99 |
| | HardCorr | 0.91 | 0.99 | 0.76 | 0.85 | 0.81 | 0.84 | 0.68 | 0.86 | 0.72 | **1.00** | 0.83 | 0.98 | 0.54 | 0.82 | 0.63 | 0.80 | 0.21 | 0.83 | 0.62 | **1.00** |
| | LinReg | **1.00** | **1.00** | **1.00** | 0.95 | 0.67 | **1.00** | 0.59 | 0.99 | **1.00** | **1.00** | **1.00** | **1.00** | **1.00** | 0.45 | **1.00** | 0.03 | | 0.99 | **1.00** | **1.00** |
| | OrthProc | **1.00** | **1.00** | **1.00** | 0.97 | 0.80 | 0.99 | 0.66 | 0.95 | 0.99 | 0.99 | **1.00** | **1.00** | **1.00** | 0.91 | 0.62 | 0.98 | 0.45 | 0.97 | 0.99 | 0.99 |
| | PermProc | 0.91 | **1.00** | **1.00** | **1.00** | 0.68 | 0.72 | 0.75 | 0.69 | 0.87 | **1.00** | 0.92 | **1.00** | **1.00** | **1.00** | 0.60 | 0.68 | 0.63 | 0.24 | 0.88 | **1.00** |
| | ProcDist | 0.99 | **1.00** | **1.00** | 0.97 | 0.75 | **1.00** | 0.78 | 0.93 | 0.96 | **1.00** | 0.99 | **1.00** | **1.00** | 0.91 | 0.66 | **1.00** | 0.62 | 0.83 | 0.85 | **1.00** |
| | SoftCorr | 0.95 | 0.99 | 0.78 | 0.82 | 0.89 | 0.88 | 0.64 | 0.91 | 0.73 | **1.00** | 0.96 | 0.99 | 0.50 | 0.74 | 0.79 | 0.89 | 0.11 | 0.91 | 0.62 | **1.00** |
| RSM | CKA | 0.98 | **1.00** | **1.00** | **1.00** | 0.79 | 0.92 | 0.59 | 0.87 | 0.64 | 0.95 | 0.98 | **1.00** | **1.00** | **1.00** | 0.60 | 0.89 | 0.39 | 0.85 | 0.46 | 0.96 |
| | DistCorr | **1.00** | **1.00** | **1.00** | 0.99 | 0.81 | 0.99 | 0.63 | 0.88 | 0.65 | 0.92 | **1.00** | **1.00** | **1.00** | **1.00** | 0.63 | 0.99 | 0.47 | 0.81 | 0.52 | 0.93 |
| | EOS | **1.00** | **1.00** | **1.00** | 0.98 | **1.00** | **1.00** | 0.92 | **1.00** | **1.00** | **1.00** | **1.00** | **1.00** | **1.00** | 0.99 | **1.00** | **1.00** | 0.92 | **1.00** | **1.00** | **1.00** |
| | GULP | **1.00** | **1.00** | **1.00** | 0.90 | 0.83 | **1.00** | 0.72 | 0.82 | **1.00** | **1.00** | **1.00** | **1.00** | **1.00** | 0.82 | 0.54 | **1.00** | 0.33 | 0.73 | **1.00** | **1.00** |
| | RSA | 0.90 | 0.97 | **1.00** | 0.72 | 0.70 | 0.98 | 0.70 | 0.81 | 0.97 | 0.94 | 0.84 | 0.91 | **1.00** | 0.52 | 0.58 | 0.99 | 0.44 | 0.52 | 0.97 | 0.94 |
| | RSMDiff | 0.99 | **1.00** | **1.00** | 0.97 | 0.66 | 0.68 | 0.92 | 0.85 | 0.93 | **1.00** | 0.99 | **1.00** | **1.00** | 0.91 | 0.54 | 0.65 | 0.81 | 0.85 | 0.93 | **1.00** |
| Neighbors | 2nd-Cos | **1.00** | **1.00** | **1.00** | 0.92 | 0.91 | 0.96 | 0.67 | 0.95 | **1.00** | **1.00** | **1.00** | **1.00** | **1.00** | 0.92 | 0.81 | 0.96 | 0.52 | 0.97 | **1.00** | **1.00** |
| | Jaccard | **1.00** | **1.00** | **1.00** | **1.00** | 0.94 | 0.96 | 0.96 | 0.96 | 0.98 | **1.00** | **1.00** | **1.00** | **1.00** | **1.00** | 0.95 | 0.97 | 0.95 | 0.97 | 0.99 | **1.00** |
| | RankSim | **1.00** | **1.00** | **1.00** | 0.99 | 0.94 | 0.95 | **0.98** | 0.99 | 0.98 | **1.00** | **1.00** | **1.00** | **1.00** | 0.99 | 0.95 | 0.97 | 0.97 | 0.99 | 0.99 | **1.00** |
| Topology | IMD | **1.00** | **1.00** | **1.00** | 0.95 | **1.00** | 0.94 | 0.97 | 0.85 | **1.00** | **1.00** | **1.00** | **1.00** | **1.00** | 0.89 | **1.00** | 0.82 | 0.97 | 0.55 | **1.00** | **1.00** |
| | RTD | **1.00** | **1.00** | **1.00** | 0.95 | 0.54 | 0.99 | 0.63 | **1.00** | **1.00** | **1.00** | **1.00** | **1.00** | **1.00** | 0.90 | 0.18 | 0.98 | 0.37 | **1.00** | 0.99 | **1.00** |
| Statistic | ConcDiff | 0.85 | 0.52 | 0.62 | 0.52 | 0.57 | 0.36 | 0.58 | 0.70 | 0.74 | 0.67 | 0.85 | 0.25 | 0.40 | 0.14 | 0.38 | -0.27 | 0.24 | 0.73 | 0.39 | 0.40 |
| | MagDiff | 0.63 | 0.72 | 0.92 | 0.97 | 0.65 | 0.63 | 0.58 | 0.52 | 0.71 | 0.87 | 0.55 | 0.49 | 0.89 | 0.91 | 0.58 | 0.50 | 0.38 | 0.06 | 0.29 | 0.63 |
| | UnifDiff | 0.70 | 0.70 | 0.69 | 0.69 | 0.94 | 0.92 | 0.94 | 0.60 | 0.41 | 0.83 | 0.67 | 0.67 | 0.59 | 0.56 | 0.82 | 0.89 | 0.82 | 0.33 | -0.45 | 0.68 |

## C.2 LANGUAGE RESULTS

For the language domain, we present the results in Tables 13-18. Any missing values for RTD and IMD shown as `nan` are due to excessive runtime. Missing values for other measures are due to numerical instability.

Table 13: Results of Test 1 (*Correlation to Accuracy Difference*) and Test 2 (*Correlation to Output Difference*) for the language domain using CLS token representations for BERT and ALBERT and final prompt token for SmolLM2.

| | Test Eval. Dataset Arch. | Acc. Corr. Spearman | | | | | | | | | JSD Corr. Spearman | | | | | | | | | Disagr. Corr. Spearman | | | | | | | | |
|---|---|---|---|---|---|---|---|---|---|---|---|---|---|---|---|---|---|---|---|---|---|---|---|---|---|---|---|
| | | | MNLI | | | SST2 | | | MNLI | | | SST2 | | | MNLI | | | SST2 | | | MNLI | | | SST2 | | |
| | | BERT | ALBERT | SmolLM2 | BERT | ALBERT | SmolLM2 | BERT | ALBERT | SmolLM2 | BERT | ALBERT | SmolLM2 | BERT | ALBERT | SmolLM2 | BERT | ALBERT | SmolLM2 | | | | | | | | |
| CCA | PWCCA | 0.01 | -0.03 | 0.04 | -0.33* | nan | nan | 0.22 | -0.17 | 0.53** | -0.32** | nan | nan | -0.37* | -0.36** | 0.32** | -0.22** | nan | nan | | | | | | | | |
| | SVCCA | 0.32* | -0.00 | 0.17 | -0.08 | 0.66** | 0.40** | 0.47** | 0.12 | 0.06 | 0.47** | 0.35* | 0.49** | 0.00 | 0.33* | 0.08 | 0.49** | 0.58** | 0.47** | | | | | | | | |
| Alignment | AlignCos | 0.25 | 0.00 | 0.19 | 0.02 | 0.13 | 0.44** | 0.37* | 0.09 | 0.66** | 0.49** | 0.77** | 0.39** | -0.16 | -0.03 | 0.49** | 0.28** | 0.51** | 0.37* | | | | | | | | |
| | AngShape | 0.28 | 0.12 | 0.14 | -0.14 | 0.34* | 0.46** | 0.26 | -0.08 | 0.39** | 0.40** | 0.40** | 0.36* | -0.02 | -0.01 | 0.23 | 0.48** | 0.51** | 0.24 | | | | | | | | |
| | HardCorr | 0.04 | 0.21 | 0.01 | -0.33* | 0.27 | 0.53** | -0.27 | -0.03 | 0.50** | -0.01 | 0.40** | 0.33* | -0.43** | 0.00 | 0.30* | 0.34** | 0.56** | 0.23 | | | | | | | | |
| | LinReg | 0.18 | 0.05 | -0.01 | -0.38** | 0.23* | 0.03 | 0.28** | 0.01 | 0.62** | 0.02 | 0.37** | 0.34** | -0.03 | -0.13 | 0.33** | 0.32** | 0.37** | 0.31** | | | | | | | | |
| | OrthProc | 0.28 | 0.12 | 0.14 | -0.14 | 0.34* | 0.46** | 0.26 | -0.08 | 0.39** | 0.40** | 0.40** | 0.36* | -0.02 | -0.01 | 0.23 | 0.48** | 0.51** | 0.24 | | | | | | | | |
| | PermProc | 0.09 | -0.02 | 0.15 | -0.09 | 0.01 | 0.47** | -0.06 | 0.04 | 0.45** | 0.04 | 0.79** | 0.16 | -0.30* | -0.04 | 0.31* | 0.18* | 0.44** | 0.42** | | | | | | | | |
| | ProcDist | 0.28 | -0.04 | 0.17 | 0.10 | 0.03 | 0.57** | 0.07 | -0.00 | 0.60** | 0.49** | 0.76** | 0.36* | -0.38* | -0.07 | 0.43** | 0.38** | 0.47** | 0.52** | | | | | | | | |
| | SoftCorr | 0.11 | 0.18 | 0.07 | -0.33* | 0.29 | 0.55** | -0.23 | -0.02 | 0.61** | -0.01 | 0.37* | 0.37* | -0.42** | 0.12 | 0.37* | 0.36** | 0.56** | 0.31* | | | | | | | | |
| RSM | CKA | 0.18 | 0.17 | 0.22 | -0.06 | 0.66** | 0.48** | 0.30* | 0.28 | 0.08 | 0.48** | 0.37* | 0.52** | -0.01 | 0.57** | 0.12 | 0.51** | 0.58** | 0.50** | | | | | | | | |
| | DistCorr | 0.15 | 0.25 | 0.18 | -0.10 | 0.56** | 0.49** | 0.39** | 0.32* | 0.13 | 0.51** | 0.51** | 0.57** | 0.12 | 0.57** | 0.17 | 0.53** | 0.58** | 0.51** | | | | | | | | |
| | EOS | 0.03 | -0.10 | -0.11 | -0.38** | -0.06 | -0.27 | 0.36* | -0.10 | -0.30 | -0.21** | 0.13 | 0.06 | 0.01 | -0.16 | -0.23 | -0.07 | -0.05 | 0.06 | | | | | | | | |
| | GULP | -0.01 | -0.12 | 0.01 | -0.36* | -0.17 | -0.10 | 0.35* | -0.10 | 0.25 | -0.30** | 0.25 | nan | 0.02 | -0.16 | 0.13 | -0.12 | -0.13 | nan | | | | | | | | |
| | RSA | 0.00 | 0.18 | 0.24 | -0.23 | 0.43** | 0.39** | 0.27 | 0.23 | -0.01 | 0.44** | 0.53** | 0.26 | 0.19 | 0.47** | 0.00 | 0.59** | 0.58** | 0.15 | | | | | | | | |
| | RSMDiff | 0.30* | -0.15 | 0.06 | 0.20 | -0.05 | 0.36* | -0.18 | -0.02 | 0.35* | 0.24** | 0.43** | -0.03 | -0.19 | 0.12 | 0.24 | -0.10 | 0.28 | 0.15 | | | | | | | | |
| Neighbors | 2nd-Cos | -0.26 | -0.25 | 0.03 | 0.30* | 0.24 | 0.10 | 0.16 | 0.04 | -0.07 | 0.49** | 0.30* | 0.18 | 0.55** | 0.12 | 0.01 | 0.15* | 0.16 | -0.20 | | | | | | | | |
| | Jaccard | -0.21 | -0.25 | -0.03 | 0.17 | 0.23 | 0.31* | 0.13 | 0.02 | 0.09 | 0.54** | 0.32* | 0.24 | 0.17 | -0.05 | -0.03 | 0.23** | 0.12 | 0.06 | | | | | | | | |
| | RankSim | -0.09 | -0.27 | -0.05 | 0.14 | 0.12 | 0.18 | 0.08 | -0.06 | 0.08 | 0.56** | 0.26 | 0.15 | 0.05 | -0.13 | -0.01 | 0.24** | 0.09 | 0.01 | | | | | | | | |
| Topology | IMD | -0.26 | -0.08 | -0.13 | -0.06 | 0.02 | 0.31* | -0.39** | -0.29* | 0.12 | 0.02 | 0.22 | 0.23 | -0.06 | -0.16 | 0.09 | 0.21** | 0.08 | 0.25 | | | | | | | | |
| | RTD | 0.09 | -0.27 | -0.02 | 0.05 | -0.10 | 0.32* | 0.04 | -0.34* | 0.33* | 0.33** | -0.08 | 0.27 | 0.10 | -0.18 | 0.12 | 0.13 | 0.03 | 0.06 | | | | | | | | |
| Statistic | ConcDiff | -0.00 | -0.07 | 0.01 | 0.12 | -0.20 | 0.06 | 0.02 | -0.07 | 0.46** | -0.02 | 0.59** | -0.11 | -0.07 | 0.03 | 0.30* | -0.26** | 0.29 | 0.06 | | | | | | | | |
| | MagDiff | 0.22 | -0.06 | -0.20 | -0.13 | 0.01 | -0.10 | 0.01 | 0.01 | -0.39** | 0.11 | 0.10 | -0.07 | -0.03 | 0.08 | -0.46** | 0.04 | 0.13 | -0.17 | | | | | | | | |
| | UnifDiff | 0.14 | -0.16 | -0.12 | 0.35* | 0.27 | -0.13 | -0.02 | -0.30* | -0.05 | 0.38** | 0.20 | 0.05 | -0.14 | -0.24 | 0.03 | -0.01 | 0.17 | -0.01 | | | | | | | | |

Table 14: Results of Test 1 (*Correlation to Accuracy Difference*) and Test 2 (*Correlation to Output Difference*) for the language domain using mean-pooled token representations.

| | Test Eval. Dataset Arch. | JSD Corr. Spearman | | | | Disagr. Corr. Spearman | | | |
|---|---|---|---|---|---|---|---|---|---|
| | | MNLI | | SST2 | | MNLI | | SST2 | |
| | | BERT | ALBERT | BERT | ALBERT | BERT | ALBERT | BERT | ALBERT |
| CCA | PWCCA | -0.11 | 0.56** | -0.42* | 0.35* | 0.01 | 0.53* | 0.10 | 0.19 |
| | SVCCA | 0.46** | 0.12 | 0.44** | 0.26 | 0.13 | 0.14 | 0.41** | 0.38* |
| Alignment | AlignCos | -0.00 | 0.30* | 0.48** | 0.53** | -0.12 | 0.25 | 0.29 | 0.33* |
| | AngShape | 0.27 | -0.09 | 0.34* | 0.22 | 0.36* | 0.06 | 0.41** | 0.22 |
| | HardCorr | -0.03 | -0.10 | -0.25 | 0.24 | 0.29* | -0.02 | 0.10 | 0.27 |
| | LinReg | -0.11 | 0.05 | -0.07 | -0.04 | 0.19 | 0.02 | 0.22* | 0.05 |
| | OrthProc | 0.27 | -0.09 | 0.34* | 0.22 | 0.36* | 0.06 | 0.41** | 0.22 |
| | PermProc | -0.05 | 0.07 | -0.16 | 0.53** | -0.28 | 0.16 | -0.04 | 0.24 |
| | ProcDist | -0.05 | 0.05 | 0.44** | 0.54** | -0.18 | 0.12 | 0.31* | 0.29* |
| | SoftCorr | 0.00 | -0.03 | -0.27 | 0.10 | 0.31* | 0.02 | 0.11 | 0.24 |
| RSM | CKA | 0.45** | 0.07 | 0.48** | 0.26 | 0.33* | 0.16 | 0.47** | 0.38* |
| | DistCorr | 0.49** | 0.07 | 0.54** | 0.33* | 0.38** | 0.18 | 0.52** | 0.41** |
| | EOS | 0.35* | 0.02 | -0.39* | 0.20 | 0.10 | 0.03 | -0.13 | -0.18 |
| | GULP | 0.34* | 0.02 | -0.46** | 0.36* | 0.09 | 0.03 | -0.24 | 0.06 |
| | RSA | 0.43** | -0.03 | 0.44** | 0.46** | 0.38* | 0.10 | 0.61** | 0.54** |
| | RSMDiff | -0.14 | -0.16 | 0.23 | 0.40** | -0.09 | -0.02 | 0.05 | 0.17 |
| Neighbors | 2nd-Cos | 0.34* | -0.04 | 0.58** | 0.28 | -0.04 | 0.05 | 0.21 | 0.06 |
| | Jaccard | 0.33* | -0.03 | 0.59** | 0.33* | -0.02 | 0.04 | 0.36* | 0.08 |
| | RankSim | 0.15 | -0.02 | 0.65** | 0.30* | -0.29 | 0.05 | 0.41** | 0.11 |
| Topology | IMD | 0.15 | 0.13 | 0.13 | 0.30* | -0.04 | 0.27 | 0.10 | 0.05 |
| | RTD | -0.08 | -0.18 | 0.16 | 0.21 | -0.01 | -0.02 | -0.06 | 0.11 |
| Statistic | ConcDiff | -0.03 | 0.24 | 0.03 | 0.43** | -0.14 | 0.25 | -0.14 | 0.19 |
| | MagDiff | -0.01 | -0.06 | 0.04 | -0.09 | 0.02 | 0.18 | -0.09 | -0.09 |
| | UnifDiff | -0.02 | -0.17 | 0.25 | -0.42** | 0.14 | -0.10 | -0.14 | -0.25 |

Table 15: Results of Test 3 (*Label Randomization*) for the language domain.

(a) Results of Test 3 (*Label Randomization*) using CLS/final token representations.

| | Eval. Dataset Arch. | AUPRC | | | | | | Conformity Rate | | | | | |
|---|---|---|---|---|---|---|---|---|---|---|---|---|---|
| | | | MNLI | | SST2 | | | | MNLI | | SST2 | | |
| | | BERT | ALBERT | SmolLM2 | BERT | ALBERT | SmolLM2 | BERT | ALBERT | SmolLM2 | BERT | ALBERT | SmolLM2 |
| CCA | PWCCA | 0.65 | 0.50 | 1.00 | 0.27 | nan | nan | 0.91 | 0.66 | 1.00 | 0.48 | nan | nan |
| | SVCCA | 0.69 | 0.68 | 1.00 | 0.64 | 0.69 | 0.80 | 0.86 | 0.88 | 1.00 | 0.80 | 0.82 | 0.89 |
| Alignment | AlignCos | 1.00 | 0.68 | 1.00 | 0.99 | 0.70 | 1.00 | 1.00 | 0.91 | 1.00 | 1.00 | 0.83 | 1.00 |
| | AngShape | 0.90 | 0.97 | 1.00 | 0.49 | 0.67 | 0.98 | 0.99 | 0.99 | 1.00 | 0.75 | 0.86 | 1.00 |
| | HardCorr | 0.75 | 0.68 | 1.00 | 0.44 | 0.67 | 1.00 | 0.90 | 0.86 | 1.00 | 0.77 | 0.86 | 1.00 |
| | LinReg | 0.57 | 0.47 | 1.00 | 0.40 | 0.21 | 0.54 | 0.85 | 0.73 | 1.00 | 0.57 | 0.50 | 0.73 |
| | OrthProc | 0.90 | 0.97 | 1.00 | 0.49 | 0.70 | 0.98 | 0.99 | 1.00 | 1.00 | 0.75 | 0.86 | 1.00 |
| | PermProc | 0.44 | 0.60 | 1.00 | 0.45 | 0.66 | 1.00 | 0.68 | 0.81 | 1.00 | 0.73 | 0.87 | 1.00 |
| | ProcDist | 0.98 | 0.84 | 1.00 | 0.86 | 0.70 | 1.00 | 0.99 | 0.96 | 1.00 | 0.95 | 0.93 | 1.00 |
| | SoftCorr | 0.75 | 0.71 | 1.00 | 0.48 | 0.72 | 1.00 | 0.92 | 0.87 | 1.00 | 0.80 | 0.88 | 1.00 |
| RSM | CKA | 0.75 | 0.84 | 1.00 | 0.59 | 0.70 | 0.99 | 0.90 | 0.93 | 1.00 | 0.78 | 0.84 | 0.99 |
| | DistCorr | 0.75 | 0.79 | 1.00 | 0.59 | 0.70 | 0.99 | 0.89 | 0.93 | 1.00 | 0.60 | 0.85 | 0.99 |
| | EOS | 0.62 | 0.70 | 1.00 | 0.36 | 0.39 | 0.59 | 0.88 | 0.88 | 1.00 | 0.60 | 0.52 | 0.50 |
| | GULP | 0.61 | 0.41 | 1.00 | 0.28 | 0.46 | nan | 0.89 | 0.77 | 1.00 | 0.54 | 0.68 | nan |
| | RSA | 0.46 | 0.62 | 1.00 | 0.48 | 0.72 | 1.00 | 0.64 | 0.79 | 1.00 | 0.73 | 0.87 | 1.00 |
| | RSMDiff | 0.52 | 1.00 | 0.91 | 0.91 | 1.00 | 1.00 | 0.67 | 1.00 | 0.93 | 0.97 | 1.00 | 1.00 |
| Neighbors | 2nd-Cos | nan | nan | 1.00 | 0.37 | 0.40 | 1.00 | nan | nan | 1.00 | 0.64 | 0.61 | 1.00 |
| | Jaccard | 0.69 | 0.56 | 1.00 | 0.35 | 0.55 | 1.00 | 0.84 | 0.70 | 1.00 | 0.69 | 0.84 | 1.00 |
| | RankSim | 0.63 | 0.47 | 1.00 | 0.34 | 0.54 | 1.00 | 0.82 | 0.62 | 1.00 | 0.63 | 0.82 | 1.00 |
| Topology | IMD | 0.74 | 0.46 | 1.00 | 0.47 | 0.31 | 0.58 | 0.92 | 0.74 | 1.00 | 0.79 | 0.59 | 0.70 |
| | RTD | 0.74 | 0.54 | 1.00 | 0.27 | 0.42 | 0.96 | 0.79 | 0.78 | 1.00 | 0.61 | 0.67 | 0.97 |
| Statistic | ConcDiff | 1.00 | 0.81 | 1.00 | 0.96 | 0.61 | 1.00 | 1.00 | 0.89 | 1.00 | 0.99 | 0.80 | 1.00 |
| | MagDiff | 0.33 | 0.56 | 1.00 | 0.38 | 0.70 | 1.00 | 0.75 | 0.76 | 1.00 | 0.69 | 0.88 | 1.00 |
| | UnifDiff | 0.76 | 0.75 | 0.84 | 0.60 | 0.77 | 1.00 | 0.91 | 0.90 | 0.83 | 0.86 | 0.91 | 1.00 |

(b) Results of Test 3 (*Label Randomization*) using mean-pooled token representations.

| | Eval. Dataset Arch. | AUPRC | | | | Conformity Rate | | | |
|---|---|---|---|---|---|---|---|---|---|
| | | MNLI | | SST2 | | MNLI | | SST2 | |
| | | BERT | ALBERT | BERT | ALBERT | BERT | ALBERT | BERT | ALBERT |
| CCA | PWCCA | 0.35 | 0.58 | nan | nan | 0.67 | 0.76 | nan | nan |
| | SVCCA | 0.66 | 0.46 | 0.61 | 0.73 | 0.89 | 0.77 | 0.77 | 0.86 |
| Alignment | AlignCos | 0.80 | 0.65 | 0.92 | 0.68 | 0.94 | 0.80 | 0.99 | 0.83 |
| | AngShape | 0.71 | 0.64 | 0.42 | 0.56 | 0.94 | 0.82 | 0.70 | 0.82 |
| | HardCorr | 0.53 | 0.55 | 0.37 | 0.56 | 0.82 | 0.81 | 0.71 | 0.85 |
| | LinReg | 0.38 | 0.39 | 0.40 | 0.25 | 0.72 | 0.69 | 0.56 | 0.52 |
| | OrthProc | 0.71 | 0.64 | 0.42 | 0.58 | 0.94 | 0.82 | 0.70 | 0.82 |
| | PermProc | 0.40 | 0.57 | 0.39 | 0.60 | 0.61 | 0.79 | 0.67 | 0.74 |
| | ProcDist | 0.69 | 0.62 | 0.79 | 0.63 | 0.87 | 0.79 | 0.95 | 0.78 |
| | SoftCorr | 0.68 | 0.64 | 0.37 | 0.53 | 0.86 | 0.89 | 0.72 | 0.87 |
| RSM | CKA | 0.64 | 0.43 | 0.56 | 0.72 | 0.84 | 0.68 | 0.74 | 0.86 |
| | DistCorr | 0.66 | 0.50 | 0.56 | 0.70 | 0.86 | 0.72 | 0.75 | 0.86 |
| | EOS | 0.57 | 0.76 | 0.26 | 0.29 | 0.86 | 0.90 | 0.57 | 0.52 |
| | GULP | 0.49 | 0.60 | 0.30 | 0.48 | 0.82 | 0.81 | 0.60 | 0.76 |
| | RSA | 0.48 | 0.44 | 0.45 | 0.60 | 0.71 | 0.68 | 0.65 | 0.86 |
| | RSMDiff | 0.86 | 0.59 | 0.93 | 0.74 | 0.94 | 0.83 | 0.99 | 0.89 |
| Neighbors | 2nd-Cos | nan | nan | 0.31 | 0.36 | nan | nan | 0.59 | 0.61 |
| | Jaccard | 0.66 | 0.52 | 0.31 | 0.44 | 0.84 | 0.72 | 0.73 | 0.78 |
| | RankSim | 0.58 | 0.36 | 0.29 | 0.42 | 0.79 | 0.63 | 0.70 | 0.76 |
| Topology | IMD | 0.58 | 0.41 | 0.45 | 0.21 | 0.86 | 0.72 | 0.75 | 0.51 |
| | RTD | 0.59 | 0.59 | 0.39 | 0.36 | 0.79 | 0.83 | 0.64 | 0.61 |
| Statistic | ConcDiff | 0.76 | 0.52 | 0.92 | 0.66 | 0.90 | 0.78 | 0.97 | 0.81 |
| | MagDiff | 0.39 | 0.48 | 0.65 | 0.48 | 0.78 | 0.81 | 0.86 | 0.82 |
| | UnifDiff | 0.88 | 0.65 | 0.57 | 0.76 | 0.94 | 0.88 | 0.85 | 0.92 |

Table 16: Results of Test 4 (*Shortcut Affinity*) for the language domain.

(a) Results of Test 4 (*Shortcut Affinity*) using CLS/final token representations.

| Eval. Dataset / Arch. | | AUPRC MNLI BERT | ALBERT | SmolLM2 | SST2 BERT | ALBERT | SmolLM2 | Conformity Rate MNLI BERT | ALBERT | SmolLM2 | SST2 BERT | ALBERT | SmolLM2 |
|---|---|---|---|---|---|---|---|---|---|---|---|---|---|
| CCA | PWCCA | 0.56 | nan | 0.75 | 0.32 | nan | nan | 0.86 | nan | 0.94 | 0.67 | nan | nan |
| | SVCCA | 0.42 | 0.60 | 0.66 | 0.36 | 0.49 | 0.88 | 0.78 | 0.62 | 0.85 | 0.66 | 0.77 | 0.98 |
| Alignment | AlignCos | 0.58 | 0.37 | 1.00 | 0.54 | 0.71 | 0.61 | 0.89 | 0.58 | 1.00 | 0.82 | 0.88 | 0.87 |
| | AngShape | 0.60 | 0.48 | 1.00 | 0.43 | 0.72 | 0.66 | 0.90 | 0.64 | 1.00 | 0.79 | 0.90 | 0.90 |
| | HardCorr | 0.55 | 0.36 | 0.99 | 0.36 | 0.64 | 0.64 | 0.82 | 0.60 | 1.00 | 0.68 | 0.88 | 0.88 |
| | LinReg | 0.46 | 0.50 | 0.71 | 0.46 | 0.66 | 0.59 | 0.85 | 0.52 | 0.91 | 0.77 | 0.86 | 0.78 |
| | OrthProc | 0.60 | 0.48 | 1.00 | 0.43 | 0.72 | 0.66 | 0.90 | 0.64 | 1.00 | 0.79 | 0.90 | 0.90 |
| | PermProc | 0.52 | 0.54 | 0.95 | 0.55 | 0.70 | 0.60 | 0.82 | 0.63 | 0.97 | 0.80 | 0.88 | 0.86 |
| | ProcDist | 0.54 | 0.54 | 1.00 | 0.52 | 0.65 | 0.63 | 0.87 | 0.62 | 1.00 | 0.79 | 0.87 | 0.86 |
| | SoftCorr | 0.55 | 0.36 | 1.00 | 0.34 | 0.62 | 0.63 | 0.82 | 0.60 | 1.00 | 0.60 | 0.86 | 0.88 |
| RSM | CKA | 0.59 | 0.63 | 0.94 | 0.38 | 0.58 | 0.87 | 0.88 | 0.64 | 0.99 | 0.70 | 0.83 | 0.97 |
| | DistCorr | 0.58 | 0.62 | 0.97 | 0.39 | 0.68 | 0.87 | 0.88 | 0.67 | 0.99 | 0.71 | 0.87 | 0.97 |
| | EOS | 0.57 | 0.46 | 0.54 | 0.33 | 0.40 | 0.22 | 0.85 | 0.58 | 0.82 | 0.62 | 0.62 | 0.48 |
| | GULP | 0.56 | 0.46 | 0.78 | 0.30 | 0.68 | nan | 0.84 | 0.58 | 0.94 | 0.67 | 0.88 | 0.84 |
| | RSA | 0.58 | 0.59 | 0.96 | 0.47 | 0.65 | 0.74 | 0.87 | 0.63 | 0.99 | 0.72 | 0.88 | 0.94 |
| | RSMDiff | 0.28 | 0.57 | 0.68 | 0.37 | 0.63 | 0.52 | 0.66 | 0.60 | 0.86 | 0.69 | 0.85 | 0.81 |
| Neighbors | 2nd-Cos | 0.59 | 0.56 | 1.00 | 0.64 | 0.68 | 0.70 | 0.82 | 0.60 | 1.00 | 0.86 | 0.85 | 0.90 |
| | Jaccard | 0.56 | 0.58 | 0.98 | 0.64 | 0.75 | 0.74 | 0.81 | 0.63 | 1.00 | 0.87 | 0.89 | 0.93 |
| | RankSim | 0.58 | 0.61 | 0.98 | 0.64 | 0.75 | 0.71 | 0.82 | 0.66 | 1.00 | 0.88 | 0.89 | 0.92 |
| Topology | IMD | 0.53 | 0.28 | 0.57 | 0.34 | 0.23 | 0.20 | 0.82 | 0.43 | 0.76 | 0.71 | 0.56 | 0.52 |
| | RTD | 0.61 | 0.41 | 0.80 | 0.39 | 0.39 | 0.62 | 0.85 | 0.57 | 0.91 | 0.73 | 0.72 | 0.83 |
| Statistic | ConcDiff | 0.38 | 0.42 | 0.57 | 0.27 | 0.65 | 0.49 | 0.75 | 0.45 | 0.86 | 0.67 | 0.87 | 0.76 |
| | MagDiff | 0.48 | 0.56 | 0.58 | 0.35 | 0.55 | 0.60 | 0.81 | 0.77 | 0.84 | 0.73 | 0.83 | 0.84 |
| | UnifDiff | 0.40 | 0.34 | 0.68 | 0.37 | 0.26 | 0.30 | 0.69 | 0.52 | 0.84 | 0.56 | 0.53 | 0.61 |

(b) Results of Test 4 (*Shortcut Affinity*) using mean-pooled token representations.

| Eval. Dataset / Arch. | | AUPRC MNLI BERT | ALBERT | SST2 BERT | ALBERT | Conformity Rate MNLI BERT | ALBERT | SST2 BERT | ALBERT |
|---|---|---|---|---|---|---|---|---|---|
| CCA | PWCCA | 0.83 | nan | 0.36 | 0.45 | 0.96 | nan | 0.72 | 0.66 |
| | SVCCA | 0.49 | 0.52 | 0.25 | 0.64 | 0.80 | 0.61 | 0.66 | 0.89 |
| Alignment | AlignCos | 0.53 | 0.45 | 0.51 | 0.61 | 0.83 | 0.58 | 0.81 | 0.81 |
| | AngShape | 0.57 | 0.49 | 0.38 | 0.63 | 0.87 | 0.64 | 0.79 | 0.89 |
| | HardCorr | 0.32 | 0.41 | 0.22 | 0.62 | 0.75 | 0.61 | 0.69 | 0.89 |
| | LinReg | 0.23 | 0.44 | 0.29 | 0.58 | 0.60 | 0.59 | 0.75 | 0.84 |
| | OrthProc | 0.57 | 0.49 | 0.38 | 0.63 | 0.87 | 0.64 | 0.79 | 0.89 |
| | PermProc | 0.35 | 0.54 | 0.38 | 0.59 | 0.60 | 0.58 | 0.74 | 0.83 |
| | ProcDist | 0.53 | 0.51 | 0.51 | 0.61 | 0.84 | 0.57 | 0.81 | 0.81 |
| | SoftCorr | 0.33 | 0.41 | 0.21 | 0.61 | 0.75 | 0.62 | 0.51 | 0.89 |
| RSM | CKA | 0.53 | 0.50 | 0.27 | 0.63 | 0.84 | 0.61 | 0.69 | 0.89 |
| | DistCorr | 0.54 | 0.54 | 0.29 | 0.65 | 0.85 | 0.63 | 0.74 | 0.89 |
| | EOS | 0.39 | 0.55 | 0.21 | 0.70 | 0.67 | 0.63 | 0.57 | 0.85 |
| | GULP | 0.44 | 0.55 | 0.22 | 0.70 | 0.74 | 0.64 | 0.56 | 0.84 |
| | RSA | 0.47 | 0.52 | 0.38 | 0.59 | 0.80 | 0.61 | 0.72 | 0.87 |
| | RSMDiff | 0.36 | 0.44 | 0.49 | 0.59 | 0.68 | 0.61 | 0.79 | 0.78 |
| Neighbors | 2nd-Cos | 0.58 | 0.51 | 0.52 | 0.60 | 0.83 | 0.64 | 0.81 | 0.81 |
| | Jaccard | 0.56 | 0.59 | 0.54 | 0.64 | 0.82 | 0.70 | 0.85 | 0.87 |
| | RankSim | 0.50 | 0.45 | 0.52 | 0.59 | 0.73 | 0.61 | 0.83 | 0.82 |
| Topology | IMD | 0.52 | 0.29 | 0.42 | 0.56 | 0.81 | 0.48 | 0.77 | 0.79 |
| | RTD | 0.65 | 0.42 | 0.38 | 0.66 | 0.85 | 0.53 | 0.76 | 0.88 |
| Statistic | ConcDiff | 0.35 | 0.40 | 0.50 | 0.56 | 0.65 | 0.48 | 0.80 | 0.77 |
| | MagDiff | 0.30 | 0.30 | 0.21 | 0.55 | 0.58 | 0.47 | 0.58 | 0.84 |
| | UnifDiff | 0.33 | 0.44 | 0.37 | 0.23 | 0.68 | 0.52 | 0.57 | 0.50 |

Table 17: Results of Test 5 (*Augmentation*) for the language domain.

(a) Results of Test 5 (*Augmentation*) using CLS token representations.

| Eval. Dataset / Arch. | | AUPRC MNLI BERT | ALBERT | SST2 BERT | ALBERT | Conformity Rate MNLI BERT | ALBERT | SST2 BERT | ALBERT |
|---|---|---|---|---|---|---|---|---|---|
| CCA | PWCCA | 0.18 | 0.58 | 0.35 | nan | 0.49 | 0.78 | 0.38 | nan |
| | SVCCA | 0.44 | 0.60 | 0.61 | 0.39 | 0.77 | 0.66 | 0.68 | 0.48 |
| Alignment | AlignCos | 0.35 | 0.77 | 0.45 | 0.33 | 0.80 | 0.80 | 0.63 | 0.47 |
| | AngShape | 0.38 | 0.91 | 0.52 | 0.36 | 0.84 | 0.94 | 0.62 | 0.46 |
| | HardCorr | 0.24 | 0.65 | 0.34 | 0.32 | 0.67 | 0.76 | 0.49 | 0.49 |
| | LinReg | 0.24 | 0.73 | 0.40 | 0.42 | 0.66 | 0.86 | 0.55 | 0.59 |
| | OrthProc | 0.38 | 0.91 | 0.52 | 0.36 | 0.84 | 0.94 | 0.62 | 0.46 |
| | PermProc | 0.18 | 0.49 | 0.31 | 0.31 | 0.49 | 0.55 | 0.44 | 0.51 |
| | ProcDist | 0.32 | 0.74 | 0.43 | 0.32 | 0.73 | 0.85 | 0.64 | 0.47 |
| | SoftCorr | 0.28 | 0.62 | 0.41 | 0.33 | 0.72 | 0.75 | 0.52 | 0.50 |
| RSM | CKA | 0.48 | 0.83 | 0.61 | 0.40 | 0.87 | 0.82 | 0.68 | 0.47 |
| | DistCorr | 0.45 | 0.86 | 0.62 | 0.40 | 0.85 | 0.87 | 0.69 | 0.46 |
| | EOS | 0.27 | 0.78 | 0.30 | 0.32 | 0.71 | 0.80 | 0.49 | 0.46 |
| | GULP | 0.26 | 0.75 | 0.33 | 0.45 | 0.69 | 0.80 | 0.47 | 0.52 |
| | RSA | 0.49 | 0.86 | 0.61 | 0.38 | 0.86 | 0.84 | 0.68 | 0.47 |
| | RSMDiff | 0.35 | 0.85 | 0.34 | 0.30 | 0.66 | 0.84 | 0.58 | 0.43 |
| Neighbors | 2nd-Cos | 0.44 | 0.70 | 0.40 | 0.36 | 0.64 | 0.74 | 0.64 | 0.49 |
| | Jaccard | 0.35 | 0.78 | 0.39 | 0.34 | 0.74 | 0.80 | 0.61 | 0.50 |
| | RankSim | 0.33 | 0.77 | 0.36 | 0.38 | 0.71 | 0.79 | 0.57 | 0.52 |
| Topology | IMD | 0.43 | 0.38 | 0.30 | 0.36 | 0.75 | 0.50 | 0.47 | 0.49 |
| | RTD | 0.54 | 0.72 | 0.38 | 0.28 | 0.83 | 0.76 | 0.54 | 0.44 |
| Statistic | ConcDiff | 0.28 | 0.55 | 0.32 | 0.34 | 0.61 | 0.67 | 0.51 | 0.43 |
| | MagDiff | 0.16 | 0.60 | 0.31 | 0.34 | 0.45 | 0.79 | 0.46 | 0.52 |
| | UnifDiff | 0.61 | 0.44 | 0.33 | 0.29 | 0.84 | 0.60 | 0.46 | 0.40 |

(b) Results of Test 5 (*Augmentation*) using mean-pooled token representations.

| Eval. Dataset / Arch. | | AUPRC MNLI BERT | ALBERT | SST2 BERT | ALBERT | Conformity Rate MNLI BERT | ALBERT | SST2 BERT | ALBERT |
|---|---|---|---|---|---|---|---|---|---|
| CCA | PWCCA | 0.22 | nan | 0.32 | 0.71 | 0.54 | nan | 0.49 | 0.73 |
| | SVCCA | 0.31 | 0.63 | 0.61 | 0.29 | 0.70 | 0.71 | 0.66 | 0.44 |
| Alignment | AlignCos | 0.28 | 0.55 | 0.34 | 0.33 | 0.63 | 0.68 | 0.56 | 0.48 |
| | AngShape | 0.29 | 0.81 | 0.42 | 0.28 | 0.75 | 0.82 | 0.56 | 0.46 |
| | HardCorr | 0.21 | 0.63 | 0.28 | 0.31 | 0.57 | 0.84 | 0.45 | 0.47 |
| | LinReg | 0.27 | 0.83 | 0.36 | 0.43 | 0.63 | 0.85 | 0.51 | 0.64 |
| | OrthProc | 0.29 | 0.81 | 0.42 | 0.28 | 0.75 | 0.82 | 0.56 | 0.46 |
| | PermProc | 0.17 | 0.51 | 0.27 | 0.33 | 0.44 | 0.59 | 0.42 | 0.48 |
| | ProcDist | 0.25 | 0.49 | 0.33 | 0.32 | 0.60 | 0.59 | 0.58 | 0.47 |
| | SoftCorr | 0.21 | 0.59 | 0.28 | 0.32 | 0.57 | 0.81 | 0.46 | 0.51 |
| RSM | CKA | 0.34 | 0.72 | 0.60 | 0.29 | 0.76 | 0.77 | 0.66 | 0.46 |
| | DistCorr | 0.35 | 0.73 | 0.60 | 0.33 | 0.78 | 0.77 | 0.66 | 0.46 |
| | EOS | 0.24 | 0.73 | 0.26 | 0.30 | 0.64 | 0.77 | 0.40 | 0.50 |
| | GULP | 0.23 | 0.73 | 0.26 | 0.32 | 0.62 | 0.77 | 0.41 | 0.54 |
| | RSA | 0.34 | 0.75 | 0.54 | 0.33 | 0.76 | 0.78 | 0.64 | 0.49 |
| | RSMDiff | 0.24 | 0.42 | 0.29 | 0.31 | 0.58 | 0.52 | 0.48 | 0.44 |
| Neighbors | 2nd-Cos | 0.38 | 0.58 | 0.39 | 0.32 | 0.66 | 0.69 | 0.61 | 0.50 |
| | Jaccard | 0.32 | 0.63 | 0.34 | 0.34 | 0.71 | 0.73 | 0.59 | 0.47 |
| | RankSim | 0.35 | 0.50 | 0.32 | 0.33 | 0.60 | 0.66 | 0.55 | 0.47 |
| Topology | IMD | 0.20 | 0.30 | 0.33 | 0.29 | 0.51 | 0.46 | 0.45 | 0.44 |
| | RTD | 0.29 | 0.39 | 0.29 | 0.29 | 0.66 | 0.53 | 0.46 | 0.44 |
| Statistic | ConcDiff | 0.27 | 0.44 | 0.30 | 0.27 | 0.65 | 0.52 | 0.48 | 0.42 |
| | MagDiff | 0.28 | 0.68 | 0.32 | 0.28 | 0.65 | 0.86 | 0.43 | 0.44 |
| | UnifDiff | 0.42 | 0.33 | 0.33 | 0.30 | 0.75 | 0.50 | 0.47 | 0.48 |

Table 18: Results of Test 6 (*Layer Monotonicity*) for the language domain.

(a) Results of Test 6 (*Layer Monotonicity*) using CLS/final token representations.

| Eval. Dataset / Arch. | | Conformity Rate MNLI BERT | ALBERT | SmolLM2 | SST2 BERT | ALBERT | SmolLM2 | Spearman MNLI BERT | ALBERT | SmolLM2 | SST2 BERT | ALBERT | SmolLM2 |
|---|---|---|---|---|---|---|---|---|---|---|---|---|---|
| CCA | PWCCA | 1.00 | 1.00 | 0.95 | 1.00 | 1.00 | 1.00 | 1.00 | 1.00 | 0.93 | 1.00 | nan | nan |
| | SVCCA | 0.91 | 0.85 | 0.85 | 0.88 | 0.92 | 0.87 | 0.91 | 0.78 | 0.72 | 0.64 | 0.84 | 0.69 |
| Alignment | AlignCos | 1.00 | 0.95 | 0.86 | 0.99 | 0.98 | 0.92 | 1.00 | 0.95 | 0.81 | 0.99 | 0.99 | 0.89 |
| | AngShape | 0.99 | 0.99 | 0.98 | 0.98 | 0.99 | 0.95 | 1.00 | 0.99 | 0.99 | 1.00 | 1.00 | 0.97 |
| | HardCorr | 1.00 | 0.99 | 0.97 | 0.99 | 0.99 | 0.98 | 1.00 | 1.00 | 0.98 | 0.99 | 0.99 | 0.99 |
| | LinReg | 0.83 | 0.88 | 0.96 | 0.93 | 0.96 | 0.74 | 0.66 | 0.76 | 0.91 | 0.90 | 0.95 | 0.27 |
| | OrthProc | 0.98 | 0.97 | 0.97 | 0.98 | 0.97 | 0.95 | 0.99 | 0.99 | 0.98 | 0.99 | 0.99 | 0.97 |
| | PermProc | 0.80 | 0.95 | 0.89 | 0.83 | 0.98 | 0.89 | 0.73 | 0.96 | 0.85 | 0.75 | 0.98 | 0.85 |
| | ProcDist | 0.96 | 0.98 | 0.90 | 0.97 | 0.94 | 0.98 | 0.99 | 0.99 | 0.93 | 0.95 | 0.98 | 0.98 |
| | SoftCorr | 0.98 | 0.98 | 0.94 | 0.96 | 0.97 | 0.98 | 0.99 | 0.99 | 0.93 | 0.95 | 0.98 | 0.98 |
| RSM | CKA | 0.98 | 0.98 | 0.95 | 0.98 | 0.96 | 0.93 | 0.99 | 0.99 | 0.94 | 0.96 | 0.98 | 0.91 |
| | DistCorr | 0.96 | 0.99 | 0.96 | 0.97 | 0.98 | 0.96 | 0.97 | 0.99 | 0.94 | 0.98 | 0.99 | 0.91 |
| | EOS | 0.98 | 0.96 | 0.93 | 0.91 | 0.88 | 0.59 | 0.99 | 0.96 | 0.88 | 0.92 | 0.86 | 0.16 |
| | GULP | 0.84 | 0.83 | 0.86 | 0.78 | 0.76 | 1.00 | 0.51 | 0.49 | 0.63 | 0.45 | 0.44 | nan |
| | RSA | 0.99 | 0.94 | 0.94 | 0.95 | 0.97 | 0.96 | 1.00 | 0.96 | 0.90 | 0.96 | 0.97 | 0.97 |
| | RSMDiff | 0.94 | 0.90 | 0.91 | 0.95 | 0.95 | 0.91 | 0.84 | 0.89 | 0.86 | 0.84 | 0.94 | 0.85 |
| Neighbors | 2nd-Cos | 0.94 | 0.93 | 0.98 | 0.96 | 0.97 | 0.93 | 0.94 | 0.92 | 0.98 | 0.97 | 0.96 | 0.92 |
| | Jaccard | 0.95 | 0.94 | 0.96 | 0.94 | 0.93 | 0.95 | 0.96 | 0.95 | 0.98 | 0.96 | 0.95 | 0.97 |
| | RankSim | 0.93 | 0.94 | 0.93 | 0.94 | 0.94 | 0.95 | 0.95 | 0.90 | 0.94 | 0.87 | 0.96 | 0.97 |
| Topology | IMD | 0.68 | 0.84 | nan | 0.64 | nan | 0.61 | 0.44 | 0.79 | nan | 0.54 | 0.43 | nan |
| | RTD | 0.76 | 0.80 | 0.73 | 0.73 | 0.66 | 0.71 | 0.74 | 0.42 | 0.43 | 0.48 | 0.39 | 0.42 |
| Statistic | ConcDiff | 0.99 | 0.90 | 0.73 | 0.96 | 0.96 | 0.93 | 0.99 | 0.87 | 0.55 | 0.96 | 0.90 | 0.71 |
| | MagDiff | 0.64 | 0.93 | 0.89 | 0.62 | 0.92 | 0.89 | 0.52 | 0.90 | 0.86 | 0.48 | 0.84 | 0.86 |
| | UnifDiff | 0.94 | 0.91 | 0.97 | 0.85 | 0.76 | 0.98 | 0.81 | 0.87 | 0.82 | 0.77 | 0.48 | 0.78 |

(b) Results of Test 6 (*Layer Monotonicity*) using mean-pooled token representations.

| Eval. Dataset / Arch. | | Conformity Rate MNLI BERT | ALBERT | SST2 BERT | ALBERT | Spearman MNLI BERT | ALBERT | SST2 BERT | ALBERT |
|---|---|---|---|---|---|---|---|---|---|
| CCA | PWCCA | 1.00 | 1.00 | 1.00 | 1.00 | 1.00 | 1.00 | nan | nan |
| | SVCCA | 0.85 | 0.87 | 0.95 | 0.98 | 0.78 | 0.75 | 0.83 | 0.93 |
| Alignment | AlignCos | 1.00 | 0.99 | 1.00 | 1.00 | 1.00 | 1.00 | 1.00 | 1.00 |
| | AngShape | 1.00 | 1.00 | 1.00 | 1.00 | 1.00 | 1.00 | 1.00 | 1.00 |
| | HardCorr | 1.00 | 1.00 | 1.00 | 1.00 | 1.00 | 1.00 | 1.00 | 1.00 |
| | LinReg | 0.86 | 0.75 | 0.97 | 0.85 | 0.73 | 0.42 | 0.98 | 0.57 |
| | OrthProc | 1.00 | 1.00 | 1.00 | 1.00 | 1.00 | 1.00 | 1.00 | 1.00 |
| | PermProc | 0.90 | 0.93 | 0.94 | 1.00 | 0.90 | 0.95 | 0.94 | 1.00 |
| | ProcDist | 0.99 | 0.99 | 1.00 | 1.00 | 0.99 | 0.99 | 1.00 | 1.00 |
| | SoftCorr | 0.99 | 0.99 | 1.00 | 1.00 | 1.00 | 1.00 | 1.00 | 1.00 |
| RSM | CKA | 0.99 | 0.99 | 1.00 | 1.00 | 1.00 | 0.99 | 1.00 | 1.00 |
| | DistCorr | 0.99 | 0.99 | 1.00 | 1.00 | 1.00 | 0.99 | 1.00 | 1.00 |
| | EOS | 1.00 | 1.00 | 0.98 | 1.00 | 1.00 | 1.00 | 0.99 | 1.00 |
| | GULP | 1.00 | 0.93 | 0.97 | 0.93 | 1.00 | 0.89 | 0.98 | 0.91 |
| | RSA | 1.00 | 1.00 | 1.00 | 1.00 | 1.00 | 1.00 | 1.00 | 1.00 |
| | RSMDiff | 0.91 | 0.94 | 0.95 | 0.99 | 0.85 | 0.88 | 0.89 | 0.98 |
| Neighbors | 2nd-Cos | 1.00 | 1.00 | 1.00 | 1.00 | 1.00 | 1.00 | 1.00 | 1.00 |
| | Jaccard | 1.00 | 1.00 | 1.00 | 1.00 | 1.00 | 1.00 | 1.00 | 1.00 |
| | RankSim | 0.79 | 0.87 | 1.00 | 1.00 | 0.52 | 1.00 | 1.00 | 1.00 |
| Topology | IMD | 0.74 | 0.89 | 0.85 | 0.90 | 0.81 | 0.71 | 0.74 | 0.65 |
| | RTD | 0.93 | 0.88 | 0.95 | 0.91 | 0.95 | 0.89 | 0.94 | 0.89 |
| Statistic | ConcDiff | 0.62 | 0.62 | 0.64 | 0.98 | 0.50 | 0.34 | 0.53 | 0.96 |
| | MagDiff | 0.81 | 0.88 | 0.82 | 0.92 | 0.66 | 0.60 | 0.67 | 0.77 |
| | UnifDiff | 0.95 | 0.98 | 0.86 | 0.75 | 0.83 | 0.91 | 0.70 | 0.43 |

## C.3 VISION RESULTS

We present the results of the vision domain in Tables 19-30. Results are presented in test order, with each test featuring one table. Some entries in the result tables feature `NaN` values. This was caused by various reasons: (i) Numerical instability due to operations like Singular Value Decompositions or negative values in square roots and (ii) identical similarity values, leading to failure of correlation values. Whenever such failure occurred for an entire group of models, the entire measure was excluded for this case, as, for instance, removing the model group of Gaussian noise **M** may simplify the overall task of distinguishing models.

Table 19: Results of Test 1 (*Correlation to Accuracy Difference*) for the vision domain on ImageNet-100.

| | Eval. Dataset Arch. | Spearman IN100 | | | | | | |
| --- | --- | --- | --- | --- | --- | --- | --- | --- |
| | | RNet18 | RNet34 | RNet101 | VGG11 | VGG19 | ViT B32 | ViT L32 |
| CCA | PWCCA | -0.02 | -0.20 | -0.09 | -0.18 | **0.11** | -0.08 | 0.09 |
| | SVCCA | 0.29* | 0.27 | -0.00 | -0.04 | -0.30* | -0.01 | -0.17 |
| Alignment | AlignCos | -0.08 | -0.35* | -0.01 | -0.13 | -0.12 | 0.07 | 0.05 |
| | AngShape | 0.21 | -0.16 | 0.15 | -0.02 | 0.03 | 0.07 | 0.06 |
| | HardCorr | 0.21 | 0.13 | -0.01 | -0.01 | -0.03 | 0.35* | -0.17 |
| | LinReg | 0.19 | -0.11 | 0.09 | -0.04 | 0.09 | 0.15 | 0.05 |
| | OrthProc | 0.21 | -0.16 | 0.15 | -0.02 | 0.03 | 0.07 | 0.06 |
| | PermProc | 0.07 | 0.09 | 0.08 | 0.14 | -0.02 | -0.06 | -0.33* |
| | ProcDist | 0.08 | 0.00 | 0.14 | 0.13 | 0.08 | 0.16 | 0.05 |
| | SoftCorr | 0.27 | 0.08 | 0.04 | -0.03 | -0.10 | 0.36* | -0.19 |
| RSM | CKA | **0.36*** | -0.07 | **0.16** | 0.03 | -0.20 | -0.26 | 0.05 |
| | DistCorr | 0.31* | -0.08 | 0.08 | 0.03 | -0.21 | -0.26 | 0.03 |
| | EOS | 0.05 | -0.17 | 0.11 | -0.22 | 0.08 | **0.47**** | 0.03 |
| | GULP | 0.02 | -0.18 | 0.12 | -0.17 | 0.10 | 0.18 | 0.04 |
| | RSA | 0.06 | -0.17 | 0.09 | **0.24** | -0.35* | -0.12 | -0.11 |
| | RSMDiff | 0.09 | -0.10 | 0.11 | -0.04 | -0.08 | 0.01 | -0.06 |
| Neighbors | 2nd-Cos | -0.08 | -0.15 | 0.05 | -0.20 | -0.18 | -0.22 | 0.17 |
| | Jaccard | -0.11 | -0.13 | -0.04 | -0.22 | 0.07 | -0.01 | 0.25 |
| | RankSim | 0.07 | 0.04 | 0.13 | -0.01 | 0.03 | 0.17 | **0.35*** |
| Topology | IMD | 0.17 | -0.20 | 0.12 | 0.08 | -0.17 | -0.23 | -0.19 |
| | RTD | 0.09 | 0.12 | -0.20 | -0.19 | 0.05 | -0.11 | 0.10 |
| Statistic | ConcDiff | -0.11 | **0.34*** | -0.04 | -0.11 | -0.13 | 0.00 | 0.18 |
| | MagDiff | -0.16 | 0.02 | -0.06 | -0.07 | -0.12 | 0.07 | 0.15 |
| | UnifDiff | -0.18 | nan | nan | 0.17 | -0.04 | nan | nan |

Table 20: Results of Test 1 (*Correlation to Accuracy Difference*) for the vision domain on CIFAR-100.

| | Eval. Dataset Arch. | Spearman CIFAR100 | | | | | | |
| --- | --- | --- | --- | --- | --- | --- | --- | --- |
| | | RNet18 | RNet34 | RNet101 | VGG11 | VGG19 | ViT B32 | ViT L32 |
| CCA | PWCCA | -0.30** | 0.07 | 0.23* | nan | nan | nan | nan |
| | SVCCA | -0.13 | 0.23 | -0.21 | -0.16 | -0.07 | -0.03 | 0.82** |
| Alignment | AlignCos | -0.26 | 0.09 | 0.37* | 0.21 | 0.41** | -0.12 | 0.62** |
| | AngShape | -0.31* | 0.14 | **0.44**** | -0.01 | 0.40** | -0.12 | 0.88** |
| | HardCorr | 0.08 | 0.09 | 0.24 | 0.06 | **0.50**** | **0.15** | 0.77** |
| | LinReg | -0.34** | 0.12 | 0.42** | nan | nan | -0.09 | -0.24* |
| | OrthProc | -0.31* | 0.14 | **0.44**** | -0.01 | 0.40** | -0.12 | 0.88** |
| | PermProc | -0.06 | -0.04 | 0.06 | -0.08 | 0.27 | 0.14 | **0.93**** |
| | ProcDist | -0.37* | -0.05 | 0.33* | -0.05 | 0.44** | -0.16 | 0.83** |
| | SoftCorr | -0.06 | 0.05 | 0.37* | 0.07 | 0.48** | 0.13 | 0.81** |
| RSM | CKA | -0.20 | 0.02 | 0.35* | 0.23 | 0.40** | -0.17 | 0.90** |
| | DistCorr | -0.23 | 0.06 | 0.33* | **0.24** | 0.40** | -0.17 | 0.90** |
| | EOS | -0.38* | 0.04 | -0.01 | 0.13 | 0.46** | 0.06 | 0.90** |
| | GULP | -0.36* | 0.03 | 0.43** | -0.12 | -0.24 | -0.31* | 0.62** |
| | RSA | -0.16 | -0.09 | 0.21 | 0.04 | 0.36* | -0.18 | 0.89** |
| | RSMDiff | -0.15 | 0.18 | 0.08 | 0.07 | 0.17 | 0.04 | 0.65** |
| Neighbors | 2nd-Cos | -0.33* | -0.11 | 0.32* | 0.19 | -0.07 | -0.21 | 0.88** |
| | Jaccard | -0.46** | 0.06 | 0.26 | **0.24** | 0.14 | -0.19 | 0.88** |
| | RankSim | -0.15 | -0.20 | 0.10 | 0.08 | 0.09 | -0.14 | 0.88** |
| Topology | IMD | -0.12 | 0.06 | -0.00 | -0.18 | 0.40* | -0.11 | 0.69** |
| | RTD | **0.13** | 0.12 | 0.12 | -0.03 | 0.31* | 0.13 | **0.93**** |
| Statistic | ConcDiff | -0.04 | **0.37*** | -0.21 | 0.23 | 0.25 | -0.05 | 0.53** |
| | MagDiff | -0.17 | 0.24 | 0.01 | -0.03 | 0.31* | -0.02 | 0.33* |
| | UnifDiff | -0.00 | -0.15 | -0.12 | -0.01 | 0.03 | nan | -0.08 |

Table 21: Results of Test 2 (*Correlation to Output Difference*) for the vision domain on ImageNet-100.

| Type | Test | | | | | | | | | | | | | |
|------|------|--|--|--|--|--|--|--|--|--|--|--|--|--|
| | | | | Grounding by Prediction | | | | | | | | | | |
| | | | | JSD Corr. Spearman IN100 | | | | | | | Disagr. Corr. Spearman IN100 | | | |
| | Eval. Dataset Arch. | RNet18 | RNet34 | RNet101 | VGG11 | VGG19 | ViT B32 | ViT L32 | RNet18 | RNet34 | RNet101 | VGG11 | VGG19 | ViT B32 | ViT L32 |
| CCA | PWCCA | 0.13 | 0.15 | 0.15 | -0.13 | 0.15 | 0.21 | -0.24* | 0.27** | 0.33** | 0.06 | -0.12 | -0.36** | 0.02 | -0.18 |
| | SVCCA | 0.21 | -0.00 | 0.25 | -0.11 | 0.16 | 0.05 | **0.18** | 0.39** | 0.07 | 0.14 | 0.03 | 0.01 | -0.06 | 0.07 |
| Alignment | AlignCos | 0.08 | 0.05 | 0.38* | 0.10 | -0.20 | -0.22 | -0.06 | 0.19 | **0.50**** | 0.17 | 0.16 | -0.13 | -0.08 | 0.00 |
| | AngShape | 0.24 | 0.22 | 0.34* | -0.01 | 0.19 | -0.26 | -0.15 | 0.24 | 0.40** | 0.19 | -0.14 | -0.33* | -0.11 | -0.06 |
| | HardCorr | 0.28 | 0.31* | 0.02 | -0.22 | -0.05 | 0.03 | -0.26 | 0.28 | -0.06 | -0.09 | -0.15 | -0.18 | 0.27 | -0.16 |
| | LinReg | 0.21* | 0.21* | 0.41** | -0.01 | 0.25* | -0.13 | -0.14 | 0.19 | 0.25* | 0.29** | -0.17 | -0.24* | 0.04 | -0.07 |
| | OrthProc | 0.24 | 0.22 | 0.34* | -0.02 | 0.19 | -0.26 | -0.15 | 0.24 | 0.40** | 0.19 | -0.14 | -0.33* | -0.11 | -0.06 |
| | PermProc | 0.18 | 0.18 | 0.27 | -0.18 | 0.06 | 0.36* | -0.06 | 0.13 | -0.25 | -0.04 | 0.02 | 0.20 | **0.37*** | **0.10** |
| | ProcDist | 0.10 | 0.14 | 0.39* | -0.05 | 0.27 | -0.05 | 0.02 | 0.08 | -0.08 | 0.11 | -0.10 | -0.07 | -0.07 | 0.08 |
| | SoftCorr | **0.45**** | 0.27 | 0.11 | -0.04 | -0.16 | 0.01 | -0.31* | **0.47**** | -0.13 | -0.07 | -0.03 | -0.29 | 0.27 | -0.16 |
| RSM | CKA | 0.30* | 0.08 | 0.30* | -0.13 | -0.06 | 0.04 | -0.07 | 0.37* | 0.08 | 0.22 | 0.01 | -0.24 | 0.00 | -0.02 |
| | DistCorr | 0.26 | 0.05 | 0.31* | -0.10 | 0.04 | 0.05 | -0.12 | 0.36* | 0.01 | 0.28 | -0.00 | -0.25 | 0.02 | -0.05 |
| | EOS | 0.09 | **0.49**** | 0.33* | -0.11 | 0.15 | -0.18 | -0.28 | 0.11 | 0.25 | 0.14 | -0.31* | -0.41** | 0.01 | -0.15 |
| | GULP | 0.07 | **0.49**** | 0.35* | -0.05 | 0.15 | -0.05 | -0.28 | 0.10 | 0.26 | 0.13 | -0.27 | -0.41** | 0.19 | -0.15 |
| | RSA | 0.12 | 0.18 | 0.09 | -0.18 | -0.19 | -0.11 | -0.20 | 0.19 | 0.33* | 0.10 | -0.05 | 0.04 | 0.00 | -0.04 |
| | RSMDiff | -0.41** | -0.22 | 0.30* | -0.27 | 0.07 | 0.02 | -0.28 | -0.17 | -0.20 | 0.18 | -0.01 | -0.03 | -0.21 | -0.17 |
| Neighbors | 2nd-Cos | -0.13 | 0.16 | 0.29 | 0.07 | -0.29 | 0.43** | -0.35* | -0.21 | 0.45** | 0.10 | 0.11 | -0.07 | 0.19 | -0.27 |
| | Jaccard | 0.36* | 0.26 | 0.32* | 0.05 | **0.33*** | 0.47** | -0.30* | 0.25 | 0.47** | 0.23 | 0.14 | -0.11 | 0.34* | -0.18 |
| | RankSim | -0.15 | -0.00 | 0.22 | 0.01 | 0.05 | 0.25 | -0.05 | -0.10 | -0.05 | 0.02 | 0.01 | -0.32* | 0.14 | -0.33* |
| Topology | IMD | -0.11 | 0.08 | 0.21 | **0.20** | 0.11 | 0.41** | 0.07 | -0.07 | 0.00 | 0.26 | 0.26 | 0.02 | 0.23 | 0.07 |
| | RTD | -0.18 | 0.02 | 0.17 | -0.02 | -0.21 | 0.20 | -0.27 | -0.08 | -0.06 | 0.33* | **0.27** | -0.29 | 0.36* | -0.23 |
| Statistic | ConcDiff | -0.29* | 0.24 | -0.11 | -0.17 | -0.13 | -0.11 | -0.37* | -0.09 | 0.00 | -0.21 | -0.11 | -0.06 | -0.08 | -0.29 |
| | MagDiff | -0.38* | -0.20 | 0.02 | -0.16 | -0.28 | 0.02 | -0.32* | -0.17 | -0.22 | -0.05 | -0.09 | 0.04 | -0.01 | -0.22 |
| | UnifDiff | -0.34* | nan | nan | 0.04 | -0.17 | nan | nan | -0.02 | nan | nan | 0.17 | **0.39**** | nan | nan |

Table 22: Results of Test 2 (*Correlation to Output Difference*) for the vision domain on CIFAR-100.

| Type | Test | | | | | | | | | | | | | |
|------|------|--|--|--|--|--|--|--|--|--|--|--|--|--|
| | | | | Grounding by Prediction | | | | | | | | | | |
| | | | | JSD Corr. Spearman CIFAR100 | | | | | | | Disagr. Corr. Spearman CIFAR100 | | | |
| | Eval. Dataset Arch. | RNet18 | RNet34 | RNet101 | VGG11 | VGG19 | ViT B32 | ViT L32 | RNet18 | RNet34 | RNet101 | VGG11 | VGG19 | ViT B32 | ViT L32 |
| CCA | PWCCA | -0.02 | 0.31** | 0.32** | nan | nan | nan | nan | -0.08 | 0.27** | 0.19 | nan | nan | nan | nan |
| | SVCCA | 0.18 | -0.28 | -0.43** | -0.24 | 0.23 | 0.39** | 0.81** | -0.01 | -0.22 | -0.25 | -0.23 | 0.17 | 0.07 | 0.78** |
| Alignment | AlignCos | 0.03 | 0.47** | 0.62** | **0.53**** | **0.71**** | 0.86** | 0.58** | -0.25 | 0.31* | 0.29 | **0.53**** | 0.44** | 0.56** | 0.55** |
| | AngShape | 0.12 | 0.41** | **0.64**** | 0.24 | 0.69** | 0.86** | 0.88** | -0.24 | 0.19 | 0.30* | 0.05 | 0.38* | 0.57** | 0.79** |
| | HardCorr | -0.09 | 0.11 | 0.11 | 0.01 | 0.28 | 0.44** | 0.74** | -0.24 | 0.13 | -0.10 | 0.23 | -0.22 | 0.39* | 0.66** |
| | LinReg | 0.03 | 0.34** | **0.64**** | nan | 0.00 | -0.31* | -0.31** | -0.32** | 0.19 | 0.30** | nan | nan | 0.11 | -0.36** |
| | OrthProc | 0.12 | 0.41** | **0.64**** | 0.23 | 0.69** | 0.86** | 0.88** | -0.36* | 0.19 | 0.30 | 0.05 | 0.38* | 0.57** | 0.79** |
| | PermProc | 0.06 | 0.11 | 0.10 | -0.35* | 0.42** | 0.51** | 0.95** | 0.08 | 0.15 | -0.03 | -0.05 | 0.23 | 0.37* | 0.90** |
| | ProcDist | 0.18 | 0.23 | 0.61** | 0.06 | 0.64** | **0.87**** | 0.85** | -0.02 | 0.20 | 0.32* | 0.01 | 0.33* | 0.60** | 0.79** |
| | SoftCorr | -0.06 | 0.15 | 0.43** | 0.13 | 0.20 | 0.41** | 0.77** | -0.21 | 0.15 | 0.16 | 0.32* | -0.27 | 0.37* | 0.69** |
| RSM | CKA | 0.16 | 0.47** | 0.47** | 0.13 | 0.68** | 0.81** | 0.89** | -0.31* | 0.17 | 0.15 | 0.14 | 0.37* | 0.52** | 0.80** |
| | DistCorr | 0.17 | 0.48** | 0.52** | 0.15 | 0.69** | 0.82** | 0.89** | -0.27 | 0.18 | 0.21 | 0.13 | 0.38* | 0.54** | 0.80** |
| | EOS | 0.22 | 0.27 | 0.00 | 0.21 | 0.25 | 0.57** | 0.94** | -0.20 | 0.20 | -0.09 | 0.38** | -0.24 | 0.21 | 0.91** |
| | GULP | 0.20 | 0.28 | 0.57** | 0.19 | 0.22 | 0.73** | 0.53** | -0.21 | 0.20 | 0.26 | 0.09 | **0.53**** | 0.47** | 0.38* |
| | RSA | 0.02 | 0.48** | 0.02 | -0.29 | 0.65** | 0.79** | 0.87** | -0.16 | 0.22 | -0.19 | -0.16 | 0.35* | 0.50** | 0.79** |
| | RSMDiff | **0.38**** | 0.24 | 0.13 | 0.20 | 0.28 | 0.22 | 0.69** | 0.34* | 0.06 | -0.08 | 0.15 | -0.02 | 0.17 | 0.68** |
| Neighbors | 2nd-Cos | 0.23 | 0.38* | 0.52** | 0.28 | 0.18 | 0.86** | 0.93** | 0.10 | 0.28 | 0.17 | 0.42** | 0.34* | 0.60** | **0.92**** |
| | Jaccard | 0.22 | 0.48** | 0.47** | 0.35* | 0.54** | 0.82** | 0.92** | 0.02 | 0.26 | 0.17 | 0.43** | 0.49** | 0.53** | 0.90** |
| | RankSim | 0.23 | 0.32* | 0.16 | 0.33* | 0.37* | 0.73** | 0.92** | -0.03 | 0.17 | -0.07 | 0.37* | 0.40* | 0.61** | **0.92**** |
| Topology | IMD | 0.25 | 0.01 | 0.09 | 0.06 | 0.42** | 0.01 | 0.60** | **0.36*** | -0.03 | 0.03 | -0.09 | 0.12 | -0.02 | 0.50** |
| | RTD | 0.09 | **0.53**** | 0.01 | 0.05 | 0.54** | 0.24 | **0.96**** | -0.03 | 0.30* | -0.05 | 0.11 | 0.20 | 0.09 | **0.92**** |
| Statistic | ConcDiff | 0.04 | 0.18 | -0.13 | 0.16 | 0.47** | 0.24 | 0.51** | -0.06 | **0.32*** | -0.19 | 0.30* | 0.21 | 0.13 | 0.49** |
| | MagDiff | 0.15 | 0.20 | 0.26 | 0.36* | 0.47** | 0.20 | 0.30* | -0.05 | 0.10 | 0.08 | 0.43** | 0.14 | 0.07 | 0.29 |
| | UnifDiff | -0.18 | -0.13 | 0.13 | 0.13 | 0.19 | nan | -0.13 | 0.05 | -0.16 | 0.17 | 0.26 | -0.07 | nan | -0.16 |

Table 23: Results of Test 3 (*Label Randomization*) for the vision domain on ImageNet-100.

| Type | | | | AUPRC IN100 | | | | | | | Conformity Rate IN100 | | | |
|------|--|--|--|--|--|--|--|--|--|--|--|--|--|--|
| | Eval. Dataset Arch. | RNet18 | RNet34 | RNet101 | VGG11 | VGG19 | ViT B32 | ViT L32 | RNet18 | RNet34 | RNet101 | VGG11 | VGG19 | ViT B32 | ViT L32 |
| CCA | PWCCA | 0.81 | 0.71 | 0.55 | nan | 0.74 | **1.00** | 0.89 | 0.93 | 0.83 | 0.82 | nan | 0.90 | **1.00** | 0.89 |
| | SVCCA | **1.00** | 0.98 | **1.00** | 0.98 | **1.00** | 0.42 | 0.46 | **1.00** | 0.99 | **1.00** | 0.99 | **1.00** | 0.50 | 0.71 |
| Alignment | AlignCos | 0.45 | 0.43 | 0.84 | 0.46 | 0.72 | 0.42 | 0.89 | 0.67 | 0.59 | 0.92 | 0.69 | 0.83 | 0.50 | 0.89 |
| | AngShape | 0.72 | 0.70 | **1.00** | **1.00** | **1.00** | 0.42 | 0.89 | 0.83 | 0.83 | **1.00** | **1.00** | **1.00** | 0.50 | 0.89 |
| | HardCorr | 0.72 | 0.69 | 0.71 | 0.98 | 0.75 | 0.71 | 0.79 | 0.83 | 0.82 | 0.74 | **1.00** | 0.86 | 0.83 | 0.84 |
| | LinReg | 0.91 | 0.63 | 0.72 | nan | **1.00** | 0.65 | 0.89 | 0.96 | 0.80 | 0.83 | nan | **1.00** | 0.78 | 0.89 |
| | OrthProc | 0.72 | 0.70 | **1.00** | **1.00** | **1.00** | 0.42 | 0.89 | 0.83 | 0.83 | **1.00** | **1.00** | **1.00** | 0.50 | 0.89 |
| | PermProc | 0.70 | 0.42 | 0.42 | 0.73 | 0.70 | 0.45 | **0.94** | 0.67 | 0.50 | 0.51 | 0.84 | 0.67 | 0.66 | **0.98** |
| | ProcDist | 0.70 | 0.43 | 0.71 | 0.79 | 0.70 | 0.44 | 0.91 | 0.68 | 0.55 | 0.79 | 0.94 | 0.67 | 0.65 | 0.95 |
| | SoftCorr | 0.72 | 0.45 | 0.70 | 0.83 | 0.70 | 0.72 | 0.79 | 0.83 | 0.68 | 0.73 | 0.95 | 0.73 | 0.83 | 0.82 |
| RSM | CKA | **1.00** | **1.00** | **1.00** | **1.00** | **1.00** | 0.42 | 0.89 | **1.00** | **1.00** | **1.00** | **1.00** | **1.00** | 0.50 | 0.89 |
| | DistCorr | **1.00** | **1.00** | **1.00** | **1.00** | **1.00** | 0.42 | 0.89 | **1.00** | **1.00** | **1.00** | **1.00** | **1.00** | 0.50 | 0.89 |
| | EOS | 0.84 | 0.71 | 0.96 | 0.70 | 0.76 | 0.73 | 0.89 | 0.95 | 0.83 | 0.99 | 0.89 | 0.89 | 0.91 | 0.89 |
| | GULP | 0.89 | 0.71 | 0.96 | 0.85 | 0.77 | **1.00** | 0.89 | 0.97 | 0.83 | 0.99 | 0.95 | 0.90 | **1.00** | 0.89 |
| | RSA | 0.75 | 0.63 | 0.95 | 0.97 | 0.98 | 0.42 | 0.89 | 0.89 | 0.79 | 0.98 | 0.99 | 0.99 | 0.50 | 0.89 |
| | RSMDiff | **1.00** | **1.00** | **1.00** | 0.96 | **1.00** | **1.00** | 0.85 | **1.00** | **1.00** | **1.00** | 0.99 | **1.00** | **1.00** | 0.93 |
| Neighbors | 2nd-Cos | **1.00** | **1.00** | **1.00** | 0.98 | **1.00** | 0.72 | 0.89 | **1.00** | **1.00** | **1.00** | **1.00** | **1.00** | 0.83 | 0.89 |
| | Jaccard | **1.00** | 0.73 | **1.00** | 0.84 | 0.99 | 0.47 | 0.89 | **1.00** | 0.87 | **1.00** | 0.95 | **1.00** | 0.73 | 0.89 |
| | RankSim | **1.00** | 0.89 | **1.00** | 0.80 | 0.98 | 0.73 | 0.89 | **1.00** | 0.96 | **1.00** | 0.93 | **1.00** | 0.84 | 0.89 |
| Topology | IMD | **1.00** | **1.00** | 0.77 | 0.99 | **1.00** | **1.00** | 0.57 | **1.00** | **1.00** | 0.94 | **1.00** | **1.00** | **1.00** | 0.87 |
| | RTD | **1.00** | **1.00** | **1.00** | **1.00** | **1.00** | 0.96 | 0.87 | **1.00** | **1.00** | **1.00** | **1.00** | **1.00** | 0.99 | 0.91 |
| Statistic | ConcDiff | 0.81 | 0.96 | **1.00** | 0.93 | 0.29 | 0.99 | 0.77 | 0.96 | 0.99 | **1.00** | 0.98 | 0.62 | **1.00** | 0.88 |
| | MagDiff | **1.00** | 0.73 | 0.53 | 0.97 | **1.00** | **1.00** | 0.86 | **1.00** | 0.94 | 0.81 | 0.99 | **1.00** | **1.00** | 0.93 |
| | UnifDiff | 0.21 | 0.18 | 0.17 | 0.45 | 0.29 | 0.17 | 0.17 | 0.38 | 0.21 | 0.00 | 0.67 | 0.62 | 0.00 | 0.00 |

Table 24: Results of Test 3 (*Label Randomization*) for the vision domain on CIFAR-100.

| | Eval. Dataset Arch. | AUPRC CIFAR100 | | | | | | | Conformity Rate CIFAR100 | | | | | | |
|---|---|---|---|---|---|---|---|---|---|---|---|---|---|---|---|
| | | RNet18 | RNet34 | RNet101 | VGG11 | VGG19 | ViT B32 | ViT L32 | RNet18 | RNet34 | RNet101 | VGG11 | VGG19 | ViT B32 | ViT L32 |
| CCA | PWCCA | 0.52 | 0.53 | nan | nan | nan | nan | nan | 0.73 | 0.67 | nan | nan | nan | nan | nan |
| | SVCCA | 0.97 | 0.72 | 0.82 | **1.00** | 0.70 | 0.63 | 0.33 | 0.99 | 0.84 | 0.94 | **1.00** | 0.88 | 0.73 | 0.70 |
| Alignment | AlignCos | 0.45 | 0.42 | 0.77 | 0.72 | 0.99 | 0.73 | 0.49 | 0.66 | 0.52 | 0.87 | 0.82 | **1.00** | 0.85 | 0.69 |
| | AngShape | 0.42 | 0.43 | 0.72 | 0.47 | **1.00** | 0.73 | 0.51 | 0.50 | 0.61 | 0.83 | 0.70 | **1.00** | 0.85 | 0.74 |
| | HardCorr | 0.45 | 0.44 | 0.70 | 0.44 | 0.61 | 0.81 | 0.41 | 0.66 | 0.63 | 0.67 | 0.63 | 0.76 | 0.95 | 0.68 |
| | LinReg | 0.30 | 0.31 | nan | nan | nan | 0.36 | 0.20 | 0.59 | 0.52 | nan | nan | nan | 0.48 | 0.49 |
| | OrthProc | 0.42 | 0.43 | 0.72 | 0.47 | **1.00** | 0.73 | 0.51 | 0.50 | 0.61 | 0.83 | 0.70 | **1.00** | 0.85 | 0.74 |
| | PermProc | 0.55 | 0.64 | 0.83 | 0.74 | 0.76 | 0.72 | 0.48 | 0.61 | 0.65 | 0.94 | 0.88 | 0.91 | 0.83 | 0.64 |
| | ProcDist | 0.43 | 0.53 | 0.72 | 0.70 | **1.00** | 0.72 | 0.52 | 0.52 | 0.59 | 0.83 | 0.67 | **1.00** | 0.84 | 0.74 |
| | SoftCorr | 0.44 | 0.44 | 0.70 | 0.43 | 0.59 | 0.78 | 0.38 | 0.65 | 0.63 | 0.67 | 0.60 | 0.77 | 0.93 | 0.65 |
| RSM | CKA | 0.45 | 0.44 | 0.69 | 0.87 | **1.00** | 0.71 | 0.42 | 0.66 | 0.62 | 0.66 | 0.96 | **1.00** | 0.80 | 0.69 |
| | DistCorr | 0.45 | 0.44 | 0.70 | 0.94 | **1.00** | 0.72 | 0.43 | 0.67 | 0.63 | 0.67 | 0.98 | **1.00** | 0.83 | 0.71 |
| | EOS | 0.72 | 0.53 | 0.51 | 0.43 | 0.74 | 0.63 | 0.42 | 0.83 | 0.66 | 0.58 | 0.55 | 0.82 | 0.78 | 0.68 |
| | GULP | 0.72 | 0.58 | **1.00** | 0.37 | 0.41 | 0.74 | 0.34 | 0.83 | 0.66 | **1.00** | 0.61 | 0.79 | 0.88 | 0.63 |
| | RSA | 0.45 | 0.44 | 0.72 | 0.47 | 0.81 | 0.71 | 0.52 | 0.66 | 0.63 | 0.80 | 0.71 | 0.93 | 0.79 | 0.73 |
| | RSMDiff | 0.96 | 0.86 | **1.00** | 0.99 | **1.00** | 0.95 | **0.72** | 0.98 | 0.93 | **1.00** | **1.00** | **1.00** | **0.99** | **0.89** |
| Neighbors | 2nd-Cos | 0.72 | 0.72 | 0.75 | 0.71 | 0.45 | 0.66 | 0.51 | 0.83 | 0.83 | 0.90 | 0.78 | 0.57 | 0.85 | 0.64 |
| | Jaccard | 0.42 | 0.43 | 0.72 | 0.73 | 0.58 | 0.88 | 0.51 | 0.50 | 0.59 | 0.83 | 0.84 | 0.78 | 0.98 | 0.69 |
| | RankSim | 0.43 | 0.43 | 0.72 | 0.72 | 0.60 | 0.92 | 0.53 | 0.54 | 0.60 | 0.83 | 0.84 | 0.79 | 0.97 | 0.68 |
| Topology | IMD | 0.58 | 0.83 | 0.98 | 0.76 | 0.83 | 0.61 | 0.46 | 0.83 | 0.92 | **1.00** | 0.85 | 0.95 | 0.87 | 0.68 |
| | RTD | **1.00** | **0.93** | **1.00** | 0.97 | 0.81 | 0.88 | 0.47 | **1.00** | **0.98** | **1.00** | 0.99 | 0.93 | 0.94 | 0.70 |
| Statistic | ConcDiff | 0.52 | 0.67 | 0.61 | 0.82 | 0.23 | 0.53 | 0.26 | 0.82 | 0.92 | 0.89 | 0.90 | 0.51 | 0.76 | 0.58 |
| | MagDiff | **1.00** | 0.58 | 0.59 | 0.78 | **1.00** | **0.97** | 0.66 | **1.00** | 0.84 | 0.84 | 0.93 | **1.00** | **0.99** | 0.85 |
| | UnifDiff | 0.50 | 0.54 | 0.55 | 0.47 | 0.51 | 0.27 | 0.48 | 0.81 | 0.77 | 0.76 | 0.68 | 0.77 | 0.60 | 0.71 |

Table 25: Results of Test 4 (*Shortcut Affinity*) for the vision domain on ImageNet-100.

| | Eval. Dataset Arch. | AUPRC IN100 | | | | | | | Conformity Rate IN100 | | | | | | |
|---|---|---|---|---|---|---|---|---|---|---|---|---|---|---|---|
| | | RNet18 | RNet34 | RNet101 | VGG11 | VGG19 | ViT B32 | ViT L32 | RNet18 | RNet34 | RNet101 | VGG11 | VGG19 | ViT B32 | ViT L32 |
| CCA | PWCCA | 0.99 | **1.00** | 0.65 | 0.99 | 0.75 | 0.82 | 0.93 | **1.00** | **1.00** | 0.71 | **1.00** | 0.90 | 0.97 | 0.97 |
| | SVCCA | 0.55 | 0.68 | 0.51 | 0.68 | 0.29 | 0.60 | 0.28 | 0.81 | 0.84 | 0.81 | 0.91 | 0.62 | 0.82 | 0.57 |
| Alignment | AlignCos | **1.00** | **1.00** | **1.00** | **1.00** | **1.00** | **1.00** | 0.99 | **1.00** | **1.00** | **1.00** | **1.00** | **1.00** | **1.00** | **1.00** |
| | AngShape | **1.00** | **1.00** | **1.00** | **1.00** | **1.00** | **1.00** | 0.99 | **1.00** | **1.00** | **1.00** | **1.00** | **1.00** | **1.00** | **1.00** |
| | HardCorr | 0.97 | **1.00** | 0.99 | 0.92 | 0.91 | 0.97 | 0.90 | 0.99 | **1.00** | **1.00** | 0.97 | 0.98 | 0.99 | 0.98 |
| | LinReg | 0.99 | **1.00** | **1.00** | **1.00** | 0.98 | 0.57 | 0.92 | 0.99 | **1.00** | **1.00** | **1.00** | 0.99 | 0.92 | 0.98 |
| | OrthProc | **1.00** | **1.00** | **1.00** | **1.00** | **1.00** | **1.00** | 0.99 | **1.00** | **1.00** | **1.00** | **1.00** | **1.00** | **1.00** | **1.00** |
| | PermProc | 0.72 | 0.80 | 0.94 | 0.66 | 0.82 | 0.97 | 0.77 | 0.89 | 0.94 | 0.99 | 0.87 | 0.93 | 0.98 | 0.89 |
| | ProcDist | **1.00** | **1.00** | **1.00** | **1.00** | **1.00** | **1.00** | 0.89 | **1.00** | **1.00** | **1.00** | **1.00** | **1.00** | **1.00** | 0.95 |
| | SoftCorr | 0.97 | 0.98 | 0.99 | 0.90 | 0.84 | 0.98 | 0.94 | 0.99 | **1.00** | **1.00** | 0.97 | 0.96 | **1.00** | 0.99 |
| RSM | CKA | **1.00** | **1.00** | **1.00** | **1.00** | **1.00** | **1.00** | 0.87 | **1.00** | **1.00** | **1.00** | **1.00** | **1.00** | **1.00** | 0.96 |
| | DistCorr | **1.00** | **1.00** | **1.00** | **1.00** | **1.00** | **1.00** | 0.88 | **1.00** | **1.00** | **1.00** | **1.00** | **1.00** | **1.00** | 0.96 |
| | EOS | **1.00** | **1.00** | 0.99 | 0.95 | 0.88 | 0.93 | 0.93 | **1.00** | **1.00** | **1.00** | 0.98 | 0.96 | 0.98 | 0.97 |
| | GULP | **1.00** | **1.00** | **1.00** | 0.96 | 0.88 | **1.00** | 0.93 | **1.00** | **1.00** | **1.00** | 0.98 | 0.96 | **1.00** | 0.97 |
| | RSA | **1.00** | **1.00** | **1.00** | **1.00** | **1.00** | **1.00** | 0.72 | **1.00** | **1.00** | **1.00** | **1.00** | **1.00** | **1.00** | 0.91 |
| | RSMDiff | 0.57 | 0.42 | 0.59 | 0.87 | 0.59 | 0.50 | 0.47 | 0.81 | 0.71 | 0.82 | 0.97 | 0.80 | 0.82 | 0.69 |
| Neighbors | 2nd-Cos | **1.00** | **1.00** | **1.00** | **1.00** | **1.00** | **1.00** | 0.92 | **1.00** | **1.00** | **1.00** | **1.00** | **1.00** | **1.00** | 0.97 |
| | Jaccard | **1.00** | **1.00** | **1.00** | **1.00** | **1.00** | **1.00** | 0.90 | **1.00** | **1.00** | **1.00** | **1.00** | **1.00** | **1.00** | 0.98 |
| | RankSim | 0.99 | 0.99 | **1.00** | **1.00** | **1.00** | **1.00** | 0.89 | **1.00** | **1.00** | **1.00** | **1.00** | **1.00** | **1.00** | 0.97 |
| Topology | IMD | 0.67 | 0.75 | 0.55 | 0.78 | 0.65 | 0.38 | 0.34 | 0.87 | 0.88 | 0.73 | 0.93 | 0.77 | 0.74 | 0.66 |
| | RTD | **1.00** | 0.98 | **1.00** | **1.00** | **1.00** | 0.75 | 0.88 | **1.00** | **1.00** | **1.00** | **1.00** | **1.00** | 0.93 | 0.96 |
| Statistic | ConcDiff | 0.53 | 0.70 | 0.50 | 0.78 | 0.27 | 0.28 | 0.25 | 0.83 | 0.86 | 0.81 | 0.95 | 0.67 | 0.58 | 0.64 |
| | MagDiff | 0.37 | 0.37 | 0.44 | 0.53 | 0.23 | 0.23 | 0.47 | 0.62 | 0.75 | 0.79 | 0.85 | 0.51 | 0.57 | 0.79 |
| | UnifDiff | 0.75 | 0.73 | 0.87 | 0.60 | 0.55 | 0.61 | 0.17 | 0.90 | 0.91 | 0.96 | 0.83 | 0.83 | 0.84 | 0.00 |

Table 26: Results of Test 4 (*Shortcut Affinity*) for the vision domain on CIFAR-100.

| | Eval. Dataset Arch. | AUPRC CIFAR100 | | | | | | | Conformity Rate CIFAR100 | | | | | | |
|---|---|---|---|---|---|---|---|---|---|---|---|---|---|---|---|
| | | RNet18 | RNet34 | RNet101 | VGG11 | VGG19 | ViT B32 | ViT L32 | RNet18 | RNet34 | RNet101 | VGG11 | VGG19 | ViT B32 | ViT L32 |
| CCA | PWCCA | **1.00** | **1.00** | **1.00** | nan | nan | nan | nan | **1.00** | **1.00** | **1.00** | nan | nan | nan | nan |
| | SVCCA | 0.66 | 0.91 | 0.91 | 0.87 | 0.50 | 0.79 | 0.53 | 0.80 | 0.97 | 0.97 | 0.93 | 0.78 | 0.90 | 0.65 |
| Alignment | AlignCos | **1.00** | **1.00** | **1.00** | **1.00** | **1.00** | 0.99 | 0.71 | **1.00** | **1.00** | **1.00** | **1.00** | **1.00** | **1.00** | 0.74 |
| | AngShape | **1.00** | **1.00** | **1.00** | **1.00** | **1.00** | 0.99 | 0.77 | **1.00** | **1.00** | **1.00** | **1.00** | **1.00** | **1.00** | 0.76 |
| | HardCorr | **1.00** | **1.00** | **1.00** | 0.85 | 0.76 | 0.82 | 0.78 | **1.00** | **1.00** | **1.00** | 0.92 | 0.91 | 0.87 | 0.77 |
| | LinReg | **1.00** | **1.00** | **1.00** | nan | nan | 0.20 | 0.38 | **1.00** | **1.00** | **1.00** | nan | nan | 0.43 | 0.50 |
| | OrthProc | **1.00** | **1.00** | **1.00** | **1.00** | **1.00** | 0.99 | 0.77 | **1.00** | **1.00** | **1.00** | **1.00** | **1.00** | **1.00** | 0.76 |
| | PermProc | 0.92 | 0.84 | **1.00** | 0.84 | 0.87 | 0.91 | 0.66 | 0.98 | 0.95 | **1.00** | 0.93 | 0.97 | 0.95 | 0.72 |
| | ProcDist | **1.00** | **1.00** | **1.00** | **1.00** | **1.00** | 0.92 | 0.70 | **1.00** | **1.00** | **1.00** | **1.00** | **1.00** | 0.97 | 0.72 |
| | SoftCorr | **1.00** | **1.00** | **1.00** | 0.71 | 0.79 | 0.91 | 0.78 | **1.00** | **1.00** | **1.00** | 0.89 | 0.92 | 0.95 | 0.76 |
| RSM | CKA | **1.00** | **1.00** | **1.00** | **1.00** | **1.00** | 0.91 | 0.74 | **1.00** | **1.00** | **1.00** | **1.00** | **1.00** | 0.96 | 0.75 |
| | DistCorr | **1.00** | **1.00** | **1.00** | **1.00** | **1.00** | 0.92 | 0.72 | **1.00** | **1.00** | **1.00** | **1.00** | **1.00** | 0.96 | 0.75 |
| | EOS | **1.00** | **1.00** | 0.42 | 0.94 | 0.95 | **1.00** | 0.78 | **1.00** | **1.00** | 0.50 | 0.98 | 0.98 | **1.00** | 0.79 |
| | GULP | **1.00** | **1.00** | **1.00** | 0.71 | 0.65 | 0.64 | 0.46 | **1.00** | **1.00** | **1.00** | 0.93 | 0.89 | 0.91 | 0.71 |
| | RSA | **1.00** | **1.00** | **1.00** | **1.00** | **1.00** | 0.91 | 0.69 | **1.00** | **1.00** | **1.00** | **1.00** | **1.00** | 0.95 | 0.74 |
| | RSMDiff | **1.00** | 0.96 | **1.00** | 0.93 | 0.93 | 0.86 | 0.41 | **1.00** | 0.98 | **1.00** | 0.97 | 0.98 | 0.86 | 0.65 |
| Neighbors | 2nd-Cos | **1.00** | **1.00** | **1.00** | **1.00** | 0.88 | **1.00** | 0.79 | **1.00** | **1.00** | **1.00** | **1.00** | 0.97 | **1.00** | **0.82** |
| | Jaccard | **1.00** | **1.00** | **1.00** | **1.00** | **1.00** | **1.00** | 0.79 | **1.00** | **1.00** | **1.00** | **1.00** | **1.00** | **1.00** | 0.81 |
| | RankSim | **1.00** | **1.00** | **1.00** | **1.00** | **1.00** | **1.00** | 0.79 | **1.00** | **1.00** | **1.00** | **1.00** | **1.00** | **1.00** | 0.81 |
| Topology | IMD | 0.95 | 0.94 | 0.83 | 0.99 | 0.60 | 0.95 | 0.50 | 0.99 | 0.98 | 0.91 | **1.00** | 0.84 | 0.98 | 0.78 |
| | RTD | **1.00** | **1.00** | **1.00** | **1.00** | 0.99 | 0.99 | **0.79** | **1.00** | **1.00** | **1.00** | **1.00** | **1.00** | **1.00** | 0.81 |
| Statistic | ConcDiff | 0.56 | 0.70 | **1.00** | 0.99 | 0.15 | 0.36 | 0.50 | 0.84 | 0.94 | **1.00** | **1.00** | 0.42 | 0.65 | 0.67 |
| | MagDiff | 0.99 | 0.65 | 0.89 | **1.00** | 0.51 | 0.22 | 0.42 | **1.00** | 0.91 | 0.97 | **1.00** | 0.81 | 0.52 | 0.71 |
| | UnifDiff | 0.47 | 0.47 | 0.30 | 0.21 | 0.50 | 0.17 | 0.18 | 0.70 | 0.80 | 0.68 | 0.41 | 0.71 | 0.00 | 0.20 |

Table 27: Results of Test 5 (*Augmentation*) for the vision domain on ImageNet-100.

| | Eval. Dataset Arch. | AUPRC IN100 | | | | | | | Conformity Rate IN100 | | | | | | |
|---|---|---|---|---|---|---|---|---|---|---|---|---|---|---|---|
| | | RNet18 | RNet34 | RNet101 | VGG11 | VGG19 | ViT B32 | ViT L32 | RNet18 | RNet34 | RNet101 | VGG11 | VGG19 | ViT B32 | ViT L32 |
| CCA | PWCCA | 0.90 | 0.57 | 0.51 | 0.96 | 0.73 | nan | 0.69 | 0.97 | 0.86 | 0.79 | 0.99 | 0.92 | nan | 0.89 |
| | SVCCA | 0.40 | 0.58 | 0.47 | 0.78 | 0.47 | 0.21 | 0.29 | 0.71 | 0.78 | 0.68 | 0.92 | 0.71 | 0.52 | 0.61 |
| Alignment | AlignCos | 0.71 | 0.84 | 0.90 | **1.00** | 0.81 | 0.76 | 0.75 | 0.94 | 0.96 | 0.97 | **1.00** | 0.95 | **0.92** | 0.94 |
| | AngShape | 0.71 | 0.52 | 0.56 | 0.99 | 0.74 | 0.74 | 0.80 | 0.90 | 0.81 | 0.83 | **1.00** | 0.94 | 0.91 | 0.96 |
| | HardCorr | 0.46 | 0.44 | 0.50 | 0.59 | 0.43 | 0.42 | 0.64 | 0.71 | 0.68 | 0.76 | 0.84 | 0.82 | 0.78 | 0.86 |
| | LinReg | 0.94 | 0.64 | 0.75 | **1.00** | 0.73 | 0.37 | 0.85 | 0.98 | 0.90 | 0.92 | **1.00** | 0.94 | 0.65 | 0.94 |
| | OrthProc | 0.71 | 0.52 | 0.56 | 0.99 | 0.74 | 0.74 | 0.80 | 0.90 | 0.81 | 0.83 | **1.00** | 0.94 | 0.91 | 0.96 |
| | PermProc | 0.41 | 0.33 | 0.54 | 0.63 | 0.55 | 0.28 | 0.61 | 0.64 | 0.57 | 0.85 | 0.87 | 0.79 | 0.65 | 0.86 |
| | ProcDist | 0.58 | 0.45 | 0.67 | 0.96 | 0.79 | **0.81** | **0.91** | 0.82 | 0.65 | 0.89 | 0.99 | 0.93 | **0.92** | **0.97** |
| | SoftCorr | 0.45 | 0.45 | 0.50 | 0.53 | 0.47 | 0.43 | 0.64 | 0.69 | 0.65 | 0.75 | 0.81 | 0.85 | 0.79 | 0.86 |
| RSM | CKA | 0.90 | 0.73 | 0.67 | 0.99 | 0.75 | 0.37 | 0.71 | 0.97 | 0.92 | 0.89 | **1.00** | 0.95 | 0.73 | 0.94 |
| | DistCorr | 0.83 | 0.68 | 0.62 | 0.99 | 0.78 | 0.39 | 0.70 | 0.96 | 0.90 | 0.87 | **1.00** | **0.96** | 0.74 | 0.94 |
| | EOS | 0.93 | 0.59 | 0.69 | 0.98 | **0.88** | 0.60 | 0.72 | 0.98 | 0.86 | 0.89 | **1.00** | **0.96** | 0.85 | 0.88 |
| | GULP | 0.92 | 0.60 | 0.69 | 0.98 | **0.88** | 0.45 | 0.72 | 0.98 | 0.86 | 0.89 | **1.00** | **0.96** | 0.85 | 0.88 |
| | RSA | **0.98** | 0.82 | 0.78 | 0.75 | 0.46 | 0.34 | 0.59 | **1.00** | 0.95 | 0.92 | 0.93 | 0.81 | 0.71 | 0.88 |
| | RSMDiff | 0.45 | 0.47 | 0.55 | 0.57 | 0.58 | 0.24 | 0.32 | 0.79 | 0.79 | 0.78 | 0.85 | 0.87 | 0.57 | 0.70 |
| Neighbors | 2nd-Cos | 0.78 | 0.88 | **0.91** | 0.76 | 0.80 | 0.58 | 0.84 | 0.95 | 0.98 | **0.98** | 0.93 | 0.94 | 0.85 | 0.94 |
| | Jaccard | 0.79 | 0.72 | 0.86 | 0.95 | 0.81 | 0.72 | 0.81 | 0.95 | 0.93 | 0.95 | 0.99 | 0.94 | 0.91 | 0.93 |
| | RankSim | 0.71 | 0.57 | 0.89 | 0.83 | 0.80 | 0.51 | 0.65 | 0.92 | 0.87 | 0.96 | 0.95 | 0.92 | 0.82 | 0.87 |
| Topology | IMD | 0.56 | 0.65 | 0.64 | 0.78 | 0.75 | 0.23 | 0.21 | 0.82 | 0.90 | 0.89 | 0.94 | 0.92 | 0.52 | 0.50 |
| | RTD | 0.57 | 0.59 | 0.76 | 0.89 | 0.75 | 0.30 | 0.39 | 0.86 | 0.88 | 0.92 | 0.98 | 0.91 | 0.66 | 0.74 |
| Statistic | ConcDiff | 0.43 | **0.96** | 0.66 | **1.00** | 0.30 | 0.41 | 0.28 | 0.75 | **0.99** | 0.91 | **1.00** | 0.67 | 0.74 | 0.51 |
| | MagDiff | 0.37 | 0.81 | 0.46 | 0.99 | 0.22 | 0.43 | 0.18 | 0.74 | 0.95 | 0.79 | **1.00** | 0.54 | 0.75 | 0.46 |
| | UnifDiff | 0.17 | 0.18 | 0.22 | 0.18 | 0.21 | 0.17 | 0.17 | 0.00 | 0.45 | 0.51 | 0.27 | 0.42 | 0.00 | 0.00 |

Table 28: Results of Test 5 (*Augmentation*) for the vision domain on CIFAR-100.

| | Eval. Dataset Arch. | AUPRC CIFAR100 | | | | | Conformity Rate CIFAR100 | | | | |
|---|---|---|---|---|---|---|---|---|---|---|---|
| | | RNet18 | RNet34 | RNet101 | VGG11 | VGG19 | RNet18 | RNet34 | RNet101 | VGG11 | VGG19 |
| CCA | PWCCA | 0.43 | 0.47 | 0.94 | nan | nan | 0.60 | 0.73 | 0.98 | nan | nan |
| | SVCCA | 0.41 | 0.55 | 0.61 | 0.42 | 0.25 | 0.62 | 0.81 | 0.80 | 0.65 | 0.49 |
| Alignment | AlignCos | 0.98 | **1.00** | **1.00** | 0.98 | 0.50 | 0.99 | **1.00** | **1.00** | 0.99 | 0.82 |
| | AngShape | 0.43 | 0.54 | 0.98 | 0.43 | 0.49 | 0.58 | 0.77 | **1.00** | 0.57 | 0.81 |
| | HardCorr | 0.49 | 0.54 | 0.99 | 0.41 | 0.38 | 0.73 | 0.82 | **1.00** | 0.57 | 0.71 |
| | LinReg | 0.45 | 0.50 | 0.98 | nan | nan | 0.67 | 0.76 | 0.99 | nan | nan |
| | OrthProc | 0.43 | 0.54 | 0.98 | 0.43 | 0.49 | 0.58 | 0.77 | 0.99 | 0.57 | 0.81 |
| | PermProc | 0.51 | 0.66 | 0.52 | 0.44 | 0.36 | 0.74 | 0.83 | 0.77 | 0.71 | 0.69 |
| | ProcDist | 0.42 | 0.76 | **1.00** | 0.43 | 0.43 | 0.50 | 0.92 | **1.00** | 0.58 | 0.78 |
| | SoftCorr | 0.51 | 0.53 | **1.00** | 0.41 | 0.42 | 0.74 | 0.81 | **1.00** | 0.56 | 0.73 |
| RSM | CKA | 0.45 | 0.73 | **1.00** | 0.69 | 0.49 | 0.68 | 0.86 | **1.00** | 0.92 | 0.83 |
| | DistCorr | 0.63 | 0.74 | **1.00** | 0.79 | 0.52 | 0.83 | 0.88 | **1.00** | 0.94 | 0.84 |
| | EOS | 0.45 | 0.46 | 0.84 | 0.52 | 0.54 | 0.66 | 0.71 | 0.94 | 0.81 | 0.80 |
| | GULP | 0.45 | 0.46 | **1.00** | 0.34 | 0.33 | 0.66 | 0.72 | **1.00** | 0.68 | 0.70 |
| | RSA | 0.66 | 0.71 | **1.00** | 0.66 | 0.42 | 0.91 | 0.88 | **1.00** | 0.88 | 0.77 |
| | RSMDiff | 0.92 | 0.89 | **1.00** | 0.63 | 0.28 | 0.98 | 0.97 | **1.00** | 0.83 | 0.52 |
| Neighbors | 2nd-Cos | 0.46 | 0.45 | 0.63 | 0.47 | 0.55 | 0.70 | 0.69 | 0.89 | 0.75 | 0.82 |
| | Jaccard | 0.81 | 0.91 | **1.00** | **1.00** | 0.74 | 0.94 | 0.98 | **1.00** | **1.00** | 0.92 |
| | RankSim | 0.96 | 0.99 | **1.00** | 0.89 | 0.69 | 0.99 | **1.00** | **1.00** | 0.96 | 0.92 |
| Topology | IMD | **1.00** | 0.87 | 0.89 | **1.00** | 0.29 | **1.00** | 0.96 | 0.97 | **1.00** | 0.56 |
| | RTD | **1.00** | **1.00** | **1.00** | **1.00** | 0.88 | **1.00** | **1.00** | **1.00** | **1.00** | 0.95 |
| Statistic | ConcDiff | **1.00** | **1.00** | **1.00** | **1.00** | 0.39 | **1.00** | **1.00** | **1.00** | **1.00** | 0.78 |
| | MagDiff | **1.00** | 0.81 | 0.63 | **1.00** | 0.25 | **1.00** | 0.96 | 0.89 | **1.00** | 0.64 |
| | UnifDiff | 0.89 | 0.98 | **1.00** | 0.20 | 0.63 | 0.97 | **1.00** | **1.00** | 0.57 | 0.84 |

Table 29: Results of Test 6 (*Layer Monotonicity*) for the vision domain on ImageNet-100.

| | Eval. Dataset Arch. | Conformity Rate IN100 | | | | | | | Spearman IN100 | | | | | | |
|---|---|---|---|---|---|---|---|---|---|---|---|---|---|---|---|
| | | RNet18 | RNet34 | RNet101 | VGG11 | VGG19 | ViT B32 | ViT L32 | RNet18 | RNet34 | RNet101 | VGG11 | VGG19 | ViT B32 | ViT L32 |
| CCA | PWCCA | 0.82 | 0.89 | 0.95 | 0.75 | 0.83 | **1.00** | **1.00** | 0.11 | 0.30 | **1.00** | 0.32 | 0.55 | **1.00** | **1.00** |
| | SVCCA | 0.72 | 0.58 | 0.72 | 0.75 | 0.69 | 0.86 | 0.78 | 0.20 | 0.27 | 0.43 | 0.42 | 0.40 | 0.87 | 0.61 |
| Alignment | AlignCos | 0.84 | 0.86 | 0.80 | 0.90 | 0.68 | **1.00** | **1.00** | 0.52 | 0.63 | 0.52 | 0.93 | 0.11 | **1.00** | **1.00** |
| | AngShape | 0.85 | 0.90 | 0.90 | 0.97 | 0.98 | **1.00** | **1.00** | 0.55 | 0.65 | 0.65 | 0.96 | 0.99 | **1.00** | **1.00** |
| | HardCorr | 0.61 | 0.85 | 0.89 | 0.92 | 0.93 | **1.00** | **1.00** | 0.01 | 0.53 | 0.76 | 0.91 | 0.73 | **1.00** | **1.00** |
| | LinReg | 0.85 | 0.94 | 0.89 | 0.82 | 0.91 | 0.99 | **1.00** | 0.55 | 0.95 | 0.90 | 0.48 | 0.77 | **1.00** | **1.00** |
| | OrthProc | 0.85 | 0.90 | 0.90 | 0.97 | 0.98 | **1.00** | **1.00** | 0.55 | 0.65 | 0.65 | 0.96 | 0.99 | **1.00** | **1.00** |
| | PermProc | 0.63 | 0.70 | 0.61 | 0.71 | 0.59 | 0.70 | **1.00** | 0.20 | 0.60 | 0.39 | 0.69 | 0.14 | 0.71 | **1.00** |
| | ProcDist | 0.85 | 0.80 | 0.79 | 0.80 | 0.81 | 0.70 | **1.00** | 0.55 | 0.42 | 0.39 | 0.48 | 0.67 | 0.71 | **1.00** |
| | SoftCorr | 0.71 | 0.85 | 0.88 | 0.80 | 0.89 | **1.00** | **1.00** | 0.11 | 0.50 | 0.70 | 0.52 | 0.64 | **1.00** | **1.00** |
| RSM | CKA | 0.93 | 0.97 | **0.99** | 0.90 | 0.92 | **1.00** | **1.00** | 0.87 | 0.82 | 0.97 | 0.88 | 0.93 | **1.00** | **1.00** |
| | DistCorr | 0.95 | **0.98** | **0.99** | 0.90 | 0.92 | **1.00** | **1.00** | **0.97** | 0.79 | 0.97 | 0.88 | 0.93 | **1.00** | **1.00** |
| | EOS | 0.90 | 0.95 | 0.96 | **1.00** | **1.00** | **1.00** | **1.00** | 0.88 | **0.96** | 0.97 | **1.00** | **1.00** | **1.00** | **1.00** |
| | GULP | 0.70 | 0.70 | 0.80 | **1.00** | **1.00** | **1.00** | **1.00** | 0.53 | 0.47 | 0.64 | **1.00** | **1.00** | **1.00** | **1.00** |
| | RSA | 0.95 | 0.94 | 0.97 | 0.81 | 0.91 | **1.00** | **1.00** | **0.97** | 0.72 | 0.88 | 0.58 | 0.66 | **1.00** | **1.00** |
| | RSMDiff | 0.45 | 0.50 | 0.48 | 0.85 | 0.69 | 0.75 | **1.00** | -0.33 | -0.09 | -0.21 | 0.85 | 0.65 | 0.75 | **1.00** |
| Neighbors | 2nd-Cos | 0.85 | 0.91 | 0.95 | **1.00** | **1.00** | **1.00** | **1.00** | 0.55 | 0.77 | 0.92 | **1.00** | **1.00** | **1.00** | **1.00** |
| | Jaccard | 0.85 | 0.90 | 0.91 | **1.00** | **1.00** | **1.00** | **1.00** | 0.55 | 0.65 | 0.75 | **1.00** | **1.00** | **1.00** | **1.00** |
| | RankSim | 0.85 | 0.90 | 0.90 | **1.00** | **1.00** | **1.00** | **1.00** | 0.55 | 0.65 | 0.67 | **1.00** | **1.00** | **1.00** | **1.00** |
| Topology | IMD | 0.52 | 0.66 | 0.61 | 0.48 | 0.51 | 0.86 | 0.54 | -0.00 | 0.23 | 0.07 | -0.03 | 0.09 | 0.59 | 0.38 |
| | RTD | **0.99** | 0.92 | 0.82 | 0.83 | 0.95 | 0.93 | **1.00** | **0.97** | 0.79 | 0.43 | 0.52 | 0.82 | 0.90 | **1.00** |
| Statistic | ConcDiff | 0.20 | 0.46 | 0.46 | 0.32 | 0.43 | 0.90 | **1.00** | -0.78 | -0.05 | -0.14 | -0.25 | -0.27 | 0.65 | **1.00** |
| | MagDiff | 0.35 | 0.45 | 0.55 | 0.65 | 0.65 | 0.86 | **1.00** | -0.37 | 0.13 | 0.14 | 0.21 | 0.29 | 0.84 | **1.00** |
| | UnifDiff | 0.65 | 0.68 | 0.81 | 0.39 | 0.49 | **1.00** | **1.00** | 0.18 | 0.20 | 0.55 | -0.30 | -0.06 | nan | nan |

Table 30: Results of Test 6 (*Layer Monotonicity*) for the vision domain on CIFAR-100.

| | Eval. Dataset Arch. | Conformity Rate CIFAR100 | | | | | | | Spearman CIFAR100 | | | | | | |
|---|---|---|---|---|---|---|---|---|---|---|---|---|---|---|---|
| | | RNet18 | RNet34 | RNet101 | VGG11 | VGG19 | ViT B32 | ViT L32 | RNet18 | RNet34 | RNet101 | VGG11 | VGG19 | ViT B32 | ViT L32 |
| CCA | PWCCA | 0.88 | 0.88 | 0.90 | **1.00** | **1.00** | 0.96 | **1.00** | 0.02 | 0.65 | **1.00** | **1.00** | nan | 0.10 | nan |
| | SVCCA | 0.53 | 0.68 | 0.58 | 0.66 | 0.70 | 0.80 | 0.93 | -0.28 | 0.27 | -0.06 | 0.40 | 0.31 | 0.66 | 0.87 |
| Alignment | AlignCos | 0.77 | 0.76 | 0.68 | 0.82 | 0.77 | 0.96 | **1.00** | 0.19 | 0.41 | 0.18 | 0.53 | 0.41 | 0.82 | **1.00** |
| | AngShape | 0.80 | 0.87 | 0.69 | **1.00** | 0.93 | 0.97 | 0.97 | 0.22 | 0.51 | 0.18 | **1.00** | 0.94 | 0.98 | 0.90 |
| | HardCorr | 0.75 | 0.77 | 0.59 | 0.85 | 0.89 | **1.00** | 0.95 | 0.13 | 0.25 | -0.11 | 0.55 | 0.65 | **1.00** | 0.81 |
| | LinReg | 0.95 | 0.88 | 0.75 | 0.98 | 0.60 | 0.42 | 0.47 | 0.92 | 0.82 | 0.50 | 0.98 | -0.05 | -0.27 | -0.13 |
| | OrthProc | 0.80 | 0.87 | 0.69 | **1.00** | 0.93 | 0.97 | 0.96 | 0.22 | 0.51 | 0.18 | **1.00** | 0.94 | 0.98 | 0.88 |
| | PermProc | 0.75 | 0.86 | **0.92** | 0.74 | 0.61 | 0.99 | **1.00** | 0.32 | 0.68 | 0.69 | 0.61 | 0.18 | 0.99 | **1.00** |
| | ProcDist | 0.70 | 0.92 | 0.84 | 0.70 | 0.76 | 0.97 | 0.97 | 0.10 | 0.76 | 0.33 | 0.60 | 0.66 | 0.98 | 0.92 |
| | SoftCorr | 0.76 | 0.80 | 0.59 | 0.81 | 0.91 | **1.00** | 0.97 | 0.14 | 0.31 | -0.10 | 0.52 | 0.66 | **1.00** | 0.86 |
| RSM | CKA | 0.65 | 0.77 | 0.87 | 0.95 | 0.99 | 0.97 | 0.97 | 0.07 | 0.28 | 0.62 | 0.94 | **0.99** | 0.98 | 0.87 |
| | DistCorr | 0.65 | 0.80 | 0.88 | 0.99 | 0.98 | 0.97 | 0.97 | 0.07 | 0.30 | 0.64 | 0.99 | **0.99** | 0.98 | 0.88 |
| | EOS | **1.00** | **0.95** | 0.85 | **1.00** | 0.51 | 0.91 | 0.95 | **1.00** | **0.96** | 0.78 | **1.00** | 0.06 | 0.89 | 0.92 |
| | GULP | 0.80 | 0.80 | 0.82 | 0.99 | 0.72 | 0.80 | 0.78 | 0.65 | 0.66 | 0.67 | 0.99 | 0.52 | 0.60 | 0.61 |
| | RSA | 0.65 | 0.92 | **0.92** | **1.00** | 0.83 | 0.99 | **1.00** | 0.07 | 0.56 | 0.70 | **1.00** | 0.68 | 0.99 | **1.00** |
| | RSMDiff | 0.45 | 0.52 | 0.56 | 0.50 | 0.69 | 0.84 | 0.97 | -0.26 | 0.34 | 0.30 | 0.25 | 0.67 | 0.82 | 0.93 |
| Neighbors | 2nd-Cos | 0.90 | 0.89 | 0.74 | **1.00** | 0.88 | **1.00** | **1.00** | 0.58 | 0.62 | 0.20 | **1.00** | 0.74 | **1.00** | **1.00** |
| | Jaccard | 0.85 | 0.83 | 0.65 | 0.95 | 0.89 | **1.00** | **1.00** | 0.55 | 0.52 | 0.02 | 0.97 | 0.72 | **1.00** | **1.00** |
| | RankSim | 0.85 | 0.85 | 0.64 | 0.95 | 0.90 | **1.00** | **1.00** | 0.55 | 0.57 | 0.02 | 0.97 | 0.75 | **1.00** | **1.00** |
| Topology | IMD | 0.51 | 0.42 | 0.67 | 0.62 | 0.68 | 0.67 | 0.70 | 0.28 | -0.02 | 0.36 | 0.25 | 0.37 | 0.53 | 0.42 |
| | RTD | 0.95 | 0.92 | 0.72 | **1.00** | 0.96 | 0.91 | 0.95 | 0.67 | 0.93 | 0.59 | **1.00** | 0.95 | 0.93 | 0.93 |
| Statistic | ConcDiff | 0.36 | 0.47 | 0.62 | 0.30 | 0.44 | 0.80 | 0.78 | -0.40 | 0.00 | 0.22 | -0.28 | -0.35 | 0.56 | 0.72 |
| | MagDiff | 0.48 | 0.55 | 0.56 | 0.39 | 0.58 | 0.61 | **1.00** | -0.12 | 0.21 | 0.31 | -0.30 | 0.42 | 0.16 | **1.00** |
| | UnifDiff | 0.78 | 0.74 | 0.76 | 0.40 | 0.63 | 0.98 | **1.00** | 0.48 | 0.16 | 0.16 | -0.41 | 0.50 | 0.79 | 0.96 |

# D    RUNTIME EXPERIMENTS

While running our experiments, we have also recorded computational runtimes of all comparisons per measure. In Table 31, we provide an overview of runtimes averaged across all experiments, separated by architecture. We note that these similarities were not all computed on the same machine, and also, there may be variations over different datasets that were aggregated here, given that these often varied in size.

Yet, one can identify clear differences in the runtimes of different measures, which even span different orders of magnitudes. Specifically, the topology-based IMD and RTD scores were consistently among the slowest measures, and further, RSM-based measures such as RSA, EOS and in particular RSMDiff are also significantly slower than, for instance, most alignment-based measures. Given that the computational cost of RSMs is generally quadratic in the number of inputs, it is also no surprise that these algorithms do not scale that well.

Table 31: *Runtimes of the similarity measures in seconds, averaged over all comparisons.* Since embedding dimensions and the number of inputs vary for different datasets, and we aggregate across experiments and datasets within each domain, the runtime should be interpreted as a broad estimate of the runtime to be expected.

| Modality | Graphs | | | | NLP | | | | | | Vision | | | |
| Architecture | GCN | SAGE | GAT | PGNN | BERT | ALBERT | SmolLM2 | RNet18 | RNet34 | RNet101 | VGG11 | VGG19 | ViT-B/32 | ViT-L/32 |
| --- | --- | --- | --- | --- | --- | --- | --- | --- | --- | --- | --- | --- | --- | --- |
| 2nd-Cos | 3.935 | 3.984 | 4.142 | 0.073 | 1.991 | 2.237 | 3.984 | 2.635 | 2.382 | 5.202 | 3.635 | 3.807 | 3.748 | 5.224 |
| AlignCos | 0.060 | 0.060 | 0.061 | 0.014 | 0.833 | 0.296 | 2.405 | 0.325 | 0.275 | 6.140 | 0.409 | 0.424 | 0.998 | 2.604 |
| AngShape | 0.041 | 0.042 | 0.043 | 0.001 | 0.794 | 0.224 | 2.409 | 0.330 | 0.292 | 6.580 | 0.431 | 0.386 | 1.062 | 2.686 |
| CKA | 0.035 | 0.032 | 0.034 | 0.001 | 0.114 | 0.155 | 0.415 | 5.196 | 6.301 | 110.376 | 7.235 | 10.264 | 0.333 | 0.743 |
| ConcDiff | 0.010 | 0.010 | 0.011 | 0.002 | 0.083 | 0.055 | 0.125 | 0.111 | 0.083 | 1.230 | 0.218 | 0.268 | 0.034 | 0.055 |
| DistCorr | 2.272 | 2.141 | 2.281 | 0.018 | 1.182 | 1.202 | 2.945 | 6.038 | 6.256 | 82.490 | 10.742 | 10.871 | 3.215 | 4.783 |
| EOS | 23.954 | 22.028 | 24.517 | 0.029 | 5.210 | 4.963 | 14.802 | 11.148 | 12.435 | 32.257 | 23.746 | 16.018 | 20.656 | 30.714 |
| GULP | 0.042 | 0.043 | 0.044 | 0.001 | 0.990 | 0.393 | 3.527 | 0.400 | 0.361 | 9.583 | 0.512 | 0.538 | 0.754 | 1.945 |
| HardCorr | 0.025 | 0.023 | 0.025 | 0.001 | 0.251 | 0.120 | 1.427 | 0.170 | 0.168 | 2.906 | 0.254 | 0.258 | 0.185 | 0.393 |
| IMD | 248.439 | 250.130 | 261.883 | 34.068 | 163.663 | 184.229 | 113.272 | 742.055 | 920.261 | 4231.700 | 549.456 | 582.878 | 517.934 | 587.420 |
| Jaccard | 3.236 | 3.248 | 3.376 | 0.060 | 1.715 | 1.851 | 3.893 | 1.871 | 1.731 | 3.822 | 2.475 | 2.557 | 2.434 | 3.302 |
| LinReg | 0.084 | 0.043 | 0.050 | 0.002 | 0.709 | 0.546 | 14.355 | 1.048 | 0.440 | 15.250 | 0.627 | 0.603 | 2.666 | 6.028 |
| MagDiff | 0.001 | 0.001 | 0.001 | 0.000 | 0.005 | 0.005 | 0.012 | 0.009 | 0.007 | 0.096 | 0.018 | 0.023 | 0.005 | 0.007 |
| OrthProc | 0.022 | 0.023 | 0.024 | 0.001 | 0.792 | 0.218 | 2.207 | 0.251 | 0.222 | 5.486 | 0.324 | 0.280 | 0.839 | 2.196 |
| PWCCA | 0.182 | 0.265 | 0.272 | 0.003 | 3.875 | 1.532 | 19.657 | 1.155 | 1.154 | 26.363 | 1.898 | 3.031 | 2.474 | 6.806 |
| PermProc | 0.028 | 0.031 | 0.029 | 0.000 | 0.341 | 0.240 | 4.177 | 0.213 | 0.195 | 5.224 | 0.319 | 0.302 | 0.305 | 0.599 |
| ProcDist | 0.020 | 0.021 | 0.021 | 0.001 | 0.771 | 0.211 | 2.215 | 0.229 | 0.208 | 5.421 | 0.301 | 0.262 | 0.852 | 2.163 |
| RSA | 19.815 | 21.201 | 21.294 | 0.146 | 8.518 | 10.629 | 16.143 | 12.629 | 13.931 | 60.452 | 17.195 | 17.672 | 12.687 | 16.993 |
| RSMDiff | 240.992 | 228.653 | 239.948 | 0.177 | 56.854 | 59.778 | 69.778 | 82.993 | 126.385 | 403.597 | 140.079 | 117.119 | 113.932 | 153.379 |
| RTD | 15.165 | 91.340 | 672.182 | 113.033 | 287.200 | 304.870 | 293.158 | 14.159 | 21.799 | 29.058 | 171.138 | 198.786 | 33.374 | 214.915 |
| RankSim | 3.362 | 3.290 | 3.499 | 0.063 | 1.770 | 1.915 | 3.554 | 2.078 | 1.942 | 3.969 | 2.782 | 2.762 | 2.693 | 3.739 |
| SVCCA | 0.585 | 0.544 | 0.611 | 0.003 | 3.087 | 1.861 | 9.654 | 2.340 | 1.696 | 23.019 | 3.484 | 3.168 | 3.456 | 9.398 |
| SoftCorr | 0.022 | 0.020 | 0.022 | 0.001 | 0.059 | 0.043 | 0.477 | 0.147 | 0.150 | 2.472 | 0.229 | 0.222 | 0.151 | 0.285 |
| UnifDiff | 11.056 | 10.420 | 12.626 | 0.045 | 19.209 | 20.578 | 53.692 | 9.283 | 8.139 | 28.615 | 17.543 | 14.930 | 19.779 | 33.224 |

