# OpenReview forum: "ReSi: A Comprehensive Benchmark for Representational Similarity Measures"
_ICLR.cc/2025/Conference — ICLR 2025 Poster_

### Official Review · Reviewer_5jzy · 2024-10-28

**Soundness:** 4
**Presentation:** 4
**Contribution:** 4
**Rating:** 8
**Confidence:** 4

**Summary:**

The paper introduces ReSi, a comprehensive bechmark for comparing neural representations.
The benchmark consists of (i) six carefully designed tests for similarity measures, (ii) 23 similarity measures, (iii) twelvecneural network architectures, and (iv) six datasets, spanning over the graph, language, and vision domains.
The benchmark is extensible; future research can build on it and further expand it.
The source code is publicly available.

**Strengths:**

1. Novelty. To the best of my knowledge, authors have introduced the first representational similarity bechmark.
2. Inside the benchmark, several rarely used similarity measures were tested (like Jaccard similarity), however, they showed promising results.
3. An in-depth analysis is provided; cases where some similarity measures tend to work well as studied.
4. The benchmark is quite diverse: in includes datasets from spanning graph, language, and vision domains.
5. The language of the paper is fine and it's easy to follow.
6. Impact. The comparison of neural representational similarity measures is a long-standing problem in deep learning.

**Weaknesses:**

1. Authors cite (Barannikov et. al, 2022.) several times as an example of representation comparison measure and experimental methodology, but don't include the proposed measure (RTD) into comparisons.
2. Stitching (Bansal, 2021) is not included intro the study.
3. Some important details of experiments are not disclosed: for example, CKA could be evaluated on the whole dataset or batches. Other similarity measures probably include hyperparameters as well.
4. The study might benefit from an Appendix with a brief description of all similarity measures under evaluation.

Bansal, Y., Nakkiran, P., & Barak, B. (2021). Revisiting model stitching to compare neural representations. Advances in neural information processing systems, 34, 225-236.

**Questions:**

1. Can you please clarify notations at Figure 3? Does boxes and bars mean quantiles?

---

> ### Author Response · Authors · 2024-11-21
> **Rebuttal by Authors**
>
> We are very grateful for your thoughtful and appreciative feedback. Your thorough review of our work is valued greatly.
>
> > ### Weaknesses:
> > **W1:** Authors cite (Barannikov et. al, 2022.) several times as an example of representation comparison measure and experimental methodology, but don't include the proposed measure (RTD) into comparisons.
>
> **R1:** We have added RTD to the benchmark. We currently show partial results in the updated pdf. Computation is still ongoing as RTD has a large variance in its runtime, ranging from 20 seconds to more than two hours per comparison.
>
> > **W2:** Stitching (Bansal, 2021) is not included intro the study.
>
> **R2:** While Stitching is an interesting way of measuring the similarity between two models, it is not directly based on representations, rendering it out-of-scope as a measure for this study. Moreover, stitching would require solving the question of what should be considered “maximally similar” (Is it when accuracy does not decay? But what happens when it increases?) and what would be most dissimilar (when is it 0?). Due to these points, we did not consider stitching.
>
> > **W3:** Some important details of experiments are not disclosed: for example, CKA could be evaluated on the whole dataset or batches. Other similarity measures probably include hyperparameters as well.
>
> **R3:** While we have provided some details of hyperparameter choices for similarity measures in Section 3.2, we have indeed missed addressing that for CKA, we use the standard version that considers the whole dataset. Due to space constraints, we have moved descriptions of the hyperparameter choices for similarity measures to Appendix A.2, and included the choice of the non-batched version of CKA at this point.
>
> > **W4:** The study might benefit from an Appendix with a brief description of all similarity measures under evaluation.
>
> **R4:** We have added Appendix A, in which we provide explicit definitions for all the measures under study.
>
> > ### Questions:
> > **Q1:** Can you please clarify notations at Figure 3? Does boxes and bars mean quantiles?
>
> **A1:** The boxes in the boxplot show the ranks between the 25th and 75th percentile for the measure (the IQR). The whiskers extend to 1.5 times the IQR from the 25th/75th percentile or the highest/lowest rank if it is less than 1.5 IQRs away from the 25th/75th percentile. We have clarified the caption.

---

> > ### Comment · Reviewer_5jzy · 2024-11-25
> > **Response**
> >
> > Thank you for a detailed answer, doing additional experiments and expanding a paper.
> > My issues are addressed.
> > Indeed, your paper fills a gap in a study of neural representations.
> > I am raising Soundness,Presentation, and Contribution scores.

---

### Official Review · Reviewer_jBQa · 2024-10-29

**Soundness:** 3
**Presentation:** 3
**Contribution:** 3
**Rating:** 6
**Confidence:** 3

**Summary:**

This paper proposes a suite of benchmarking tasks to test the quality of a representational similarity metric. The tasks are designed around training models such that they are different in one specific way that leads to a certain expectation for similarity. Specifically, the authors propose 6 tasks:

(T1): Correlating similarity vs accuracy. Models are trained with only varied seeds.
(T2): Correlating similarity to actual model outputs.  Models are trained with only varied seeds.
(T3): Label Randomization: Randomizing different percentages of the labels. Models are expected to be more similar within the same percentage of randomization.
(T4): Shortcut affinity: Models are trained with random "shortcuts" that leak the correct label. The shortcut that always leaks the correct label, sometimes leaks the correct label and never leaks the correct label. Models should be more similar within group.
(T5): Augmentation: models are trained with different kinds of augmentations adn models are expected to be more similar within the same type of augmentations.
(T6): Layer Monotonicity: The similarity between layers is expected to be related to the distance between the layers.

They apply their benchmarking tasks to three domains, graphs, language and vision and test 23 different representational similarity methods.

The authors find that no similarity metric is always the best according to these metrics, however, there are some that tend to perform better than others, especially within a specific sub-domain (graph, language, or vision).

**Strengths:**

(S1) The goal of the paper is well-motivated and could be a valuable contribution to the community, since it is not clear how to compare representational similarity methods.

(S2) The problem is quite challenging as it is hard to know exactly how two models differ, therefore the authors take reasonable steps to develop tasks in which they can approximately group models by differences in their performance or training method.

(S3) The authors apply their benchmarking method on a comprehensive set of representational similarity methods in three popular domains in machine learning.

(S4) The authors provide some interesting insights into differences between similarity methods and show that no single similarity method always performs the best.

**Weaknesses:**

(W1) Task 6 - I find that there are some careful considerations to use task 6 (see Q1). If the layers are chosen incorrectly, task 6 would have some issues for at least some vision models since skip connections could make neighboring layers dissimilar. Correcting/explaining this issue would shift my score to a 6.

(W2) I found the depth of analysis to be somewhat limited. The authors could significantly strengthen the paper by providing some deeper analysis into a few similarity methods that explains how the similarity methods arrive at different similarity scores when comparing the same representations. For example, the authors show the impact of preprocessing by comparing two similarity methods that differ only in preprocessing: Orthogonal Procrustes (OrthProc) and Procrustes Size-and-Shape Distance (ProcDist). Can the authors analyze what the preprocessing does to the representation and how it changes the results of the method on the benchmarks? If this analysis and/or others are provided, I would consider improving the rating further.

Minor ----
(W3) Section 3.1 should indicate more details of the tasks will be provided in Sec 3.5.

**Questions:**

Q1. Which layers are being used in the vision model experiments? I find that this information is not clearly provided in Sec. A.5 Vision.

---

> ### Author Response · Authors · 2024-11-21
> **Rebuttal by Authors**
>
> Thank you very much for your thoughtful feedback and suggestions. Your thorough review of our work is highly appreciated.
>
> > **(W1)** Task 6 - I find that there are some careful considerations to use task 6 (see Q1). If the layers are chosen incorrectly, task 6 would have some issues for at least some vision models since skip connections could make neighboring layers dissimilar. Correcting/explaining this issue would shift my score to a 6.
>
> **R1:** This is absolutely correct! Moreover, this issue does supersede vision models and also affects NLP Transformer models, which commonly feature skip-connections. To avoid this issue, we paid close attention to only extract representations at locations where no residual exists. Currently, this was mentioned only in Appendix A1.3 (Former lines 943-947) “In order to measure representational similarity, representations need to be extracted at different positions in the architecture. In order to capture the full system behavior of a model, representations are only extracted at positions where no skip-connections exist. For VGG networks, this would not narrow down choices due to lack of skip connections, yet for ResNets and ViTs, this narrows the pool of choices where representations can be extracted”.
>
> To address this issue, we have added a footnote in the description of Test 6 which notes that skip connections would violate our assumptions, but that we have controlled for this issue. We further reference the reworked Appendix B.1 ‘Representation Extraction’ which explains this for all modalities.
>
> > **(W2)** I found the depth of analysis to be somewhat limited. The authors could significantly strengthen the paper by providing some deeper analysis into a few similarity methods that explains how the similarity methods arrive at different similarity scores when comparing the same representations. For example, the authors show the impact of preprocessing by comparing two similarity methods that differ only in preprocessing: Orthogonal Procrustes (OrthProc) and Procrustes Size-and-Shape Distance (ProcDist). Can the authors analyze what the preprocessing does to the representation and how it changes the results of the method on the benchmarks? If this analysis and/or others are provided, I would consider improving the rating further.
>
> **R2:** In the specific example where preprocessing between OrthProc and ProcDist differs, it is the case that OrthProc, in contrast to ProcDist, normalizes the compared representations to unit norm. If the performance on specific representations was consistently worse after this preprocessing, this could be an indicator that the absolute scale of each axis in the representations carries semantic meaning, which the Procrustes measure would need to pick up on. Conversely, if rescaling improved performance consistently, this could be an indicator that the overall scale of the axes could be considered noise. However, there were hardly such overall consistent trends that we could identify directly, so we cannot draw general conclusions regarding this difference in preprocessing, but we have adapted the paragraph on preprocessing in Section 5 to clarify this example.
> More generally, deeper analyses regarding why similarity measures achieve specific performances, are, unfortunately, hard to make. This is due to (i) the overall opaque nature of neural representations, which make it hard to analyze their interplay with similarity measures, and (ii) a lack of general patterns in the results, such as measures with similar properties performing well together, based on which such conclusions could be made.
> Yet, we have added a paragraph in Section 5 of our revision called “Potential for Deeper Insights”, in which we briefly address this issue.
>
>
> > **Minor ---- (W3)** Section 3.1 should indicate more details of the tasks will be provided in Sec 3.5.
>
> **R3:** At the beginning of Section 3.1, we have added a sentence clarifying that in all tests, models are trained for classification tasks.
>
> > ### Questions:
> > Q1. Which layers are being used in the vision model experiments? I find that this information is not clearly provided in Sec. A.5 Vision.
>
> **A1:** Please see Response **R1**.
>
> We hope to have addressed your questions and concerns adequately. Should no questions remain, we would appreciate it if you considered adapting your score.

---

> > ### Comment · Reviewer_jBQa · 2024-11-24
> >
> > Thank you for your response, my questions are answered. I have raised my score to a 6.

---

### Official Review · Reviewer_KrQ9 · 2024-11-02

**Soundness:** 3
**Presentation:** 4
**Contribution:** 2
**Rating:** 6
**Confidence:** 4

**Summary:**

This paper presents the ReSi benchmark, a framework for assessing representational similarity measures across neural architectures from different domains. It features 23 similarity measures, twelve neural network architectures, and various datasets from graph, language, and vision. ReSi is proposed to support research on representation learning, reproducibility, and exploration of new approaches for comparing neural representations.

**Strengths:**

- The challenge of understanding learned representations better, especially in an unsupervised way is highly relevant and of practical importance.
- Reproducible and extensive experiments covering a wide range of metrics, domains and tasks.
- The paper is overall well-written and easy to follow. The figures and findings are clearly presented.

**Weaknesses:**

- I think the paper misses the opportunity to give the reader a better understanding of what drives the differences and inconsistent results of measuring similarity. Which types of measures perform well together and what feature of the representation is this driven by. Relevant aspects could be due to the way representations are normalized, what arithmetic function is used for computing the distance (e.g. dot-product, summed differences, etc.)
- Another angle to better understand similarity computations is from the point of interpretability, i.e. what features do the respective models pick up on and how do these change across models. I think this is an important and complementary direction to measuring different similarity scores and should be discussed (cf. [1,2,3]).
- ReSi is presented as the “the first comprehensive benchmark for representational similarity” and can “yield rich insights” and a “theoretical understanding of representational similarity”. While I did find the general guidelines interesting and helpful, I think a deeper understanding of what causes these differences across metrics and domains is missing, and thus the above statements overstate the level of achievable insights from ReSi for me.


[1] Eberle et al. “Building and interpreting deep similarity models”, IEEE Transactions on Pattern Analysis and Machine Intelligence, 44(3), 2020.

[2] Janizek et al. "Explaining explanations: Axiomatic feature interactions for deep networks", CoRR, 2020.

[3] Vasileiou et al. “Explaining Text Similarity in Transformer Models”, NAACL 2024.

**Questions:**

- Should Eq. 3 map to R^N?
- It did not become clear initially what type of accuracy measures are considered; e.g. what task the model is trained to solved?
- I think coloring the labels of the similarity measures in Table 3 may be helpful for the reader (similar to the coloring in Fig. 3)

---

> ### Author Response · Authors · 2024-11-21
> **Rebuttal by Authors**
>
> Thank you very much for your thoughtful feedback and suggestions. We appreciate the time and effort you have dedicated to reviewing our work.
>
> > ### Weaknesses
> > **W1:** I think the paper misses the opportunity to give the reader a better understanding of what drives the differences and inconsistent results of measuring similarity. Which types of measures perform well together and what feature of the representation is this driven by. Relevant aspects could be due to the way representations are normalized, what arithmetic function is used for computing the distance (e.g. dot-product, summed differences, etc.)
>
> **R1:** While we also believe that such explanations would strongly improve our paper, we do not agree that such specific explanations can be given at this point. In general, neural representations are very opaque by their nature, and the limited understanding of what features or differences in representations influence the similarity measures were one of the main motivations to build our benchmark in the first place.
> While trends can be identified in limited cases within our results, such as neighborhood-based measures performing well in the graph domain—which may indicate that GNNs will form similar neighborhoods in their representations when trained on similar objectives—it is still not directly clear what specific mechanic in the GNN models this behavior can be attributed to.
> Yet, we believe the tests implemented in ```ReSi``` could enable some more systematic analyses in this direction. For instance, one could conduct studies in which parameters such as the distance functions used in RSM-based similarity measures, or neural network  parameters such as activation functions or normalization layers could be varied in a controlled measure, and the tests from our benchmark could serve as a tool to identify whether parameter choices would yield consistently better results. This, however, we would consider to be future work.
> We have added a new paragraph to Section 5 in which we discuss how ```ReSi``` could be used as a tool for future research.
>
> > **W2:** Another angle to better understand similarity computations is from the point of interpretability, i.e. what features do the respective models pick up on and how do these change across models. I think this is an important and complementary direction to measuring different similarity scores and should be discussed (cf. [...]).
>
> **R2:** We agree that this is an interesting direction and have highlighted this as a direction for future work in the manuscript. For example, it is known that models that rely on different input features still produce representations that are seen as highly similar by CKA [1]. Thus, an additional test could focus on the extent that representational similarity measures capture similarity in usage of input features. However, a procedure to develop models with reliance on certain input features would be required. Our augmentation, shortcut, and memorization tests do this implicitly to varying degrees.
>
> > **W3:** ```ReSi``` is presented as the “the first comprehensive benchmark for representational similarity” and can “yield rich insights” and a “theoretical understanding of representational similarity”. While I did find the general guidelines interesting and helpful, I think a deeper understanding of what causes these differences across metrics and domains is missing, and thus the above statements overstate the level of achievable insights from ```ReSi``` for me.
>
> **R3:** We have reworked the corresponding paragraph to more accurately state that ```ReSi``` can (i) serve as test environment for new measures, (ii) provide guidance for choosing measures in an application at hand, and (iii) serve as a tool to inspire and assist future research efforts.
>
> > ### Questions:
> > **Q1:** Should Eq. 3 map to R^N?
>
> **A1:** No, representational similarity measures always produce scalar values. We have adapted Section 2 to further clarify this.
>
> > **Q2:** It did not become clear initially what type of accuracy measures are considered; e.g. what task the model is trained to solved?
>
> **A2:** Indeed, we have missed clarifying this prior to describing our tests, and have added a corresponding sentence in the beginning of Section 3.1. We compare classification models in all tests.
>
> > **Q3:** I think coloring the labels of the similarity measures in Table 3 may be helpful for the reader (similar to the coloring in Fig. 3)
>
> **A3:** We have rearranged all tables in the paper such that measures are now grouped by their respective categories. We believe that this way, it will be easier to connect the groups in Figure 3 to each of these categories.
>
>
> ### References:
>
> [1] Jones, H. T., Springer, J. M., Kenyon, G. T., & Moore, J. S. (2022, August). If you’ve trained one you’ve trained them all: inter-architecture similarity increases with robustness. In *Uncertainty in Artificial Intelligence* (pp. 928-937). PMLR.

---

> > ### Comment · Reviewer_KrQ9 · 2024-11-24
> > **Rebuttal response**
> >
> > Thanks for the detailed feedback and changes made to the manuscript. I think these have improved the manuscript, and I have increased my score for presentation.

---

### Official Review · Reviewer_vg5P · 2024-11-03

**Soundness:** 3
**Presentation:** 3
**Contribution:** 2
**Rating:** 6
**Confidence:** 4

**Summary:**

This work presents a comprehensive benchmark for evaluating 23 representational similarity measures across six similarity tests, twelve different architectures, and six datasets from diverse domains. Additionally, the paper offers insights for future research and aims to democratize a unified framework that ensures fair comparisons across different studies.

**Strengths:**

* The paper is well-written and easy to follow, with experiments clearly presented and tasks well-defined. Figures and tables are self-explanatory, significantly enhancing the manuscript's readability and accessibility.

* The authors have provided the code, ensuring full reproducibility of results and facilitating the community's ability to expand and build upon the benchmark. This openness supports transparency and encourages collaborative advancement in the field.

* The problem addressed is highly relevant and important for the community, as it remains an open question which metric is best suited for different scenarios.

**Weaknesses:**

* It’s unclear whether the similarities are calculated between two latent spaces (as suggested in formula 3, line 094) or between two individual representations (a single point), as written in line 095. Clarifying this distinction (and how each metric is adapted accordingly) could improve the reader's understanding. It seems to be between two latent spaces, but in Figure 2 (left), the metric is calculated between two points.

* When evaluating the text architectures:
  * Could you clarify the choice to test the same model with 25 different seeds, rather than exploring architectural variations like RoBERTa or ALBERT, which are available pretrained on Hugging Face? Including a wider variety of architectures might enhance the robustness and generalizability of the findings.
  * It would be helpful to either move the explanation of the choice to use the CLS token to the main manuscript or to reference the appendix. Additionally, you might consider testing alternative aggregation modalities, such as the mean or the last token, since these can yield different representations and may impact the results.

* The vision task may benefit from broader testing since only ImageNet with 100 subclasses is used. Expanding the evaluation to include the full ImageNet dataset or other widely-used datasets, such as ImageNet-1k, CIFAR-10, or CIFAR-100, could provide a more comprehensive assessment.

* The analysis could benefit from stronger takeaways. For instance, in lines 438-439, you suggest that the choice of measures should be task-specific, and in lines 453-455, you advocate for the development of best practices for using similarity metrics. However, these insights feel somewhat broad, and more concrete guidance could help readers understand how to apply the benchmark effectively.

* In line 413, it’s mentioned that some domain-specific trends can be identified, yet later (line 424), you note that no single measure consistently outperforms others across all tests, even within a single dataset. The first claim—that some metrics appear more suitable for specific modalities—is interesting and would benefit from a more in-depth discussion. Also, adding a summary table of these findings of the paper could help readers better understand the details.

**Questions:**

* What happens with different architectures for the text modality (e.g. some LLMs)?
* What is "IN21k" in line 282? Please write out the full name instead of using the acronym, as it’s unclear without a reference to the actual name.

---

> ### Author Response · Authors · 2024-11-21
> **Rebuttal by Authors (Part 1/2)**
>
> Thank you very much for your thoughtful and detailed feedback. We appreciate the time and effort you have dedicated to reviewing our work.
>
> > ### Weaknesses:
> > **W1:** It’s unclear whether the similarities are calculated between two latent spaces (as suggested in formula 3, line 094) or between two individual representations (a single point), as written in line 095. Clarifying this distinction (and how each metric is adapted accordingly) could improve the reader's understanding. It seems to be between two latent spaces, but in Figure 2 (left), the metric is calculated between two points.
>
> **R1:** We understand the confusion resulting from the mismatch between the formal definitions and the graphical illustration in Figure 2. Indeed, it is the case that representational similarity measures compare full matrices. For the illustration, we faced the issue that such matrices, however, cannot be drawn, and thus we had to make this simplification. We have adapted Section 2 to more clearly point out that matrices rather than instance representations are compared. Further, we adapted the headers of the coordinate axes in Figure 2.
>
> >**W2:** When evaluating the text architectures:
> > **W2.1.** Could you clarify the choice to test the same model with 25 different seeds, rather than exploring architectural variations like RoBERTa or ALBERT, which are available pretrained on Hugging Face? Including a wider variety of architectures might enhance the robustness and generalizability of the findings.
>
> **R2.1:** We agree that a wider range of architectures would strengthen our work. We have added ALBERT and are currently exploring ways to add a decoder-only LLM as well.
>
> Particularly for tests grounded by design, we aim to have models that mainly differ in a precise way, i.e., mostly through the changes we impose via training. Comparing models with different loss functions (e.g., BERT vs Roberta) would add confounders—we would not know whether changes in representations would be due to, e.g. different shortcuts, or rather due to a difference in loss function. Thus, we constrained our comparisons to fixed architectures. We focused on BERT, because the resource of pre-trained models allowed for more diversity in our final fine-tuned models, thus making the tests more challenging. Fine-tuning with different seeds from a single fixed pretraining model likely leads to only small changes in models [1], which could make the groups too homogenous.
> Because ALBERT shares parameters over layers, fine-tuning should lead to sufficiently different models even when starting from the same model.
>
> We have also clarified our reasoning in Appendix B.2.
>
> > **W2.2:** It would be helpful to either move the explanation of the choice to use the CLS token to the main manuscript or to reference the appendix. Additionally, you might consider testing alternative aggregation modalities, such as the mean or the last token, since these can yield different representations and may impact the results.
>
> **R2.2:**  We have added results for mean-pooled representations across tokens and added a brief explanation to the main text, as well as a reference to Appendix B.1 for more details. For our current models, the final token has no particular role. We thus do not compare these representations. However, for a decoder-based LLM, the final token would take the role of the CLS token and thus be important.
> In our preliminary results, the representation type seems to influence the scores, but measures that perform well on one representation also perform well on the other.
>
> > **W3:** The vision task may benefit from broader testing since only ImageNet with 100 subclasses is used. Expanding the evaluation to include the full ImageNet dataset or other widely-used datasets, such as ImageNet-1k, CIFAR-10, or CIFAR-100, could provide a more comprehensive assessment.
>
> **R3:** To increase the breadth of vision datasets, we included the CIFAR-100 dataset. While not all are fully finished yet, we will provide a fully populated table of this for the camera-ready version of the paper. Preliminary tables with the quicker-to-calculate measures are already provided in our revision in Appendix C.3.

---

> ### Author Response · Authors · 2024-11-21
> **Rebuttal by Authors (Part 2/2)**
>
> > **W4:** The analysis could benefit from stronger takeaways. For instance, in lines 438-439, you suggest that the choice of measures should be task-specific, and in lines 453-455, you advocate for the development of best practices for using similarity metrics. However, these insights feel somewhat broad, and more concrete guidance could help readers understand how to apply the benchmark effectively.
>
> **R4:** While we have provided some “general recommendations” for the choice of measures based on our results in a corresponding paragraph in Section 5, you have a point that we have not exhaustively laid out how our benchmark could be used. Thus, we have adapted the introduction as well as the contributions paragraph in Section 6 to more accurately state that ```ReSi``` can be used as (i) a baseline for new measures to test against, (ii) a reference to guide the choice of measures in specific applications at hand, and (iii) a tool to inspire and assist future research efforts. We have also added a paragraph to Section 5 in which we more specifically address how ```ReSi``` can assist in aspect (iii).
> Unfortunately, additional recommendations beyond those in Section 5 are difficult to provide. This is due to the lack of clear trends regarding which measures perform well across multiple settings, and providing nuanced recommendations for individual datasets, architectures, or test scenarios would result in a lengthy discussion that may be harder to digest than the result tables already included in Appendix C.
>
> > **W5:** In line 413, it’s mentioned that some domain-specific trends can be identified, yet later (line 424), you note that no single measure consistently outperforms others across all tests, even within a single dataset. The first claim—that some metrics appear more suitable for specific modalities—is interesting and would benefit from a more in-depth discussion. Also, adding a summary table of these findings of the paper could help readers better understand the details.
>
> **R5:** We agree that an in-depth discussion on the reasons for these domain-specific differences would be a great addition to our paper. Unfortunately, given that there are hardly any consistent trends regarding which types of measures perform well, it is hard to pinpoint shared mechanics of measures as reasons for good performances. Without more consistent patterns, we would rather refrain from drawing speculative connections.
> However, we could identify a trend that on the graph domain, neighborhood-based measures appear to perform relatively well. We interpret this as an indicator that for graph neural networks, the structure of their representations, or more specifically the local neighborhoods, are strongly driven by the training objective, remaining similar when the objective is similar, and diverging when objectives differ.
> We have added a new paragraph (Potential for Deeper Insights) to Section 5 in which we discuss this finding, and also further illustrate how ```ReSi``` can be used as a tool to obtain more understanding of representational similarity.
> Finally, regarding the table, we agree that this would be nice to have, but given that recommendations would be either very general or extremely nuanced, as discussed in the reply to your previous point, we did not see a suitable way for such a tabular presentation, also considering the space constraints. In case you have suggestions, we would, however, be very open to discuss and implement these.
>
> > ### Questions:
> > **Q1:** What happens with different architectures for the text modality (e.g. some LLMs)?
>
> **A1:** We are currently investigating to what extent we can produce LLMs for our tests given our computational constraints. Already, we show partial new results for ALBERT models, which are architecturally similar to BERT, but different in sharing parameters across layers. We observe similar to the other domains that scores differ substantially across architectures.
>
>
> > **Q2:** What is "IN21k" in line 282? Please write out the full name instead of using the acronym, as it’s unclear without a reference to the actual name.
>
> **A2:** IN21k refers to the ImageNet-21K dataset. We replaced the corresponding abbreviation accordingly to *weights from ImageNet-21k (IN21k) (Deng et al., 2009)*---See end of Section 3.3.
>
> We hope to have addressed your questions and comments satisfactorily. Should no questions remain, we would appreciate it if you could consider raising your score.
>
>
> ### References:
> [1] [What Happens To BERT Embeddings During Fine-tuning?](https://aclanthology.org/2020.blackboxnlp-1.4) (Merchant et al., BlackboxNLP 2020)

---

> > ### Comment · Reviewer_vg5P · 2024-11-25
> >
> > I appreciate the detailed feedback and the changes the authors have made to the manuscript. Based on this, I have increased my score accordingly.

---

### Author Response · Authors · 2024-11-21
**General Response**

We thank all the reviewers for their insightful and constructive comments, which helped us improve our paper even further.
We updated the manuscript with our newly introduced changes, and marked newly added content in blue text and places where existing text was modified in magenta. These changes, among others, include extensions of our current experiments for the language domain to the ALBERT architecture and different token representations, extensions of our vision experiments to a second dataset (CIFAR-100), as well as textual changes intended to enhance clarity.

Regarding any new or outstanding points of contention, we look forward to discussing them.

---

> ### Author Response · Authors · 2024-11-28
> **End of Revision Period**
>
> We have updated our manuscript once more to include the most recent results. This includes additional results with RTD and ALBERT as well as first results using an LLM (SmolLM2-1.7B) for the language domain.

---

### Meta-Review · Area_Chair_hmis · 2024-12-23

**Metareview:**

This paper presents ReSi, described as the first comprehensive benchmark for evaluating representational similarity measures, including 6 carefully designed tests, 23 similarity measures, 12 neural architectures, and 6 datasets across graph, language and vision domains. The key claims are that ReSi enables systematic evaluation and comparison of representational similarity measures, with experimental results showing that no single measure consistently outperforms others across all tests and domains. The paper's strengths include strong practical relevance for the field, extensive experimental validation across multiple domains, good reproducibility through public code, and clear presentation. Initial weaknesses included: lack of clarity around calculation methods (between latent spaces vs individual points), limited architecture variety in text domain, restricted vision dataset testing, and insufficient depth in analysis of why different measures perform differently. During rebuttal, the authors made substantial improvements by adding ALBERT and LLM architectures, including CIFAR-100 dataset, clarifying technical details, and expanding analysis sections. The review scores (6, 6, 6, 8) and thorough responses to reviewer concerns, particularly around experimental breadth and technical clarity, support accepting this paper

**Additional Comments On Reviewer Discussion:**

The reviewers raised several key technical and presentation issues that led to constructive discussion. Reviewer vg5P requested clarification on similarity calculations and broader architecture/dataset coverage - the authors responded by adding ALBERT architecture experiments, CIFAR-100 dataset testing, and clarifying technical details. Reviewer KrQ9 pointed out missed opportunities for deeper analysis of what drives differences between measures - while the authors acknowledged limitations in providing definitive explanations due to the opaque nature of neural representations, they added new discussion sections about potential future research directions. Reviewer jBQa raised concerns about layer selection in vision models and skip connections - the authors clarified their careful consideration of these issues and added explicit documentation. Reviewer 5jzy suggested including additional similarity measures - the authors added RTD measure to their benchmark and clarified why certain measures like Stitching were out of scope. The authors were highly responsive during the rebuttal period, making substantial additions to experiments and revising the manuscript to address concerns. This led multiple reviewers (vg5P, KrQ9, jBQa) to increase their scores. The thorough engagement with reviewer feedback and meaningful improvements to the work support the final accept recommendation.

---

### Decision · Program_Chairs · 2025-01-22

Accept (Poster)